# Targeted activation of midbrain neurons restores locomotor function in mouse models of parkinsonism

Débora Masini [1] & Ole Kiehn [1,2] ✉

The pedunculopontine nucleus (PPN) is a locomotor command area containing glutamatergic neurons that control locomotor initiation and maintenance. These motor actions are deficient in Parkinson's disease (PD), where dopaminergic neurodegeneration alters basal ganglia activity. Being downstream of the basal ganglia, the PPN may be a suitable target for ameliorating parkinsonian motor symptoms. Here, we use in vivo cell-type specific PPN activation to restore motor function in two mouse models of parkinsonism made by acute pharmacological blockage of dopamine transmission. With a combination of chemo- and opto-genetics, we show that excitation of caudal glutamatergic PPN neurons can normalize the otherwise severe locomotor deficit in PD, whereas targeting the local GABAergic population only leads to recovery of slow locomotion. The motor rescue driven by glutamatergic PPN activation is independent of activity in nearby locomotor promoting glutamatergic Cuneiform neurons. Our observations point to caudal glutamatergic PPN neurons as a potential target for neuromodulatory restoration of locomotor function in PD.

[1] Department of Neuroscience, Faculty of Health and Medical Sciences, University of Copenhagen, Blegdamsvej 3B, 2200 Copenhagen, Denmark. [2] Department of Neuroscience, Karolinska Institutet, Stockholm, Sweden. ✉email: Ole.Kiehn@sund.ku.dk

The ability to execute movement is essential for survival and at the core of most behaviors. Manifestation of well-adapted motor output is contingent upon circuits specialized in planning, selection, and generation of motor commands. Without a proper performance of these circuits, movement becomes dysfunctional and maladaptive. Parkinson's disease (PD) is characterized by the death of midbrain dopaminergic neurons. This hypodopaminergic disorder affects basal ganglia circuitries associated with selection and planning of movement, and as a result, patients suffer from severe motor impairment, including tremor, akinesia (lack of movement), bradykinesia (slowness of movement), and difficulty in initiation of voluntary movement (freezing of gait or delayed initiation of a motor plan)[1,2,3,4]. These motor disabilities result from alterations in network activity within the basal ganglia (BG)[1,2].

The direct and indirect pathways within the BG, which originate from striatal GABAergic medium spiny projection neurons (MSNs), promote or inhibit movement, respectively, and work in concert to facilitate desired movements while simultaneously suppressing unwanted motor actions[5–7]. Dopamine plays an essential role in the striatum by increasing the excitability of the MSNs of the direct pathway via D1 dopamine receptors and opposing activity of MSNs of the indirect pathway via D2 receptors[8]. Consequently, loss of dopaminergic signaling increases the D2-MSNs and concomitantly decreases the D1-MSNs activity enhancing BG-output inhibition over its targets, including brain areas involved in the generation of movement[9–11]. Interventions targeting the motor thalamus are beneficial in the control of BG-induced tremor[12], but do not alleviate akinesia, bradykinesia, or freezing of gait in humans[13] suggesting that motor suppression in PD patients, to a large part, occurs due to increased BG-inhibition of brainstem motor areas[11,14–16]. The main brain region essential for the generation and maintenance of motor output is the mesencephalic locomotor region (MLR) located in the brainstem. Anatomically, the MLR contains two nuclei: the cuneiform (CnF) and the pedunculopontine (PPN) nuclei. Rodent studies have demonstrated that glutamatergic neurons in both subregions contribute to locomotor initiation in complementary ways. The activity of glutamatergic neurons in the CnF supports the entire range of speeds, including very fast escape-like locomotion, whereas the PPN favors the exploratory speed range[17–19]. Conversely, short activation of MLR GABAergic neurons can pause ongoing locomotion[18,19], while the cholinergic population, existing only in PPN may modulate locomotor output locally or via its projections to forebrain regions[20,21]. Notably, while the CnF is not substantially interconnected with the BG circuitry, the PPN is under direct BG inhibitory control[18,19,22] with γ-Aminobutyric acid (GABA) being the predominant afferent neurotransmitter within the PPN[23]. Activating the PPN to release it from the excessive BG inhibition in PD is, therefore, a logical step to ameliorate parkinsonian locomotor symptoms. However, when deep brain stimulation targeting the PPN was applied in the clinics it exhibited variable and sometimes no locomotor improvement[24–28].

In the present study, we hypothesize that the lack of consistent clinical results may be caused by the nonspecific electrical stimulation of diverse populations of PPN neurons and/or activation of diverse parts of the PPN. Motor facilitation may be restricted to the caudal PPN in healthy mice[18] and rats[29], parkinsonian animal models[30], and Parkinson's patients[31,32]. While glutamatergic neurons in the rostral PPN evoke whole-body motor arrest[33,34] which has also been reported for nonspecific electrical stimulation of that region in the cat[35]. Here, we, therefore, use cell-type-specific chemogenetic and optogenetic approaches to activate localized PPN subpopulations restricted to the caudal part of the PPN in two distinct mouse models that show strong BG-driven locomotor suppression. To induce motor suppression, we use a pharmacological approach to acutely block dopaminergic signaling by antagonizing dopamine D1 or D2 receptors. As a result, we partially mimic the biochemical signature of PD with increased BG inhibitory action over its targets, and drug-injected mice develop a robust state of akinesia, bradykinesia, and motor response delay. Combining these interventions with quantitative and qualitative behavioral analysis revealed that locomotor proficiency can be rescued by exclusive activation of caudal glutamatergic PPN neurons. This neuronal subpopulation alone can reduce akinesia, normalize speed range, and promote locomotion both over prolonged periods of time and in time-locked episodic events. This phenotypic rescue is independent of activity in the nearby CnF glutamatergic neurons and can be achieved regardless of which drug was initially used to block dopamine transmission. Furthermore, prolonged activation of caudal GABAergic PPN neurons promotes slow-speed locomotor function in mice made parkinsonian by D2-antagonism but does not restore locomotor capabilities in D1-receptor antagonized mice. Our study shows that a parkinsonian motor phenotype can be fully reverted to normal by caudal glutamatergic PPN neuron stimulation and suggests that deep brain stimulation protocols should be tailored to these neurons to facilitate episodic motor output in PD clinical settings.

## Results

**Prolonged activation of caudal glutamatergic pedunculopontine neurons promotes locomotor activity.** To investigate whether sustained activation of glutamatergic PPN neurons can promote locomotor output over an extended period of time we bilaterally injected the caudal PPN of Vglut2[cre] (vesicular glutamate transporter 2)[36] mice with a conditional AAV virus coding for the excitatory Gq-coupled modified human M3 muscarinic receptor (hM3Dgq) (80 nl/hemisphere). We confirmed that labeled neurons were preferentially concentrated in the caudal region of the PPN and exhibited descending and ascending axonal projections to known innervation targets[18,37,38] including the brainstem lateral paragigantocellular nucleus and the substantia nigra pars compacta/pars lateralis (Fig. 1a and S1a).

The hM3Dgq receptor (herein referred to as excitatory DREADD or "eD") is activated by low doses (1 mg/kg, ip) of the chemical actuator Clozapine-N-oxide (CNO). CNO activation of eD in Vglut2-PPN neurons increases neuronal activity as confirmed by higher labeling of the activity-dependent immediate early gene protein product c-Fos in infected cells (Fig. S1b–f) (Colocalized: $76.59 \pm 10.63\%$ [SD], two-tailed, paired $t$-test, $p = 0.0002$, $t = 7.706$, $df = 4$). For behavioral analysis, Vglut2[cre] mice bilaterally injected with eD (Vglut2_eD) were compared to control littermates lacking active viral vectors (Sham).

Vglut2_eD and Sham mice were injected with CNO and five minutes later placed in an Open Field. Distance moved by Sham mice injected with CNO was indiscernible from wild-type animals injected with saline (Total distance: $p = 0.1397$, two-tailed, $t$-test with Welch's correction, $t = 1.699$, $df = 6.059$. Distance over time: Two-way RM ANOVA, interaction $F_{(9, 162)} = 1.131$, $p = 0.3435$). Vglut2_eD mice exhibited increased locomotor activity, measured as distance traveled over a 5-min bin, that peaked 20 min after CNO administration and was maintained throughout the 50 min session (Fig. 1b) (Total distance: $p < 0.0001$; two-tailed, $t$-test, $t = 5.544$, $df = 23$. Distance over time: Two-way RM ANOVA, interaction $F_{(9, 207)} = 5.249$, $p < 0.0001$). The increased distance traveled was associated with changes in two parameters: time locomoting and preferred speed range. Overall, Vglut2_eD mice spent 11% more-time locomoting (>2 cm/s) than Sham mice (Fig. 1c) and the preferred speed

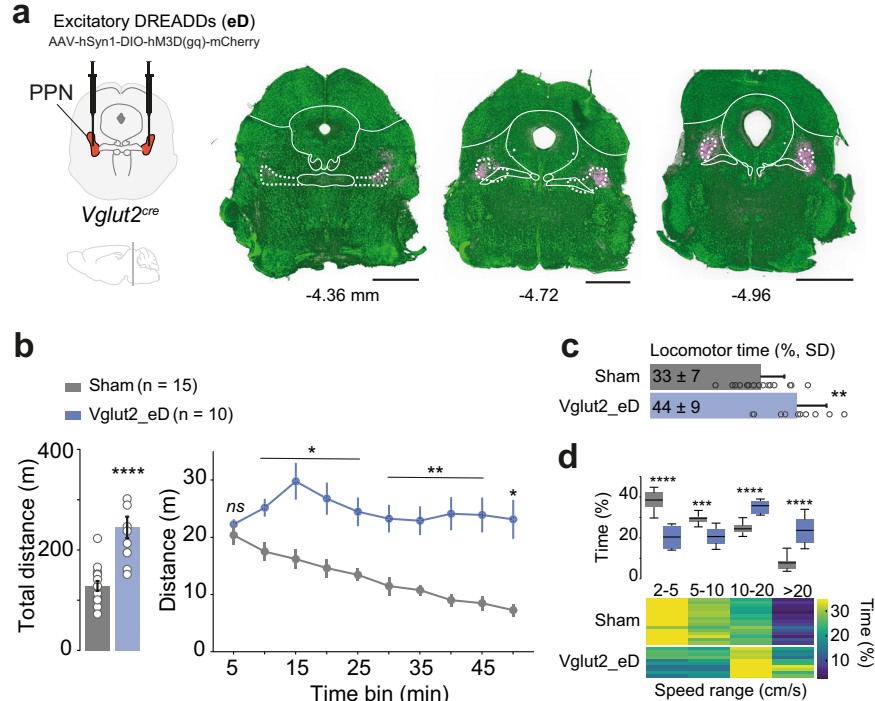

**Fig. 1 Excitatory DREADD activation of caudal glutamatergic PPN neurons increases locomotor output. a** Viral strategy to bilaterally express excitatory DREADDs (eD) in Vglut2-PPN neurons (left). Mapping of segments surrounding the injection site (mm from bregma) showing PPN border (white dashed lines), DREADDs expression (magenta), and general neuronal staining (green). Scale, 1 mm. **b** Total distance moved by Sham (gray) and Vglut2_eD (blue) mice after CNO (bar graph) (two-tailed, t-test). Timeline effect of CNO (line graph, 5 min bins) (Two-way RM ANOVA, Geisser–Greenhouse correction followed by Bonferroni's multiple comparison, report shows post hoc results). Data are presented as mean ± SEM. **c** Percentage of locomotor time (speed above >2 cm/s) during Open Field test (two-tailed, t-test). Data are presented as mean ± SD. **d** Upper panel shows speed range in Sham (gray) and Vglu2_eD (blue) mice with min-to-max boxplots (Two-way RM ANOVA, Geisser–Greenhouse correction, with a report for Bonferroni's multiple comparison). A lower panel heat map shows individual mice in horizontal lines and color code for the percentage of time spent in each speed range. Note Vglut2_eD preference for 10–20 cm/s range. Data composed of 15 Sham$^{CNO}$; 10 Vglut2_eD$^{CNO}$ mice. See detailed stats in Table S1. Source data are provided as a Source Data file.

ranges were 10–20 cm/s (Fig. 1d) (Time locomoting: $p = 0.0029$, two-tailed, t-test, $t = 3.335$, df = 23. Speed ranges: Two-way RM ANOVA interaction $F_{(3, 69)} = 79.79$, $p < 0.0001$ with speeds above 10 cm/s amounting to 32.26% of locomotor time in Sham and 58.94% in Vglut2_eD).

Our data show that bilateral chemogenetic excitation of caudal glutamatergic PPN neurons reliably promotes locomotor activity. We, therefore, proceeded to examine whether glutamatergic PPN neuron activation could counteract akinesia in mice with acute and severe dopamine signaling deficiency.

**Prolonged activation of glutamatergic PPN neurons relieves motor suppression in mice with acute and severe dopamine signaling deficiency.** To induce a rapid and pronounced parkinsonian phenotype we interfered with dopaminergic transmission by acutely and systemically blocking dopamine signaling with the dopamine-receptor antagonists haloperidol or SCH23390. These pharmacological approaches partially model the biochemistry of PD and drug-injected mice show an overall reduction in motor output, with severe akinesia, slowness of movement (bradykinesia), and longer latency to initiate motor actions. This modeling approach—although it does not cause chronic degeneration of dopaminergic neurons—has shown predictive validity in the assessment of potential treatments for PD in humans[39].

Haloperidol is a preferential D2-type receptor antagonist[40] and systemic injection of haloperidol ("halo", 0.5 mg/kg[41], ip) induced akinesia peaking 25 min after injection and robustly maintained

for at least 1 h. Therefore, to test for locomotor recovery we adapted the Open Field session by dividing it into two periods. In the first period, mice were injected with the parkinsonian state-inducing drug (injection 1) and 5 min later monitored in the Open Field for 20 min. In the second period, mice were injected with CNO (injection 2) and recorded for 30 min to assess locomotor recovery. At the end of the session, mice were challenged in the Bar test to compare latency for motor initiation[42] (Fig. 2a, left). The experimental group design included a negative control group (Sham, CNO), an interference group (Vglut2_eD, CNO), and a no interference group (WT, saline) (Fig. 2a, right).

Sham mice injected with haloperidol followed by CNO remained akinetic, rarely performing locomotor bouts, and were mostly incapable of reaching speeds above 5 cm/s (Fig. 2b). When challenged in the Bar test Sham$^{halo+CNO}$ mice were unable to initiate motor actions and remained immobile while holding to the bar ($p = 0.0020$, Wilcoxon test vs median of 20 s). In contrast, Vglut2_eD$^{halo+CNO}$ mice exhibited a 3.6-fold increase in distance traveled (Fig. 2c and S1g), had normalized speed range (Fig. 2d), and initiated movement with low latencies to descent in the Bar test (Fig. 2e) (Distance traveled: Two-way RM ANOVA, group $F_{(1, 16)} = 7.895$ with $p = 0.0126$ and Bonferroni: [Halo only] $p > 0.9999$, [+CNO] $p = 0.0019$. Max difference from WT$^{sal+sal}$ in speed range used: 35% for Sham$^{halo+CNO}$ and only 5% for Vglut2_eD$^{halo+CNO}$, Two-way RM ANOVA, interaction $F_{(6, 60)} = 10.65$ with $p < 0.0001$, graph report shows Dunnett's vs WT$^{sal+sal}$. Bar test motor initiation with low latency: median

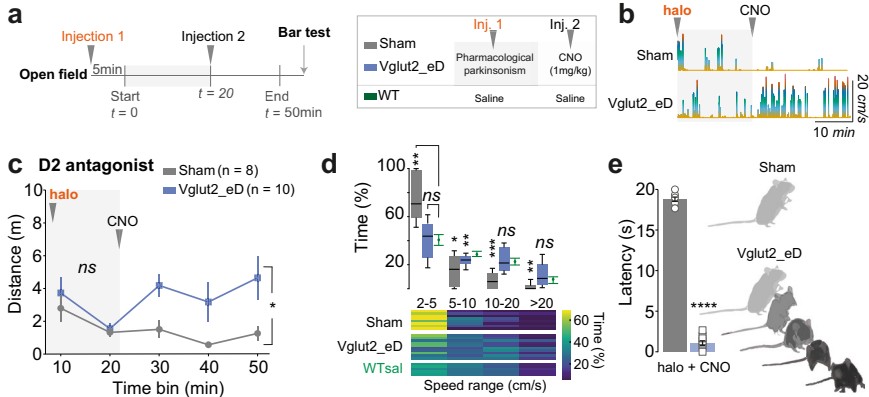

**Fig. 2 Chemogenetic activation of glutamatergic PPN neurons rescues motor suppression induced by D2 receptor antagonism. a** Chemogenetics experimental paradigm used to pharmacologically induce a parkinsonian motor phenotype (injection 1, light-gray background) and evaluate locomotor output after CNO (injection 2). Subsequently, mice were challenged with the Bar test to measure latency for motor initiation. Right panel shows the design of experimental groups. **b** Motion tracking with instantaneous speed profile in the Open Field (speed follows the divergent color scheme to facilitate reading). Mice injected with haloperidol ("halo") develop sustained immobility which can be counteracted by activation of caudal glutamatergic PPN neurons. **c** Timeline effect of D2 antagonist, haloperidol, before (light-gray background) and after injection of CNO in Sham (gray) and Vglut2_eD (blue) mice (Two-way RM ANOVA with graph reporting group effect and analysis done for each period separately). Data are presented as mean ± SEM. **d** Upper panel shows speed range in Sham (gray) and Vglu2_eD (blue) mice after CNO treatment in comparison to WT saline group (green). Min-to-max boxplots (Two-way RM ANOVA, Geisser–Greenhouse correction, report shows Dunnett's vs WT$^{sal+sal}$). Lower panel heat map with individual mice in horizontal lines and color code for percent of the time in each speed range. **e** Average motor initiation latency observed in the Bar test (20 s limit) for Sham (gray) and Vglut2_eD (blue) mice treated with haloperidol and then CNO. Individual values represent the average of three sequential trials/mouse (two-tailed, non-matched, Mann–Whitney test). The illustration shows motor response by Vglut2_eD$^{halo+CNO}$ mice as they descend from the bar and move away from the initial position, whereas Sham$^{halo+CNO}$ mice remain akinetic holding the half-reared start position. Data are presented as mean ± SEM. Data composed of five WT$^{sal+sal}$; eight Sham$^{halo+CNO}$; ten Vglut2_eD$^{halo+CNO}$ mice. See Fig.S1 and detailed stats in Table S1. Source data are provided as a Source Data file.

difference of 17.67 s between groups, $p < 0.0001$, two-tailed, Mann–Whitney).

Whereas both BG pathways are concomitantly active in freely moving animals[43] it is the activation of striatal D1 expressing MSNs that evokes locomotion[44,45]. Therefore, we investigated the effect of dampening the activity of the direct pathway by administering the D1-type receptor antagonist SCH23390[46] ("SCH") in the same cohort of mice. The dose was adjusted (0.25 mg/kg[47], ip) to induce a parkinsonian phenotype with a similar timeline as haloperidol facilitating interexperimental comparisons.

All mice injected with SCH showed akinesia that peaked within the first period of the test (Two-way RM ANOVA, group $F_{(1, 18)} = 3.003$ with $p = 0.1002$). CNO activation of Vglut2-PPN neurons increased distance moved by 2.35-fold (Fig. 3a and S1h) and enabled a broadened speed range distribution (Fig. 3b, c) (Distance traveled: Two-way RM ANOVA, group $F_{(1, 18)} = 10.93$ with $p = 0.0039$ and Bonferroni [SCH only] $p = 0.7033$, [+CNO] $p = 0.0002$. Speed range: bradykinesia seen as time within the 2–5 cm/s range equals 68% of total time for Sham$^{SCH+CNO}$ and drops to 25% of total time for Vglut2_eD$^{SCH+CNO}$, Two-way RM ANOVA, interaction $F_{(6, 66)} = 15.30$ with $p < 0.0001$. Bar test: median difference of 17 s; $p < 0.0001$, two-tailed, Mann–Whitney).

These experiments show that sustained activation of glutamatergic PPN neurons can alleviate motor suppression induced by acute dopamine depletion including when the pharmacological approach specifically silences the motor-promoting direct BG pathway.

**Dopamine depletion decreases population activity of both striatal D1-MSNs and glutamatergic PPN neurons.** The mouse models used here show a robust akinetic phenotype regardless of which of the two dopamine receptors are antagonized. To determine the extent to which this parkinsonian state is reflected within BG and PPN neuronal populations we examined the

activity of MSNs from the BG direct pathway and glutamatergic PPN neurons in freely behaving mice before and after drug-induced dopamine signaling deficiency.

To monitor cellular activity, we injected a conditional AAV virus expressing GCaMP6s into the dorsal striatum in Drd1$^{cre}$ mice (FK150 Gensat) or into the PPN of Vglut2$^{cre}$ mice (300 and 70 nl, respectively). Fluorescence dynamics were recorded using a miniaturized one-photon fluorescence microscope with parallel video monitoring of mouse behavior. An Open Field session was composed of a 12 min drug-free period followed by injection of a dopamine antagonist and, 5 min later, four calcium imaging periods with 3 min recording intervals. Two sessions were performed in each animal (with haloperidol [halo] or SCH23390 [SCH], same dosages as in the Open Field paradigm).

In the striatum (STR) neuronal calcium fluctuation was imaged from four mice (Fig. 4 and S2). During the drug-free period, with naturally occurring locomotor bouts we observed high calcium dynamics in D1-MSNs which then steadily declined after injection of haloperidol or SCH (Fig. 4c–f). Locomotor suppression was associated with silencing of 81% (128/158) of the D1-MSN neurons recorded during haloperidol administration and 78.9% (131/166) during SCH administration (Fig. 4g). The drug-induced decline in calcium dynamics were similar in all mice (Fig. S2d). For neurons identified in both experimental sessions (Fig. S2e, green bifurcated lines), and classified according to their drug response (Fig. S2f), follow up analysis revealed that in some instances neuronal response (decrease or increase activity upon drug challenge) could swop between days (Fig. S2g) (Response type: Two-Way ANOVA, interaction $F_{(3, 640)} = 285.8$ with $p < 0.0001$). These results show that a large proportion of the D1-MSN population becomes silent after an acute block of dopaminergic signaling.

In the PPN the drug-free period was characterized by active locomotion and variation of neuronal activity, which also declined dramatically when mice became akinetic after haloperidol or SCH drug injection (four mice) (Fig. 5 and S2). Of all

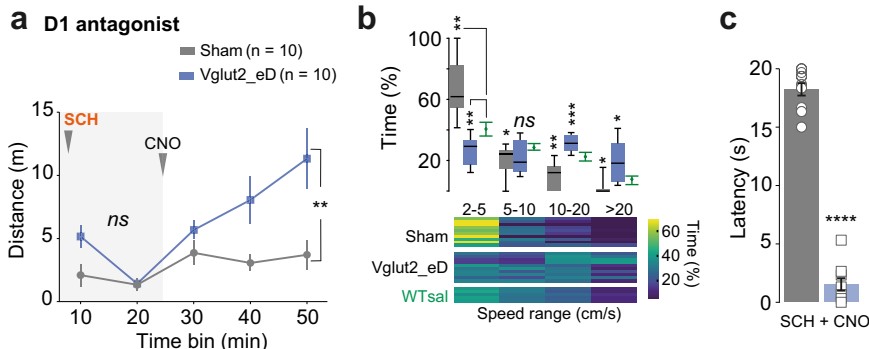

**Fig. 3 Chemogenetic activation of glutamatergic PPN neurons rescues motor suppression induced by D1-receptor antagonism. a** Timeline effect of D1 antagonist, SCH23390 ("SCH") before (light-gray background) and after injection of CNO in Sham (gray) and Vglut2_eD (blue) mice (Two-way RM ANOVA with graph reporting group effect and analysis done for each period separately). Data are presented as mean ± SEM. **b** Upper panel shows speed range in Sham (gray) and Vglut2_eD (blue) mice after CNO treatment in comparison to WT saline (green) group. Min-to-max boxplots (Two-way RM ANOVA, Geisser–Greenhouse correction, report shows Dunnett's vs WT$^{sal+sal}$). Lower panel shows a heat map with individual mice in horizontal lines and color code for the percentage of time in each speed range. **c** Average motor initiation latency observed in the Bar test (20 s limit) for Sham (gray) and Vglut2_eD (blue) mice treated with SCH and then CNO. Individual values represent the average of three sequential trials/mouse (two-tailed, non-matched, Mann–Whitney test). Data are presented as mean ± SEM. Data composed of five WT$^{sal+sal}$, same as Fig. 2; ten Sham$^{SCH+CNO}$; ten Vglut2_eD$^{SCH+CNO}$ mice. Experimental paradigm as in Fig. 2a. See also Fig. S1 and detailed stats in Table S1. Source data are provided as a Source Data file.

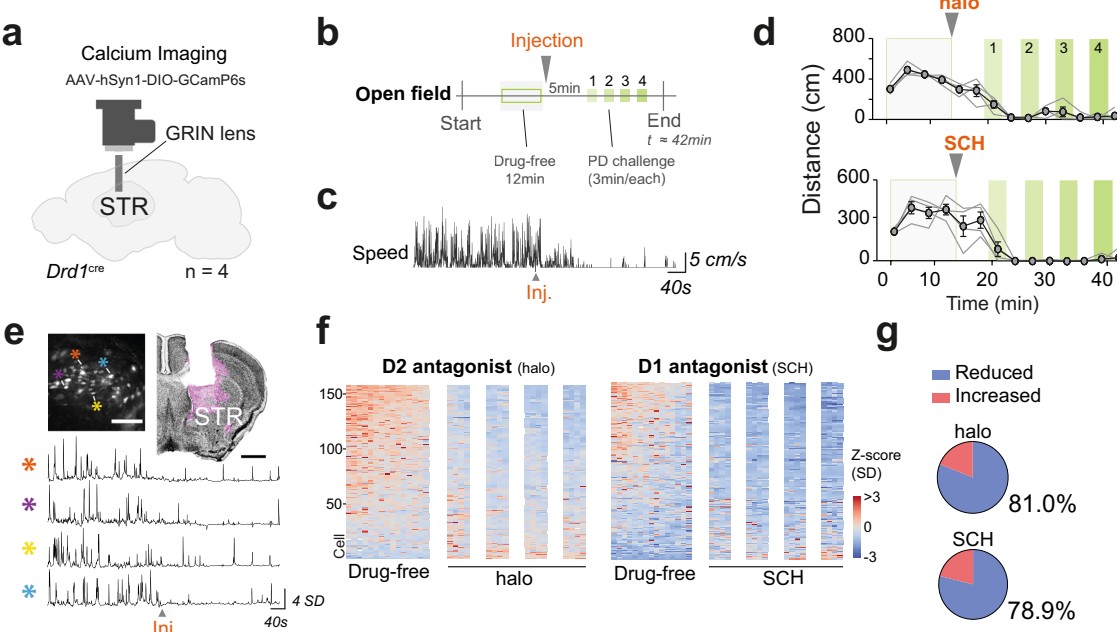

**Fig. 4 D1 and D2 receptor antagonists reduce the striatal activity of D1-MSNs. a** Miniaturized microscope positioned over the dorsal striatum (STR) of Drd1$^{cre}$ mice ($n = 4$) with local injection of AAV carrying Cre-dependent GCamP6s. **b** Experimental paradigm. **c** Example trace showing instantaneous speed profile of an animal injected with haloperidol. **d** Behavioral quantification of distance moved (cm) throughout the test. Individuals are represented by thin lines while group mean ± SEM are in black. Background colors corresponding to recording periods presented in **b**. Reduction of distance moved upon drug injection accounts for 87.46% of the total variance (Two-way RM ANOVA, time $F_{(13, 39)} = 47.42$ with $p < 0.0001$, both sessions analyzed in conjoint). **e** Left upper panel shows a field of view (FOV) with neuronal units expressing GCamP6s (scale 100 μm, image based on maximum deviation from average fluorescence, with neurons that have large differences in intensity over time becoming brighter in this projection type). The upper right shows a coronal slice with lens position above the dorsal striatum for the same animal (scale 1 mm, 0.38 mm from bregma), GCamP6s expression stained in magenta (NeuroTrace in black). The lower panel shows calcium dynamic traces from four cells identified in the FOV (*) (Z-score in units of SD). Experimental stages plotted as continuum and traces belong to haloperidol session speed profile shown in **c**. **f** Calcium dynamics of striatal Drd1 neurons before and after haloperidol (left, 158 cells) or SCH (right, 166 cells). Heat map bins of 50 s. Reduction in signal upon drug injection (Spearman's correlation: STR$^{halo}$ $r_s = -0.9582$, STR$^{SCH}$ $r_s = -0.9542$). **g** Pie charts show the fraction of all neurons that decreased (blue) or increased (red) their activity during PD state as compared to a drug-free period (for detailed analysis see Fig. S2d–g). Data are presented as Z-score in units of standard deviation (SD) from the baseline value, extracted from the 24 min recording. See also Fig. S2 and detailed stats in Table S2. Source data are provided as a Source Data file.

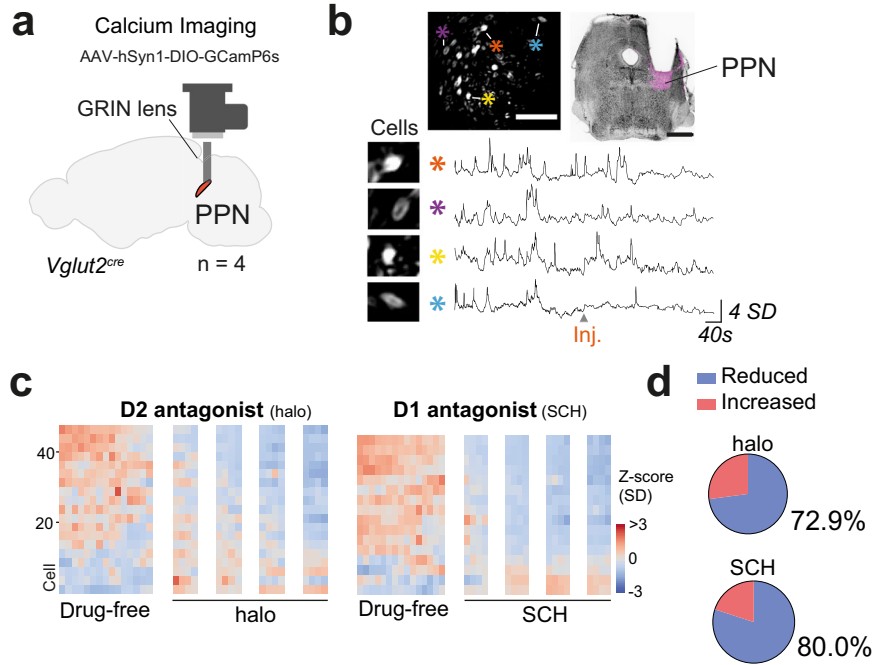

**Fig. 5 Glutamatergic PPN neuronal population shows an overall decrease in activity after dopamine-receptor antagonism. a** Microscope position and viral strategy were used to target the pedunculopontine nucleus (PPN) of Vglut2$^{cre}$ mice ($n = 4$) using AAV carrying Cre-dependent GCamP6s. **b** Left upper panel shows field of view (FOV) with neuronal units expressing GCamP6s (scale 100 μm, maximum deviation from average). The upper right shows a coronal slice with lens position above the PPN for the same animal, where GCamP6s expression is stained in magenta (scale 1 mm, −4.84 mm from bregma, NeuroTrace in black). The lower panel shows insets of neuronal shape extracted from four cells identified in the FOV (*) and their respective activity traces (Z-score in units of SD). **c** Calcium dynamics of Vglut2-PPN neurons before and after haloperidol (left, 48 cells) or SCH (right, 40 cells). Heat map bins of 50 s. Reduction in signal upon drug injection (Spearman's correlation: PPN$^{halo}$ $r_s = -0.8747$, PPN$^{SCH}$ $r_s = -0.9626$). **d** Pie charts show the fraction of all neurons that decreased (blue) or increased (red) their activity during PD state as compared to a drug-free period (for detailed analysis see Fig. S2d–g). Data are presented as Z-score in units of standard deviation (SD) from the baseline value, extracted from the 24 min recording. See also Fig. S2 and detailed stats in Table S2. Source data are provided as a Source Data file.

neurons recorded before and after haloperidol or SCH, 72.9% (35/48) and 80% (32/40), respectively, exhibited reduced activity during the parkinsonian state with the remaining neurons increasing their activity (Fig. 5d). Drug-induced silencing of the neuronal population occurred in all mice (Fig. S2d) and, similar to D1-MSNs, a few neurons within the PPN could swop response type depending on the drug used (Fig. S2e–g) (Response type: Two-Way ANOVA, interaction $F_{(3, 168)} = 169.9$ with $p < 0.0001$).

In conclusion, these data show that in these hypodopaminergic models most D1-MSNs and Vglut2-PPN neurons are silenced regardless of which dopamine-receptor is antagonized.

**Short-lasting optogenetic activation of glutamatergic PPN neurons opens a window for motor recovery.** To test whether time-locked activation of glutamatergic PPN neurons can revert the parkinsonian locomotor phenotype we injected a conditional AAV virus expressing channelrhodopsin-2 (AAVdj-DIO-ChR2-2A-mCherry, 80 nl) into the caudal PPN in Vglut2$^{cre}$ mice (Fig. 6a, upper panel). Mice were monitored for changes in locomotor activity in response to unilateral light activation of Vglut2_ChR2 neurons. The experimental session consisted of a 3 min habituation period followed by five laser-ON epochs (baseline trials 1–5), injection of saline, haloperidol, or SCH, followed by a 3 min habituation post-injection and 15 laser-ON epochs (trials 6–20). Finally, motor initiation was challenged using the Bar test (Fig. 6a, lower panel). Each laser-ON epoch consisted of trains of stimuli spaced by intertrial intervals (10 s total duration, 10 ms square pulses at 40 Hz[18,48], 473 nm blue light at 2–3.5 mW [measured from connector tip], intertrial interval 65–75 s). Analysis periods ("laser") were preceded

and followed by 10 s without stimulation ("pre" and "post", respectively).

In saline-injected Vglut2_ChR2 mice light-stimulation reliably evoked transient locomotor bouts resulting in a 2.23-fold increase in distance traveled during the laser-ON epochs (Fig. 6b and S3a) (fraction of trials where laser successfully evoked locomotion did not drop after saline injection: $p = 0.3632$, two-tailed, paired $t$-test. The locomotor increase during laser-ON: Two-way RM ANOVA with Geisser-Greenhouse correction, laser $F_{(1.266, 12.66)} = 55.29$ with $p < 0.0001$). Latency for initiation of locomotion ranged between $2.5 \pm 1$ s (SD) and was comparable with previously reported data[18]. In contrast, yellow light (593 nm, all other parameters kept equal), which is outside the effective wavelength for ChR2 activation did not affect locomotion (Fig. S3b) (Friedman test -RM, $T = 3$, $S = 10$ with $p = 0.8302$). To assess if virus transduction or light exerted nonspecific effects, we tested a control group containing mice injected with viruses that lacked the opsin component. In control mice no effect of light-stimulation was observed (Fig. S3c) (473 nm light at 3.5 mW [connector tip] all else kept equal, Two-way RM ANOVA, laser $F_{(1, 9)} = 0.6878$ with $p = 0.42$). Together, these experiments confirm that increased locomotor output is contingent on activation of caudal Vglut2-PPN neurons.

Having confirmed that optogenetic activation of Vglut2-PPN neurons reliably initiates locomotion we examined whether these effects were sufficient to drive motor output after haloperidol or SCH-induced motor suppression. During the drug-free baseline period, Vglut2_ChR2 mice responded to light (473 nm) in virtually all trials examined (Fig. S3d, e, baseline). Upon haloperidol injection mice became akinetic ("pre" and "post"),

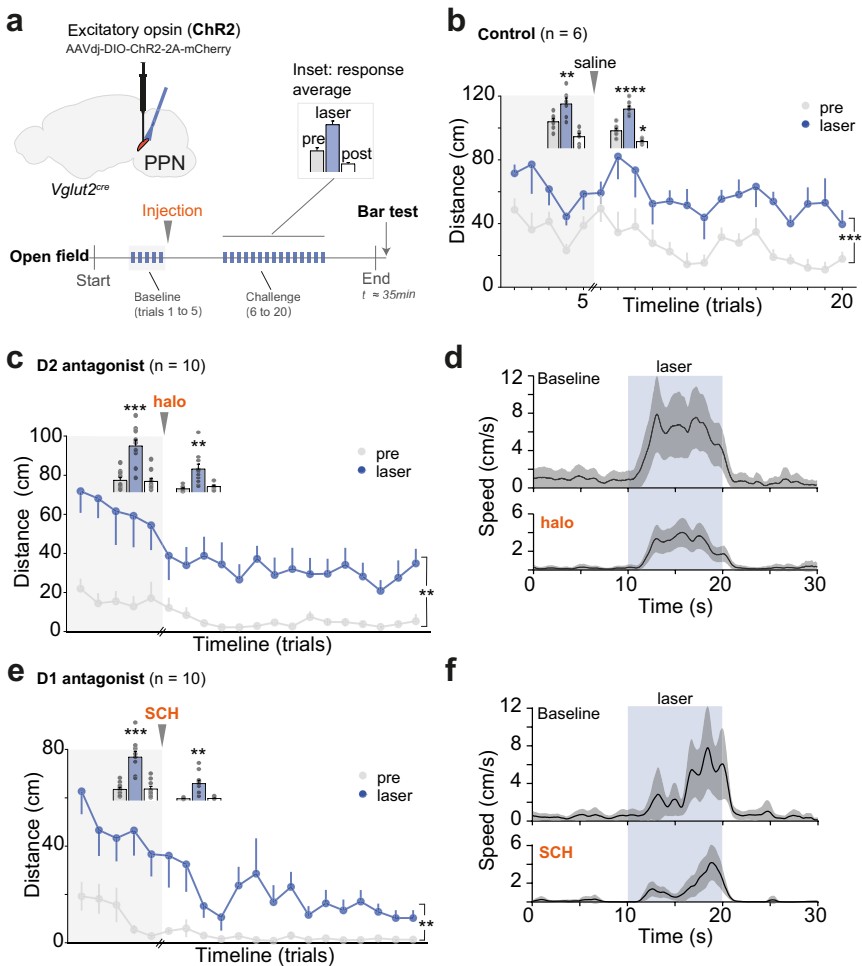

**Fig. 6 Optical excitation of glutamatergic PPN neurons acutely alleviates D1- and D2-mediated akinesia. a** Targeting strategy for unilateral optogenetic activation of caudal Vglut2-PPN neurons. The lower panel shows the experimental paradigm in the Open Field, where baseline and parkinsonian-challenge periods are composed of serial trials. For each test period, average group values during "pre" (before illumination), "laser" (during illumination), and "post" (after illumination) epochs (10 s each, interval between trials is 65–75 s) are presented as shown in the inset legend on the upper right. Data for inset comes from saline-injected mice in **b**. **b** Control experiment with saline injection in Vglut2_ChR2 mice. Line graph shows distance moved in each trial during "pre" and "laser" epochs throughout baseline (light-gray background) and after saline injection. Inset bars show group averages during each epoch ("pre", "laser", "post") for both test periods (5 and 15 trials, respectively). The same scale applies to both bar sets. Stats report at the end of the legend. **c** D2 antagonist, haloperidol ("halo") experiment. Line graph shows distance moved during "pre" and "laser" epochs, throughout baseline and parkinsonian-challenge trials. Inset bars show the average distance traveled per epoch. **d** For D2 antagonist experiment in **c**, instantaneous speed average of all trials during baseline (5 trials/mouse) and PD-challenge (15 trials/mouse). All trials included. **e** D1 antagonist, SCH23390 ("SCH") experiment. Graphed as in **c**. **f** For D1 antagonist experiment in **e**, instantaneous speed average of all mice and trials. Graphed as in **d**. Data are presented as mean ± SEM. **b**, **c**, **e** Line graphs: Two-way RM ANOVA (laser condition vs trial timeline, only events 6 to 20), the report shows the main effect of laser condition. Insets: Bar graph analysis is done with all six conditions as Two-way RM ANOVA, Geisser–Greenhouse correction, report shows Dunnett's multiple comparison to "pre". See also Figs. S3, S4, Movie 1, and detailed stats in Table S3. Source data are provided as a Source Data file.

but Vglut2-PPN activation promoted movement (6.5-fold increase in distance covered during laser-ON, "laser") albeit with longer latency (1.48 s mean increase) and a modest (-14 ± 5.48%, SEM) reduction in reliability of evoking locomotion by light stimulation when compared to baseline trials (Fig. 6c, d and S3d) (Motor recovery: Two-way RM ANOVA, laser $F_{(1, 9)} = 18.56$ with $p = 0.0020$. Latency: Wilcoxon matched-pairs, two-tailed, $p = 0.0488$, W = 39. Fraction of successful trials: $p = 0.0310$, two-tailed, paired $t$-test; $t = 2.553$, df = 9).

Likewise, activation of Vglut2-PPN neurons in SCH-induced parkinsonian mice promoted an 8.4-fold increase in distance traveled when compared with pre- and post-stimulation epochs (Fig. 6e, f) (Two-way RM ANOVA, laser $F_{(1, 9)} = 20.11$, $p = 0.0015$). When compared to baseline period (trials 1–5) we observed longer latency to locomotor initiation (1.4 s mean

increase) and reduction (-25 ± 7%, SEM) in the reliability of evoking locomotion (Fig. S3e) (Latency: Wilcoxon matched-pairs, two-tailed, $p = 0.0020$, W = 55. Fraction of successful trials: $p = 0.0081$, two-tailed, paired $t$-test; $t = 3.682$, df = 9). By comparing the distance moved in periods of no stimulation we confirmed that mice clearly developed an akinetic state in which, without the aid of the laser, motor output was constrained to 5–15% of baseline level (Fig. S4a) (Wilcoxon matched-pairs, one-tailed, saline $p = 0.3438$, halo and SCH $p = 0.0010$). Moreover, all drug-injected mice maintained their akinetic posture during the Bar test in the absence of stimulation (Fig. S4b and Movie 1) (is group median equal to the theoretical median of 20 s: halo: $p = 0.0005$ and SCH: $p = 0.0078$, one sample Wilcoxon test).

Postmortem analysis confirmed glutamatergic-ChR2 expression was higher in the caudal PPN with angled optical fiber

placement aimed at its most caudal edge. This fiber positioning together with the elongated anatomy of this region resulted in light scatter being limited to the most caudal part of the PPN (Fig. S3f–h).

Taken together, these results show that motor recovery is temporally locked to caudal Vglut2-PPN neuron activation and that mice regress to an akinetic state upon cessation of light stimuli.

**Locomotor recovery driven by glutamatergic PPN stimulation is independent of CnF and differs from direct CnF targeting.** Within the MLR both the PPN and the Cuneiform nucleus (CnF) contain Vglut2 positive neurons capable of promoting locomotion in healthy mice[18]. Yet, these two nuclei serve different behavioral functions. The PPN controls slow speed exploration, whereas CnF activity supports the entire range of speeds both slow gait and fast-paced locomotion[17,18,20,49]. A question that arises is if the locomotor promoting effect that we observe is mediated through Vglut2-CnF neurons. To evaluate this possibility, we performed a series of experiments.

The first series aimed at inhibiting Vglut2 neurons that reside within the CnF while concomitantly activating Vglut2 neurons in caudal PPN. For this, we injected the CnF in Vglut2[cre] mice with virus coding for the inhibitory Gi-coupled modified human M4 muscarinic receptor (hM4Dgi, inhibitory DREADD or "iD") (50 nl/hemisphere, bilateral) and the PPN with the conditional AAV virus expressing channelrhodopsin-2 (AAVdj-DIO-ChR2-2A-mCherry, 80 nl, unilateral) (Fig. 7a and S5a, b). This approach allows us to access if the PPN glutamatergic population alone can rescue mice from a parkinsonian phenotype.

To confirm appropriate inhibition of the CnF, mice were tested in an escape task in which the mouse runs a corridor to avoid an air puff[18] (Corridor test; injected with saline or CNO, 20 min prior start). Inhibition of glutamatergic CnF neurons reduced the average speed during the escape and limited the max acceleration that mice could perform suggesting that Vglut2-CnF functionality was suppressed by the CNO injection (Fig. 7b, c) (Reduction in average escape speed from $58 \pm 11$ to $39 \pm 7$ cm/s[SD] and max acceleration drops from $23 \pm 5.7$ to $11 \pm 2.9$ m/s$^2$ [SD], both analyzed with two-tailed, paired $t$-test, $p < 0.0001$). Next, we tested the effect of CNO in the Open Field paradigm. Here, the same group of mice were recorded during 50 min sessions (injected with saline or CNO, 20 min prior). We observed only a mild but significant effect on the cumulative distance traveled (Fig. S5c) (Total distance: $p = 0.0398$; two-tailed, paired $t$-test, $t = 2.402$, df $= 9$, with a median reduction of 0.24-fold. Distance over time: Two-way RM ANOVA with Geisser–Greenhouse correction, interaction $F_{(9, 162)} = 2.291$ with $p = 0.0190$, representing only 2.963% of the total variation. Whereas time alone represents 43.75% of the total variation, time $F_{(2.793, 49.31)} = 33.83$, $p < 0.0001$. CnF inhibition reduced the time spent locomoting with $-6 \pm 2.7\%$ (Fig. S5d) (equivalent to ~3/50 min recording) without affecting the speed ranges used during Open Field exploration (Fig. S5e) (Time locomoting: $p = 0.0389$, two-tailed, paired $t$-test, $t = 2.228$, df $= 18$. Speed range preference: $p = 0.0009$, $F_{(3, 54)} = 6.333$ corresponding to only 0.87% of the total variation, Two-way RM ANOVA). These results indicate that Vglut2-CnF neurons, responsible for the expression of high-speed locomotion, are inactivated by the iD (Corridor test) but that exploratory locomotion remains intact (Open Field).

Having confirmed that CnF could be inhibited by the chemogenetic approach we went on to stimulate the Vglut2-PPN neurons in the absence of Vglut2-CnF neuron engagement. We used the same stimulation protocol as previously applied (see previous; Fig. 6a) with an extended window of 20 min between

the drug-free trials (trials 1–5) and challenge trials (6–20) allowing CNO to reach full effect during the second period of the test. Optogenetic activation of glutamatergic PPN neurons promoted locomotion regardless of Vglut2-CnF engagement (Fig. 7d, e and S5f, g) (Locomotor increase during laser-ON: Two-way RM ANOVA, laser $F_{(1,9)} = 35.74$ with $p = 0.0020$. Fraction of trials where light successfully evoked locomotion did not drop after injection$^{sal+CNO}$: $p = 0.7025$, two-tailed, paired $t$-test. Latency for initiation of locomotion ranged between $3.31 \pm 0.81$ s [SD]). For comparison, saline-injected PPN activated Vglut2_ChR2 mice showed a 2.23-fold increase in distance traveled during laser-ON (Fig. 6b), whereas with CnF inactivation (Fig. 7d) we observed a 2.75-fold increase (comparing distance increase between experiments: $p = 0.5624$, two-tailed, unpaired $t$-test, $t = 0.593$, df $= 14$).

When mice were made parkinsonian by injection of haloperidol or SCH23390 activation of the PPN *alone* was sufficient to promote locomotion with values that were nearly indistinguishable from Vglut2_ChR2 mice in which no concomitant CnF inhibition was applied (Fig. 7f–i, S5f, g vs Fig. 6, S3) (Motor recovery: Two-way RM ANOVA with Geisser–Greenhouse correction, halo: laser $F_{(1.587, 26.56)} = 70.85$ with $p < 0.0001$ and SCH: laser $F_{(1.354, 24.37)} = 40.94$ with $p < 0.0001$. Laser-ON increased distance covered by 15- and 11-fold, respectively, with proportion of increase statistically equal between groups in Figs. 6 and 7, halo: $p = 0.07$, $t = 1.926$, df $= 18$ and SCH: $p = 0.034$, $t = 2.295$, df $= 18$, two-tailed, unpaired $t$-test. Latency to locomote: Wilcoxon matched-pairs, two-tailed, halo: mean increase of 0.82 s, $p = 0.0020$, W $= 55$ and SCH: mean increase of 1.47 s, $p = 0.0371$, W $= 41$. Here mice locomoted in nearly every trial regardless of drug, fraction of successful trials in halo: $p = 0.9059$, $t = 0.1216$ and SCH: $p = 0.1456$, $t = 1.593$, two-tailed, paired $t$-test, df $= 9$).

Drug-induced akinesia was confirmed by evaluating periods of no stimulation (Fig. S5h) (Locomotor output constrained to 5–8% of baseline level. Wilcoxon matched-pairs, one-tailed, halo and SCH $p = 0.0010$). In the Bar test, we included a measure of latency to descent prior to and during laser-ON epochs. PPN activation facilitated the initiation of a motor response with all mice descending from the bar in less than 6 s (Fig. S5i) (Latency: 4.4 s [halo+laser] and 3.9 s [SCH + laser], Wilcoxon matched-pairs, one-tailed, halo and SCH with $p = 0.0010$). Finally, no effect was observed when the laser wavelength was outside the range needed for ChR2 activation (Fig. S5j) (493 nm, CNO injected 20 min prior. Friedman test -RM, T $= 3$, S $= 10$ with $p = 0.7422$).

Our observations indicate that activation of the PPN is sufficient for the rescue of parkinsonian akinetic phenotype and that the motor recovery is independent of CnF activity.

In the second series of experiments and as a further confirmation that activation of Vglut2 PPN neurons *alone* is sufficient to promote locomotion in parkinsonian mice we evaluated if chemogenetic activation of glutamatergic CnF neurons was equivalent to Vglut2-PPN specific targeting. In a new group of Vglut2[cre] mice, we injected eD (50 nl/hemisphere, bilateral) in the CnF (Fig. S6a, b). CNO-mediated activation of CnF glutamatergic neurons caused a 251% increase in distance moved as compared to the saline-injected session (Fig. S6c) while PPN targeting increased distance by 90% with same experimental protocol (Total distance (m): mean $\pm$ SEM of saline-injected [$188.7 \pm 16.4$], CNO injected [$474.9 \pm 63.7$]: $p = 0.0084$; two-tailed, paired $t$-test, $t = 4.884$, df $= 4$. Distance over time: Two-way RM ANOVA, interaction $F_{(9, 72)} = 8.750$, $p < 0.0001$, again CNO effect becomes significant 20 min after injection). Time spent locomoting was, however, similar in the two conditions (Fig. S6d) although CnF activation was associated

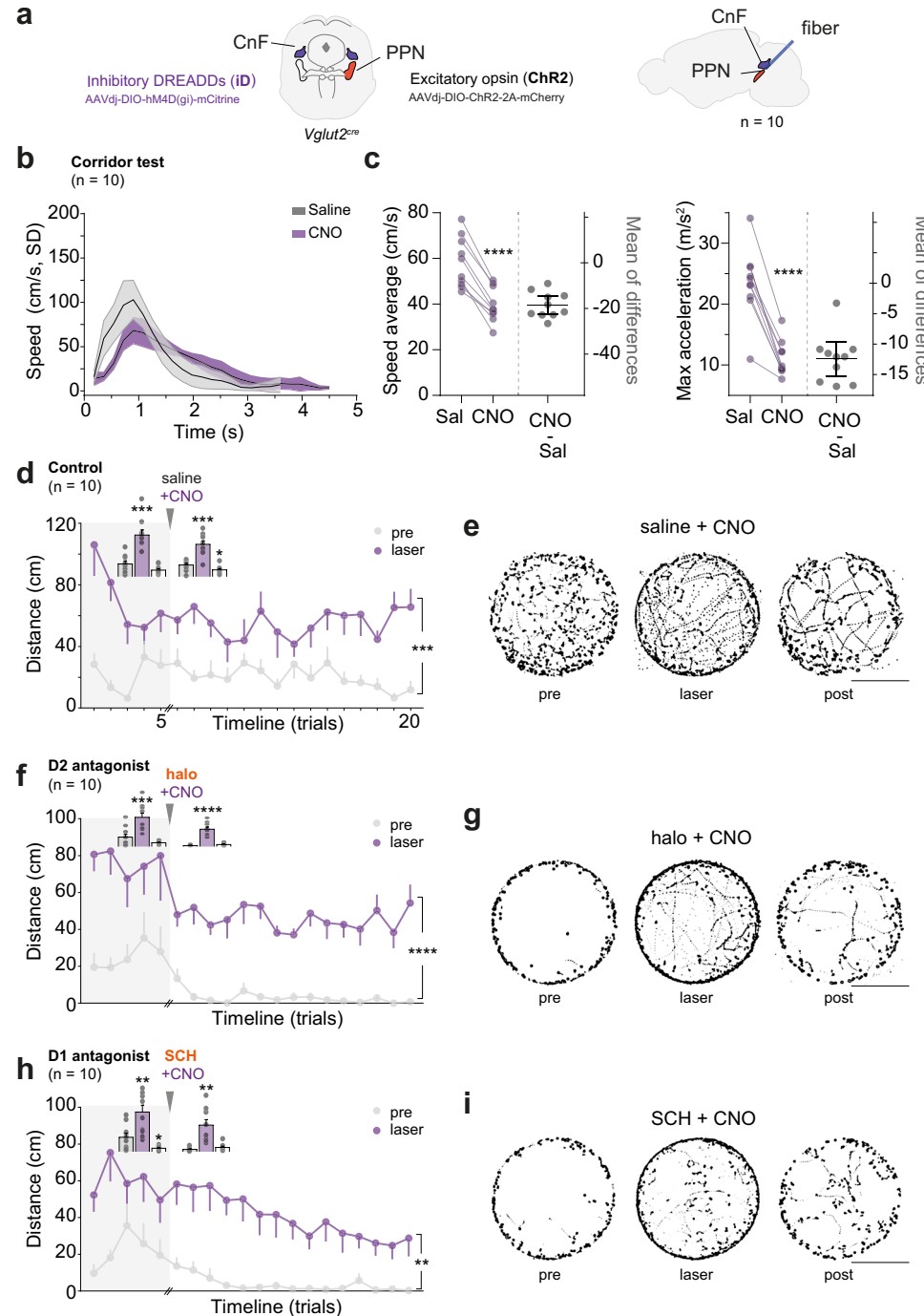

with higher velocities when moving (Fig. S6e) (Time locomoting: $p = 0.9463$, two-tailed, paired $t$-test, $t = 0.06943$, df = 8. Speed ranges: Two-way RM ANOVA, speed $F_{(3, 24)} = 39.79$ with $p < 0.0001$, where speeds >20 cm/s correspond to 30.52% of the locomotor time in CNO condition and only 12.62% if mice were given saline). The high speed caused by Vglut2-CnF activation with chemogenetics (up to 65 cm/s) was not seen with Vglut2-PPN activation (max reached 42 cm/s) or observed in Control-Sham[CNO] or naïve WT[sal] mice (Fig. S6f) (Speed average when above 20 cm/s: for PPN targeted mice was $34.70 \pm 2.7$, whereas for CnF it was $55.67 \pm 9.2$ cm/s [SD], all other tested groups remained between 23–26 cm/s. CnF-targeted mice showed significantly higher speeds than PPN targeted mice: $p = 0.0003$, one-tailed, Mann–Whitney). Another difference to Vglut2-PPN

chemogenetic stimulation was that Vglut2-CnF mice performed long stop bouts (67.2% longer than saline-injected condition: two-tailed, paired $t$-test with $p = 0.0270$, $t = 3.410$, df = 4) (Fig. S6g) followed by abrupt changes to high-speed locomotion. This locomotor pattern resembles "darting" a behavior performed as a defense mechanism to avoid detection by predators[50]. Darting can effectively reduce the cumulative time needed to cover longer distances. A reliable approach to quantify darting in rodents comes from measuring, for each locomotor bout, the ratio between its Maximal Speed divided by its Duration, the so-called MSD ratio ([cm/s/s], see[50] for its origin). In WT rodents this ratio is around 25 cm/s/s[50]. The average MSD ratio of CNO injected Vglut2-CnF mice was $77 \pm 10.6$ cm/s/s [SD] and darting behavior occurred in nearly every locomotor bout (see individual

**Fig. 7 Rescue of parkinsonian locomotor symptoms by optical excitation of glutamatergic PPN neurons is independent of Cuneiform nucleus (CnF) engagement. a** Bilateral chemogenetic silencing of CnF neurons in Vglut2[cre] mice with unilateral optogenetic activation of PPN neurons. Right panel shows a sagittal view of the mouse brain. **b** Corridor test. Mice ($n = 10$) were injected with either saline (gray) or CNO (purple) and their speed (cm/s) was measured as they crossed a corridor to escape an air puff. CnF inhibition prolonged escape time (area under the curve [AUC], difference between means $= -6.740 \pm 2.441$ SEM, two-tailed, paired $t$-test, $p = 0.0221$, $t = 2.762$, df $= 9$). Graph shows mean $\pm$ SD, all trials included in the analysis. **c** Left panel: individual average crossing speed in the Corridor test, under saline or CNO condition (before-after purple lines). In gray, difference between pairs (one circle per mouse) with overall decrease of 18.48 cm/s (mean $\pm$ CI in black, left axis. Two-tailed, paired $t$-test, $p < 0.0001$, $t = 10.30$, df $= 9$). Right panel: maximal acceleration reached by each subject with mean reduction of 12.44 m/s$^2$ (mean $\pm$ CI in black, left axis. Two-tailed, paired $t$-test, $p < 0.0001$, $t = 9.778$, df $= 9$). **d** Vglut2-PPN activation with concomitant Vglut2-CnF inhibition. Open Field control experiment with Saline+CNO injection ($n = 10$ mice). Distance moved in each trial during "pre" and "laser" epochs throughout baseline (light-gray background) and after injections. Inset bars show group averages during each epoch ("pre", "laser", "post") for both test periods. The same scale applies to both bar sets (stats report at the end of legend). **e** Open Field location map after Saline + CNO injection (from data in **d**) for "pre", "laser", and "post" epochs in all trials (15 trials/mouse; all mice included). Scale 25 cm. **f, g** D2 antagonist, haloperidol ("halo") experiment ($n = 10$). Same configuration as in **d, e. h, i** D1 antagonist, SCH23390 ("SCH") experiment ($n = 10$). Same configuration as in **d-e**. Statistics **d, f, h** Data presented as mean $\pm$ SEM. Line graphs: Two-way RM ANOVA (laser condition vs trial timeline, only events 6 to 20), the report shows the main effect of laser condition. Insets: Bar graph analysis is done with all six conditions as Two-way RM ANOVA, Geisser–Greenhouse correction, report shows Dunnett's multiple comparison to "pre". See also Fig. S5 and detailed stats in Table S3. Source data are provided as a Source Data file.

CI in Fig. S6h). For comparison, the PPN average MSD ratio was very low, at $22 \pm 1.3$ cm/s/s whereas saline-injected mice scored at $20 \pm 0.4$ (CnF-targeted mice MSD ratio: Kruskal–Wallis, saline vs CNO; CnF $p = 0.0083$, PPN $p > 0.9999$). Thus, CnF-targeted mice had ratios well above the average of WT and Vglut2-PPN mice. Darting behavior was also seen after CnF stimulation in mice made akinetic by haloperidol injection, with a mean MDS ratio of $51 \pm 11$ [SD]. Their increased distance traveled (5-fold increase, PPN targeting showed a 3.6-fold increase) was associated with unexpectedly high speeds with few locomotor bouts interspaced by long akinetic periods (Fig. S6i–o) (Distance traveled: Two-way RM ANOVA, group $F_{(1, 11)} = 10.23$ with $p = 0.0085$ and Bonferroni: [Halo only] $p > 0.9999$, [+CNO] $p = 0.0006$. Speed range use, compared to WT$^{sal+sal}$: CnF-targeted mice increased by $+20.76\%$ their preference for speeds $>20$ cm/s, Two-way RM ANOVA, interaction $F_{(3, 24)} = 16.83$ with $p < 0.0001$, graph report shows Sidak's vs WT$^{sal+sal}$. MSD ratio kept being significantly higher $+2.55$-fold above saline condition, one-tailed, paired $t$-test with $p = 0.0256$, $t = 2.754$, df $= 4$. Bar test without clear recovery: median difference of 6.7 s [before vs after CNO], $p = 0.0625$, two-tailed, Wilcoxon matched-pairs).

In conclusion, the chemogenetic activation of CnF glutamatergic neurons caused a locomotor phenotype characterized by darting that is distinctly different from the Vglut2-PPN-induced locomotor phenotype, where we observed locomotor normalization. Altogether, the data show that Vglut2-PPN-driven recovery of locomotion is not associated with Vglut2-CnF neuron activation.

**Prolonged activation of GABAergic PPN neurons drives brief stop bouts followed by locomotion.** Glutamatergic PPN neurons are intermingled with GABAergic and cholinergic neuronal populations[51]. In rodents, cholinergic neurons represent ~1/3 of all neurons in the PPN[52,53] and their selective short-lasting stimulation has limited effects on locomotion in healthy mice[18,19,54]. GABAergic neurons are abundant in the rostral part and form a high-density cluster in the most caudal PPN[52,55]. They outnumber cholinergic neurons[53] and have been shown to decrease locomotor speed when shortly stimulated with optogenetics[18,19]. The locomotor effect of prolonged GABAergic PPN excitation has not been tested. We wanted to compare the effects of activating glutamatergic or GABAergic PPN sub-populations in healthy and parkinsonian mice.

Therefore, we injected excitatory DREADDs in the caudal PPN of Vgat[cre] (Vesicular GABA transporter)[56] mice (Vgat_eD, 80 nl/ hemisphere) (Fig. 8a). CNO increased c-Fos expression in Vgat-PPN neurons, indicative of induced neuronal activity (Fig. S1c–f)

(Colocalized: $84.76 \pm 5.723\%$ [SD], two-tailed, paired $t$-test $p < 0.0001$, $t = 16.07$, df $= 4$). To allow comparisons with PPN Vglut2_eD mice, we performed the Open Field experiments following the same protocol reported in Fig. 1.

Somehow unexpectedly, prolonged activation of GABAergic PPN neurons increased the total distance traveled by 68% above that of the Sham$^{CNO}$ ($p < 0.0001$, two-tailed, $t$-test, $t = 5.442$, df $= 24$), although less pronounced than the Vglut2_eD$^{CNO}$ mice (which shows an increase of 90%) (Fig. 8b) (Total distance (m): mean $\pm$ SEM of Sham [$129.2 \pm 9.8$], Vgat_eD [$217.5 \pm 13.5$], Vglut2_eD [$245.7 \pm 21.3$]). The increased distance traveled by Vgat_eD$^{CNO}$ mice was achieved mainly by locomoting for a longer period of time (23% increased locomotor time, whereas Vglut2_eD$^{CNO}$ mice increased locomotor time by 11% above Sham$^{CNO}$) but unlike Vglut2_eD$^{CNO}$ mice, the speed range preference peaked at 5–10 cm/s (Fig. 8c, d) (Locomotor time: $p < 0.0001$, two-tailed, $t$-test, $t = 7.458$, df $= 24$. Speed ranges: Two-way RM ANOVA, interaction $F_{(3, 72)} = 9.713$ with $p < 0.0001$ and range distribution very similar to Sham$^{CNO}$, max difference between Vgat_eD and Sham $<5.7\%$).

Thus, Sham, Vgat_eD, and Vglut2_eD mice can be distinguished by comparing distance traveled vs speed profile in an Open Field test (CNO, 5 min prior) suggesting differences in the episodic expression of locomotion in the three groups. The total distance traveled is composed of locomotor bouts (start and stop of locomotion) interspersed with periods of stops bouts (periods where the mouse paused locomotion and remained still). Activation of GABAergic PPN neurons induced a 42% increase in stop count, whereas activation of glutamatergic PPN neurons decreased the number of stop bouts (25% less) as compared to Sham mice. Yet, the average stop bout duration of Vgat_eD mice was nearly half of that observed in Sham and Vglut2_eD mice (Fig. 8e, f) (Stop frequency: Brown–Forsythe and Welch ANOVA, B–F $= 80.20_{(2, 31.69)}$ with $p < 0.0001$; W $= 102.4_{(2, 21.20)}$ with $p < 0.0001$. Stop bout duration: B–F $= 21.16_{(2, 20.83)}$ with $p < 0.0001$; W $= 37.03_{(2, 17.33)}$ with $p < 0.0001$, see detailed stop analysis definition in methods).

These results show that prolonged activation of GABAergic PPN in healthy mice can drive locomotor output with an episodic pattern. Locomotion becomes interrupted by a higher number of brief stop bouts, which has also been described in the literature[18,19]. The stops of Vgat_eD$^{CNO}$ mice are not followed by continued immobility and mice quickly transition back to locomotion with a preferred speed range between 5–10 cm/s. The result is the increased distance traveled by Vgat_eD$^{CNO}$ mice as compared to Sham, despite the use of slow locomotor speed.

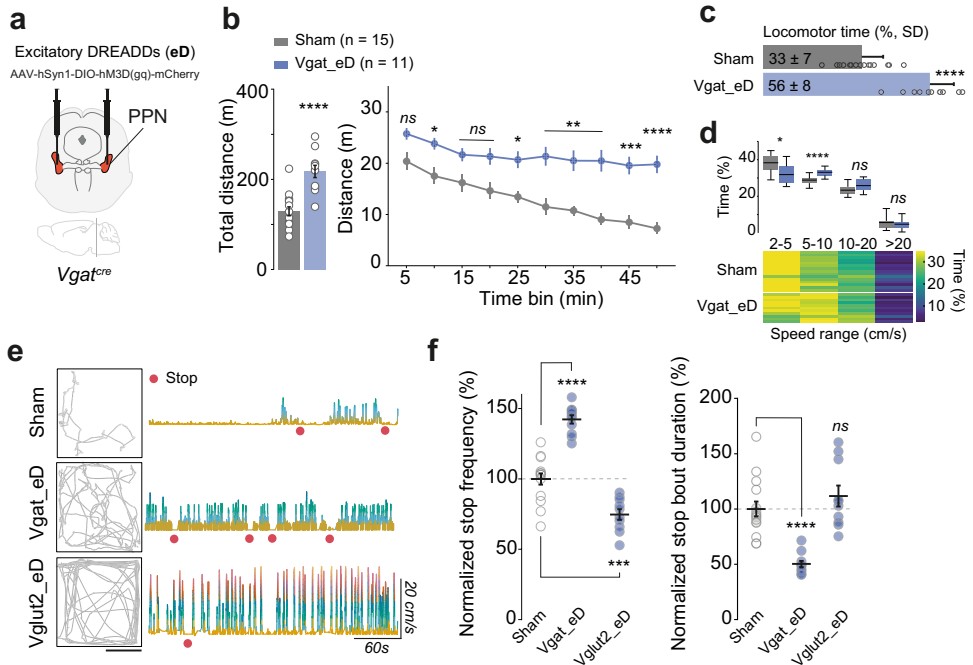

**Fig. 8 Bilateral chemogenetic activation of GABAergic PPN neurons promotes locomotor activity which is flanked by numerous brief stop bouts. a** Viral strategy to bilaterally express excitatory DREADDs (eD) in GABAergic neurons of the caudal PPN. **b** Total distance moved by Sham (gray) and Vgat_eD (blue) mice after CNO (bar graph) (two-tailed, $t$-test). Timeline effect of CNO (line graph) (Two-way RM ANOVA, Geisser–Greenhouse correction followed by Bonferroni's multiple comparison, the report shows post hoc results). Data are presented as mean ± SEM. **c** Percentage of locomotor time (speed above >2 cm/s) during Open Field test (two-tailed, $t$-test). Data are presented as mean ± SD. **d** Upper panel, speed range in Sham (gray) and Vgat_eD (blue) mice with min-to-max boxplots (Two-way RM ANOVA, Geisser–Greenhouse correction, with report for Bonferroni's multiple comparison). Lower panel, heat map with individual mice in horizontal lines, and color code for the percentage of time in each speed range. **e** Representative tracks (minute 45 to 50, scale 25 cm) of each experimental group (Sham, Vgat_eD, Vglut2_eD) in the Open Field after CNO injection. Right panel shows the instantaneous speed profile for the three tracks (speed follows the divergent color scheme to facilitate reading). CNO-mediated activation of Vgat- or Vglut2 PPN neurons resulted in significant increases in the distance traveled but with different locomotor periodicity. Red dots indicate stop events in absence of in-place behavior, as identified by the tracking software (see methods for detailed definition). **f** Stop events counted across the 50 min Open Field test with all groups having received CNO treatment 5 min prior to start. Data are normalized to Sham average (dashed line). Individual data points (circles) and group mean ± SEM (black lines). Left panel, stop frequency. Right panel, average stop bout duration (Brown–Forsythe and Welch ANOVA, report shows Dunnett's multiple comparison (T3) to Sham group). Vgat_eD$^{CNO}$ mice show a higher number of brief stop bouts. Data composed of 15 Sham$^{CNO}$; 10 Vglut2_eD$^{CNO}$ [same cohort as in Fig. 1], and 11 Vgat_eD$^{CNO}$ mice [these Open Field experiments were performed in parallel]. See detailed stats in Table S5. Source data are provided as a Source Data file.

**Activation of GABAergic PPN neurons in parkinsonian mice offers partial support to motor recovery.** To examine whether stimulation of GABAergic neurons could contribute to the amelioration of drug-induced akinesia we tested the effect of prolonged activation of Vgat-PPN neurons in the two mouse models. After haloperidol injection, Sham and Vgat_eD mice showed equal levels of akinesia and bradykinesia. Upon CNO treatment we observed a 3.66-fold increase in distance traveled by Vgat_eD$^{halo+CNO}$ mice compared to Sham$^{halo+CNO}$ mice (Fig. 9a, b and S7a) (halo-induced akinesia: Two-way RM ANOVA, group $F_{(1, 16)} = 1.723$ with $p = 0.2079$ and bradykinesia: Two-way RM ANOVA, interaction $F_{(3, 48)} = 0.9591$ with $p = 0.4197$. CNO reduces akinesia: Two-way RM ANOVA, Bonferroni's [halo] $t = 1.087$, $p = 0.5701$ [+CNO], $t = 5.730$, $p < 0.0001$, df = 32). However, both Sham and Vgat_eD mice continued being bradykinetic with a 75% preference for the slowest speed range (Fig. 9b) (Two-way RM ANOVA, interaction $F_{(6, 60)} = 10.04$ with $p < 0.0001$). When challenged with the Bar test motor initiation latency was reduced in CNO treated Vgat_eD mice (Fig. 9c) ($p < 0.0001$, two-tailed, non-matched, Mann–Whitney test). Thus, locomotor output was facilitated but the bradykinesia remained.

In contrast to the halo-induced akinetic state, in SCH-parkinsonian mice CNO-mediated activation of GABAergic

neurons did not facilitate locomotion, with mice remaining both akinetic and bradykinetic (Fig. 9d, e and S7b) (No recovery on distance traveled: Two-way RM ANOVA group $F_{(1, 18)} = 0.05971$ with $p = 0.8097$. Speed restricted to 2–5 cm/s: Mean, Sham = 67.96%, Vgat_eD = 85.33% of locomotor time, whereas WT$^{sal+sal}$ = 40.86%). Yet, CNO treatment of SCH-Vgat_eD mice facilitated motor initiation in the Bar test (Fig. 9f) ($p < 0.0001$, two-tailed, non-matched, Mann–Whitney). These results show that the phenotypic recovery profile generated by GABAergic PPN chemogenetic activation depends on which dopamine receptor type is antagonized.

The data obtained by chemogenetic activation of GABAergic PPN neurons was mirrored by the results acquired with an optogenetic approach (Figs. S8, S9, Movie 2, and Table S6).

In saline control experiments, long-lasting light stimulation (473 nm, 10 s at 40 Hz, as in Vglut2_ChR2 experiments) led to a modest (15–27%) increase in distance traveled (Fig. S8a), which was accompanied by a 15% higher frequency of stop bouts during laser-ON ("laser") as compared to "pre" values (Fig. S8b) (Locomotor increase during laser-ON: Two-way RM ANOVA with Geisser–Greenhouse correction, laser $F_{(1.903, 34.25)} = 10.96$ with $p = 0.0003$. Stop bout frequency: $p < 0.0001$, two-tailed, paired $t$-test, $t = 7.889$, df = 9).

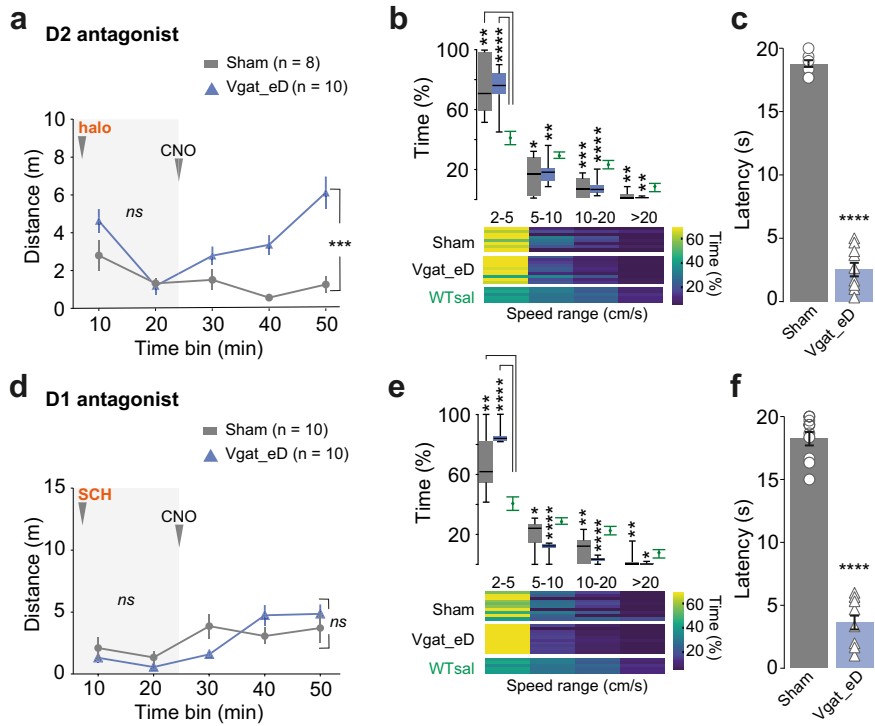

**Fig. 9 Bilateral chemogenetic activation of GABAergic PPN neurons alleviates akinesia induced by D2 but not D1-receptor antagonism. a** Timeline effect of haloperidol ("halo"), before (light-gray background) and after injection of CNO in Sham and Vgat_eD mice (mean ± SEM). Stats report at the end of the legend. **b** Upper panel, speed range in Sham (gray) and Vgat_eD (blue) mice after CNO treatment in comparison to WT saline group (green) with min-to-max boxplots. Lower panel shows a heat map with individual mice in horizontal lines and color code for the percentage of time in each speed range. **c** Average motor initiation latency observed in the Bar test (20 s limit) for Vgat_eD and Sham mice treated with haloperidol and then CNO (symbols represent the average of each mouse, group mean ± SEM). **d** Timeline effect of SCH23390 ("SCH"), before and after injection of CNO in Sham and Vgat_eD mice. Graphed as in **a**. **e** Upper panel, speed range in Sham (gray) and Vgat_eD (blue) mice after CNO treatment in comparison to WT saline group (green) with min-to-max boxplots. Lower panels show heat maps for the percentage of time in each speed range. Graphed as in **b**. **f** Average motor initiation latency obtained in the Bar test for mice treated with SCH and then CNO. Graphed as in **c**. Data composed of eight Sham[halo+CNO]; ten Sham[SCH+CNO]; ten Vgat_eD[halo+CNO]; ten Vgat_eD[SCH+CNO] mice. Experimental paradigm as in Fig. 2a. **a**, **d** Two-way RM ANOVA with graph report for group effect, analysis is done for each period in separate. **b**, **e** Two-way RM ANOVA, Geisser–Greenhouse correction, the report shows Dunnett multiple comparison vs WT[sal+sal] group. WT is the same as in Figs. 2, 3. **c**, **f** Individual values represent the average of three sequential trials/mouse. Two-tailed, non-matched, Mann–Whitney test. See also Fig. S7 and detailed stats in Table S5. Source data are provided as a Source Data file.

In haloperidol parkinsonian mice, light stimulation caused a 2-fold increase in overall locomotor output but reached only 50% of the baseline level (Fig. S8c) (Two-way RM ANOVA, laser $F_{(1, 7)} = 15.93$ with $p = 0.0052$). The low distance covered was explained by longer latency to initiate locomotion and lower reliability of evoking movement by optical stimulation compared to baseline (Fig. S8d) (Latency: $p = 0.0020$, Wilcoxon matched-pairs signed-rank test, two-tailed, W = 55. Fraction of successful trials decreases by 30%: $p = 0.0025$, two-tailed, paired $t$-test, $t = 4.608$, df = 7).

In SCH-parkinsonian mice, light stimulation had no effect on distance moved (Fig. S8e, f) (Two-way RM ANOVA, laser $F_{(1, 8)} = 4.765$ with $p = 0.0606$) despite retesting with prolonged light stimulus protocol (20 s, laser-ON: One-way RM ANOVA with Geisser–Greenhouse correction, $F_{(1.857, 14.86)} = 7.174$ with $p = 0.0074$, where motor recovery reached only 11% of baseline). Last, when this group of mice was tested with yellow light (593 nm, all other parameters kept equal), which is outside the effective wavelength for ChR2 activation, no locomotor effect was observed (Fig. S9a) (Friedman test -RM, T = 3, S = 10 with $p = 0.8302$) (for anatomical and akinetic state variables see Fig. S9b–e).

In agreement with chemogenetic data when challenged on the Bar test Vgat_ChR2 mice injected with either halo or SCH could initiate a motor response and descend from the bar if aided by optical stimulation (Movie 2).

These results show that prolonged optogenetic activation of GABAergic PPN neurons in healthy mice can promote movement, in contrast to what was observed with short-lasting (2 s) light activation[18,19]. These apparent differences in reported data are potentially due to the long latency to initiate locomotion and the increase in brief stop bouts (as reported here). When mice are akinetic after antagonizing D2 receptors (halo) but not D1 receptors (SCH) activation of GABAergic PPN neurons facilitated locomotion, but mice remained bradykinetic. Yet, a clear improvement in motor performance was observed in the Bar test for all conditions tested.

**Motor recovery driven by activation of glutamatergic PPN neurons is proficient and adaptable**. Our analysis of locomotor recovery after PPN stimulation has so far focused on the ability to initiate locomotion and quantitative measures of the locomotor performance (distance traveled, speed profiles, number of stops). This analysis ignored if the locomotor recovery is performed naturally with the same coordination as in non-parkinsonian animals and if it is adaptable to complex environmental needs. To address these issues, we performed three complementary analyses aimed at evaluating the proficiency of recovered locomotion.

First, to assess possible changes in coordination we analyzed intralimb kinematics. For this, we used markerless tracking of limb movement (DeepLabCut[57]) in the Open Field from mice

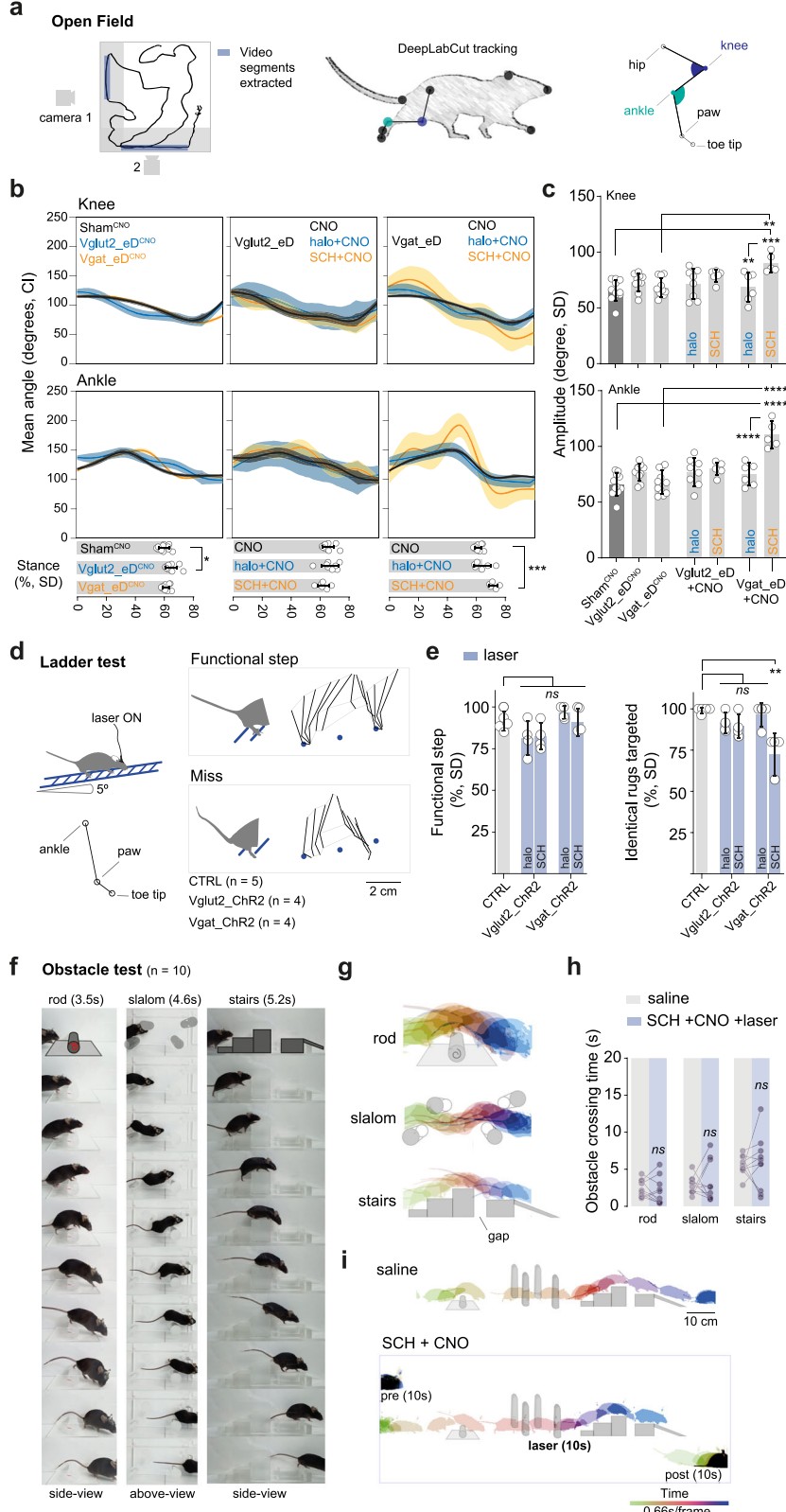

in seven different conditions: treated with CNO-only (Sham, Vglut2_eD, and Vgat_eD mice) or challenged with drug followed by treatment with CNO (Vglut2_eD and Vgat_eD-mice; challenged with halo or SCH). We relied on mice spontaneously passing close and parallel to the walls (Fig. 10a). Only a subset of mice had sufficient video segments for this analysis (5–11 mice/ condition, detailed in the figure legend) and for those, the overall

speed range profile was maintained (Fig. S10a) indicating that even though only a subset of locomotor bouts was performed in proximity to the walls those were representative of the overall behavior evaluated. For analysis, we used only the video segments where mice locomoted at average speeds between 5–20 cm/s resulting in step frequencies between 2–5 Hz (Fig. S10b, c) (average of 40.69 s of compiled video segments per mouse/condition with CI

**Fig. 10 Motor proficiency and adaptive locomotion in parkinsonian mice rescued by caudal PPN activation. a** Intralimb kinematic analysis of over-ground locomotor recovery via PPN chemogenetic activation in Vglut2cre, Vgatcre, and Sham mice injected with CNO, with or without drug-induced dopamine signaling interference. DeepLabCut[57] identification of nine reference points on the mouse body. From the hindlimb, knee and ankle joints were used for kinematic. Experimental groups: 11 Sham$^{CNO}$, 10 Vglut2_eD$^{CNO}$, 11 Vgat_eD$^{CNO}$, 8 Vglut2_eD$^{halo+CNO}$, 6 Vglut2_eD$^{SCH+CNO}$, 6 Vgat_eD$^{halo+CNO}$, and 5 Vgat_eD$^{SCH+CNO}$ mice. **b** Average angle of travel for knee and ankle joints during a complete step cycle (shadow-colored zones 95% CI). Horizontal bar graphs in lower panels show stance duration for each mouse (circles) and group mean ± SD (% of step cycle in stance). One-Way ANOVA with all seven groups in conjoint, $F_{(6, 50)} = 6.473$ with $p < 0.0001$, report for Tukey's multiple comparison. **c** Travel amplitude for knee (above) and ankle (below) joints. Individual values are represented by circles with group mean ± SD (One-Way ANOVA with all seven groups in a conjoint, report for Tukey's multiple comparison). **d** Left panel, experimental setup for inclined ladder test and joints tracked to reconstruct hindlimb motion. Right panel, stick diagrams of a single hindlimb step with functional positioning—and weight-bearing—or misses caused by misplacement or slippage. **e** Left panel, cumulative percent of functional steps across all trials for each mouse (circles) and group mean ± SD per tested condition. Control mice (CTRL, gray bar), Vglut2_ChR2, and Vgat_ChR2 mice (blue bars) (Kruskal–Wallis, report shows Dunn's multiple comparison vs CTRL). Right panel, percentage of steps in which the hindlimb is placed on the rug that was previously targeted by the ipsilateral forelimb (Kruskal–Wallis, report shows Dunn's multiple comparison vs CTRL;Vgat_ChR2$^{SCH+opto}$ significantly different form CTRL $p = 0.0064$). **f** Obstacle corridor containing a rotating rod (left), a slalom (center), and a set of stairs (right). Images show sequential frames of a drug-free mouse passing each obstacle. **g** Color-coded time-lapse of images in **f**. Mouse moves from left to right (obstacle type shown as gray schematics). **h** Average time spent to pass an obstacle before (saline-injected, baseline condition, gray) and during PPN-driven recovery of locomotion in mice where Vglut2-CnF neurons were inhibited with CNO (blue). Mice were made akinetic by D1 antagonism (SCH23390, 'SCH'), and then the locomotor output was aided by optogenetic stimulation of caudal glutamatergic PPN neurons (challenge condition). ($n = 10$ mice/condition, Two-way RM ANOVA with report for Holm–Sidak's multiple comparison). **i** Color-coded time-lapse of a mouse crossing the full corridor during baseline (saline, upper panel) and after SCH and CNO (lower panel). One frame every 0.66 s, three epochs, "pre": just before light stimulation; "laser": during laser aided locomotion; "post": after the end of stimulation (10 s for each epoch, "laser" at 40 Hz with power and parameters as in Fig. 7). Data composed of **a–c** Mice from experiments reported in Figs.1–3 and 9; **d, e** same mice cohort as in Fig. 6; **f–i** same mice cohort as in Fig. 7. halo haloperidol [D2 antagonist], SCH SCH23390 [D1 antagonist]. See also Fig. S10 and detailed stats in Table S7. Source data are provided as a Source Data file.

of 29.62 to 51.77 s, other speed ranges were not sufficiently sampled for appropriate analysis). All PPN targeted mice had shank angular excursion curves closely corresponding to reported walking patterns in mice[58] (Fig. S10d, e) (all curves peak between 79–81% of the cycle progression with exception of Vgat_eD$^{SCH+CNO}$ which peaks later, at 89.87%). Angular variations of the knee and ankle joints, as a function of time and stance phase duration, were similar between all groups but differed in Vgat_eD$^{SCH+CNO}$ mice which had longer stance phase and joint hyperextension (Fig. 10b, c) (Knee amplitude: One-way ANOVA $F_{(6, 50)} = 4.672$ with $p = 0.0008$, followed by Tukey's with only Vgat_eD$^{SCH+CNO}$ mice differing from other conditions, vs Sham$^{CNO}$ $p = 0.0004$, vs Vgat_eD$^{CNO}$ $p = 0.0020$ and vs Vgat_eD$^{halo+CNO}$ $p = 0.0096$. Ankle amplitude: One-way ANOVA $F_{(6, 50)} = 13.02$ with $p < 0.0001$, followed by Tukey's where again Vgat_eD$^{SCH+CNO}$ stands as different from all other groups with $p < 0.0001$ in all comparisons). Altogether, this indicates that (with exception of Vgat_eD$^{SCH+CNO}$ mice) chemogenetic activation evokes normal limb kinematics and that motor recovery leads to limb kinematics that are mostly indistinguishable from that of a normal mouse.

Second, to analyze gait and balance interactions we evaluated the locomotor proficiency of mice moving on an inclined Ladder test (40 cm length, 3–5 trials/mouse). Vglut2_ChR2 and Vgat_ChR2 mice, made akinetic (halo or SCH) and aided by optogenetic PPN activation were compared to a control non-interference group (CTRL). Mice were not trained prior to this task, yet the CTRL group crossed the ladder almost without slips or missteps, and hindlimbs were placed on the rug that was previously occupied by the ipsilateral forepaw, indicating perfect coordination between fore- and hindlimbs. Similarly, light stimulation allowed akinetic mice to walk up the ladder with the same proficiency as the CTRL group (Fig. 10d, e) (Functional steps range between 81.46 to 96.74% with max SD of 10.3% observed in Vglut2_ChR2$^{halo}$ group. Kruskal–Wallis between the five groups not significant, $p = 0.0674$, K–W = 8.758, laser parameters same as in the Open Field). The only noticeable difference was that Vgat_ChR2$^{SCH+opto}$ mice had reduced fore-hindlimb coordination, suggesting gait alterations reduce loco-motor proficiency in mice with stimulation of GABAergic PPN

neurons made parkinsonian with D1 antagonist SCH (identical rugs targeted more than 89% of the analyzed steps with max SD of 7.14%, Dunn's multiple comparison to CTRL group shows a significant drop in hindlimb coordination only for Vgat_ChR2$^{SCH+opto}$ mice, $p = 0.0064$ and mean rank difference of 13%, Kruskal–Wallis $p = 0.0233$, K–W = 11.30. The average speed of locomotion indicates Vgat_ChR2$^{SCH+opto}$ mice also moved at slower speeds during ladder crossing than all other groups, Dunn's $p = 0.0218$ with average speed $5.4 \pm 1.2$ [SD], whereas other groups crossed the ladder at speeds between 6.2–7.7 cm/s, One-way ANOVA, $F_{(4, 15)} = 3.332$ with $p = 0.0385$). In summary, skilled locomotion is normally executed after locomotor recovery aided by caudal Vglut2-PPN stimulation.

The third motor analysis evaluated proficiency of sensory-motor integration. For this purpose, mice were exposed to a complex environment containing three obstacles that it needed to pass (i.e., Obstacle test): a rod that rotates over a free base, a slalom that requires a zig-zag motion, and a set of stairs with ascending and descending steps separated by a central gap (Fig. 10f, g). We compared the obstacle crossing time of mice before (saline; baseline) and after SCH aided by optogenetic activation of glutamatergic PPN neurons, while Vglut2 neurons in CnF were inhibited (SCH + CNO with Opto; challenge) (same mice as in Fig. 7). To have equal sample sizes between the two conditions, the number of laser-ON epochs was tailored to the individual baseline values (with a cap number of 20 epochs, laser parameters same as in the Open Field) (Fig. S10f).

When comparing the two conditions, no effect is observed in obstacle crossing time, regardless of obstacle type (Fig. 10h, i and S10g) (Overall speed: two-tailed, paired $t$-test $p = 0.4390$, $t = 0.8096$, df = 9. Obstacle crossing time: Two-way RM ANOVA, interaction $F_{(2, 27)} = 0.1591$ with $p = 0.8536$). We confirmed the akinetic phenotype of SCH injected mice in the Bar test. Just prior to start SCH-mice were incapable of descending the bar but could promptly descent when glutama-tergic PPN neurons were activated (Fig. S10h) (Two-way RM ANOVA, condition $F_{(2, 40)} = 443.5$ with $p < 0.0001$, followed by Bonferroni Start vs End $p = 0.0172$ but mean difference of only 1.43 s whereas Start vs End$^{+opto}$ $p < 0.0001$ with a mean

difference of 12.6 s). This test shows that the recovery of motor function permits the animal to navigate a complex environment with proficiency.

Together, these experiments highlight that locomotor recovery via Vglut2-PPN stimulation, by either chemogenetics or optogenetics, leads to normal locomotor coordination with a good level of sensory-motor integration allowing the animal to perform skilled motor tasks and navigate a variety of complex environmental challenges. The recovery is not proficient when the GABAergic PPN neurons are targeted.

## Discussion

Patients with severe Parkinson's disease have profound difficulties in self-initiating and maintaining locomotion. This motor dysfunction might arise in part due to excessive basal ganglia inhibition of brainstem motor control areas. In this study we have used a pharmacological approach to partially mimic parkinsonian motor symptoms and report that targeted activation of glutamatergic neurons in the caudal Pedunculopontine nucleus counteract drug-induced akinesia and bradykinesia, effectively transitioning animals from a robust parkinsonian phenotype into a normalized and adaptable locomotor output. While prolonged activation of GABAergic PPN neurons also increased the locomotor output mice maintained a bradykinetic phenotype. Our observations pinpoint the caudal glutamatergic PPN neuronal subpopulation as a suitable target for neuromodulatory restoration of locomotor function in Parkinson's disease.

**Methodological considerations**. In the present study, we use pharmacological tools to mimic a PD state. The pharmacological models, by interference with dopaminergic signaling, partially simulate the neurochemistry of PD without producing the pathophysiological features. The ideal model of PD should consist of clinical features including motor and nonmotor symptoms, develop progressively, and demonstrate the neurodegenerative hallmarks typical of PD. Currently, available models do not encompass all of these features. For example, the 6-OHDA rodent model with bilateral injection in the striatum or medial forebrain bundle, that leads to retrograde degeneration of dopaminergic neurons in the midbrain, only reaches a stable and chronic depleted state 3–4 weeks after injection[59–62]. However, in this chronic stage rodents might not show a strong locomotor phenotype[63–65]. Rodents injected with 6-OHDA and tested during the first-week post-surgery do show akinesia[66,67]—because the initial damage to axons takes place within 24 h[59,60]—but this stage corresponds to an acute dopamine-depleted state. Moreover, these early-stage animals also show symptoms not related to PD which could add confounders to the model[63,65,68,69]. Therefore, we chose a pharmacologic approach because it allows for an acute, strong, and robust motor phenotype which is entirely caused by the hypodopaminergic syndrome and with a phenotype not seen to the same extent in toxin-induced and genetic slowly developing models of PD[39]. Moreover, working with a transient parkinsonian motor-state allowed for both healthy and parkinsonian-like phenotypes to be compared within the same experimental round. The parkinsonian mimicry of our approach is supported by the observation in rats, showing that systemic or local striatal[70] injection of D1/D2 antagonists reduces the firing rate of striatal MSNs and fast-spiking interneurons, whereas tonic firing neurons increase their activity[71]. Our calcium imaging data support these findings by showing that both haloperidol and SCH23390 lead to reduced striatal D1-MSN activity as well as silencing of glutamatergic neurons in the PPN. Therefore, although the pharmacological approaches used in this work do

not capture the full spectrum of PD it does recapitulate BG dynamic changes that lead to parkinsonian states.

The systemic administration of dopamine-receptor antagonists may also act on dopaminergic receptors in PPN itself and add to the akinetic phenotype. For instance, dopaminergic cells in Substantia nigra pars compacta (SNc) and Zona Incerta project to PPN and may have their locomotor promoting effect through D1 receptors present on PPN neurons[28,72]. Thus, systemically blocking D1 receptors -similar to when dopaminergic SNc neurons degenerate in PD—will affect both striatum and PPN leading to increased BG inhibition of its targets and loss of facilitation within PPN. Together, this would augment the akinetic state associated with D1 antagonism and any beneficial effects of PPN activation may not be solely linked to its interconnectivity with the BG but also to local D1 receptors. This is, however, not the case for haloperidol since D2 receptors have not been found in neurons that reside inside the PPN. Moreover, the akinetic and bradykinetic state observed in our dopamine-receptor antagonized mice has been described in rodents with striatal cannula infusions of these drugs, suggesting that a major part of the drug-induced effect is mediated by BG dysfunction.

Other possible confounders, that could influence the outcome of the study, are the use of viruses and chemical activators. CNO is naturally metabolized into clozapine[73] and high dosages could initiate a cascade of clozapine-like side-effects. These include hypotension, sedation, and anticholinergic syndrome which do not support increased motor output and would, in fact, worsen the akinetic phenotype. We have used a low dosage of CNO shown to preferentially drive DREADD-mediated behaviors[74] and we find that in Sham mice CNO was behaviorally inert like previous findings[18,75]. Similarly, the effects of optogenetic stimulation were observed only when we applied the optimal wavelength needed to activate the Channelrhodopsin and absent in control-virus injected mice.

**Caudal glutamatergic PPN neurons: a locomotor pathway to restore motor function**. PPN has been implicated in locomotor deficits in human studies[24,76] because it is under direct inhibitory control from BG output regions[2,15,37,77–79]. Recent studies in healthy rodents by us and others have shown that activation of glutamatergic PPN neurons can initiate and maintain locomotion primarily within the exploratory speed range[18,19,29] which would make this cell type an ideal candidate for locomotor recovery in PD. However, other studies have been unable to see locomotor initiation by optogenetic or chemogenetic stimulation of PPN-Vglut2 neurons[17,80,81]. In these studies, the optogenetic stimulation instead elicited phasic[17] or tonic muscle activity[80] in resting animals or decreased locomotor speed during ongoing locomotion[17] while chemogenetic activation of PPN-Vglut2 neurons did not change locomotor distance traveled in an open field but did increase the amount of wheel running[81]. These discrepancies may seem difficult to reconcile but likely reflect activation of functional heterogenous glutamatergic PPN neurons. A recent study in mice has indeed shown that subpopulations of glutamatergic neurons in PPN with axonal projection to either spinal cord, medulla or the substantia nigra reticulata may be related to diverse motor actions including body extension, locomotion, or rearing[82]. Moreover, we have shown that optogenetic stimulation of glutamatergic PPN neurons that express the transcription factor Chx10 produce instantaneous full-body motor arrest including the arrest of locomotion[33,34]. These PPN-Vglut2-Chx10 neurons are located in the rostral most part but absent from the caudal PPN region. Activation of the PPN-Vglut2-Chx10 neurons never induces locomotion. Vglut2

targeting of neurons in the rostral and caudal part of PPN will therefore produce opposing motor responses with rostral inducing motor arrest and caudal promoting locomotion (see also ref. [29] with data from rats). Targeted caudal PPN-DBS has also been shown to have the best clinical results[26–28,51]. In the present study, we therefore aimed at targeting the viral expression to the caudal PPN similar to what we did in our previous study using optogenetics[18]. Here, we further adjusted the fiber position so that light would only reach the caudal PPN (see Fig. S3f–h) and used 40 Hz stimulation to activate PPN neurons. With this approach, we observed that prolonged (10 s) activation of caudal glutamatergic PPN neurons consistently promotes a sustained increase in locomotor output and that optogenetic stimulation increases locomotion only during stimulation. The prolonged stimulation caused locomotor initiation in nearly every trial, even more reliable than we observed with shorter stimulation tested previously[18], possibly because of the long stimulation duration and high stimulation frequency. Similarly, targeting excitatory DREADDS to caudal glutamatergic PPN promotes a sustained increase in locomotor output. Using the same experimental approach, we then show that chemogenetic and optogenetic stimulation of glutamatergic PPN neurons reverts the motor deficits in both dopamine signaling depleted mouse models used. The activation completely alleviates the akinetic state and mice can move with varying speed ranges expressing the same speed profile of WT mice. The locomotor restoration was present even when glutamatergic neurons in CnF were silenced demonstrating that glutamatergic PPN neurons alone can facilitate locomotion. Therefore, the recovery of locomotion is not mediated through indirect or direct activation of glutamatergic CnF neurons, which recently have been shown to be able to initiate locomotion optogenetically in acute toxin-induced dopamine-depleted mouse model[66]. The strikingly different darting phenotype observed in our study after direct chemogenetic activation of glutamatergic CnF neurons also supports the notion that the stimulation reported here is specific to glutamatergic PPN neurons. Together our results show that selective targeting of glutamatergic neurons in the caudal PPN completely restore the quantitative locomotor parameters (distance and speed) in mice with acute and severe dopamine signaling depletion. The sustained phenotypic rescue raises the question if the activation of glutamatergic PPN neurons promotes qualitative normal and adaptable locomotor movement. The kinematic of locomotion after recovery was qualitatively similar to locomotion in healthy mice showing that stimulation of caudal Vglut2-PPN neurons leads to appropriate recruitment of the downstream locomotor network even if dopamine receptors are ubiquitously antagonized. The performance of skilled locomotion was also similar to healthy mice which suggests that when the glutamatergic PPN activity is increased it may enhance cortical driven responses[83] to produce normal 'self-paced' movements[84,85]. Finally, the recovery of motor proficiency observed in the obstacle course highlights that the optical activation of glutamatergic PPN neurons—in the absence of CnF activity—does not lead to automatic or 'robotic' movement, but rather releases movement that is adaptable to a complex environment. These experiments show that the caudal Vglut2-PPN neuron stimulation not only rescues quantitative locomotor parameters but also their qualitative expression including skilled and adaptive locomotion. To the best of our knowledge, no reports of recovery of such strong drug-induced akinetic phenotype exist in the literature, although there are publications showing that chronic treatments[86–89] can reduce its severity, those are mostly dependent upon pre-treatment approaches. A caveat when translating these results into a clinic is that PD is accompanied by a neurodegenerative process that may hamper the outcome of targeted brainstem stimulation as a tool to repair motor function.

**GABAergic PPN neuron activation does not revert the locomotor phenotype**. Glutamatergic neurons are intermingled in the caudal PPN with GABAergic neurons and in the present study, we directly compared both populations using the same protocols. In healthy mice, prolonged activation of caudal GABAergic PPN neurons increases the locomotor distance traveled albeit constrained to slow ranges of speed and interrupted by frequent brief stops. The increased distance traveled was surprising since previous reports from us and others showed that short-lasting light activation[18,19] of GABAergic PPN was unable to evoke locomotion or that it decreased ongoing speeds. However, here we found that the locomotor initiation from rest had a long latency which explains why it was not detected with short-lasting stimuli. Moreover, the prolonged locomotion was interrupted by frequent stops and slow speeds (5–10 cm/s). Therefore, it appears that the prolonged stimulation initiates a mixed effect of long-latency locomotor initiation superimposed by short stops. The network mechanism for the effects of GABAergic PPN neuron activation are not easily explained but they might originate from activation of intrinsic local-PPN[90] and/or PPN-BG connectivity[52]. Brief stops could arise from local GABAergic inhibition of the glutamatergic PPN population whereas long-range projections to, among others, excitatory subthalamic nucleus (STN)[78,91,92] could promote the initiation of movement[93]. In accordance, after haloperidol STN neurons in the indirect pathway are expected to be strongly active and their inhibition by GABAergic PPN neurons would therefore have a rescuing effect. In contrast, when the direct pathway is silenced (SCH23390) locomotion cannot be improved by stimulating GABAergic PPN neurons. In support of this suggestion, procedures that reduce STN output have been found to reverse the behavioral effects of dopamine depletion in rodents[94,95], primates[96], and humans[97]. Concomitant activation of long-range and intrinsic connectivity may therefore explain the GABAergic PPN behavioral phenotype.

Whatever the precise mechanism is, it is the continuous re-engagement in walking gait, reported here, which opened the possibility that by promoting caudal GABAergic PPN neuronal activity some level of motor recovery could be achieved. Although locomotion could be initiated by chemogenetic GABAergic PPN neuronal activation in healthy mice we found that in akinetic mice locomotor performance could only be partially restored. Specifically, the chemogenetic stimulation counteracted reduced locomotor activity induced by the D2 antagonist haloperidol while the effect was absent in D1-antagonized mice. The bradykinesia remained after stimulation and mice could only move slowly. The intralimb kinematics was disturbed exclusively in D1-antagonized mice, with a longer stance phase accompanied by joint hyperextension during walking gait. Probably as a consequence, these mice had reduced hindlimb-forelimb coordination during the Ladder test. When we restricted the stimulation period using optogenetics moderate or no recovery was observed. Altogether the analysis shows that although locomotor output was facilitated by caudal GABAergic PPN neuron stimulation the locomotion was not reversible as seen with caudal Vglut2-PPN stimulation.

Facilitation of motor initiation in the Bar test was the one parkinsonian feature consistently recovered by GABAergic PPN neurons. The projections from GABAergic neurons to areas within the superior colliculus, that we show here, and the well-known projections from superior colliculus to PPN demonstrate reciprocal connectivity between the two brain regions that might be used in sensory-motor gating[55,98]. We propose that GABAergic PPN neuronal stimulation promotes motor response in the Bar test because this task contains sufficient sensory salience and its performance relies on visual and tactile motor gating.

**Consequences of the findings for the treatment of motor deficits in human PD patients.** Deep Brain Stimulation (DBS) has been increasingly used in patients with intractable motor symptoms that may remain refractory to dopamine replacement therapy. DBS in the ventralis intermedius of the thalamus efficiently controls tremor while DBS in STN or Gpi have been used to treat bradykinesia but with little success in unlocking locomotor freezing[12,13,99]. More recently, stimulation targeting the PPN has been proposed as a tool to alleviate locomotor deficits in PD patients with the assumption that activating the PPN with DBS should release it from BG inhibition[26,27,76]. In support, akinesia is counteracted by PPN-DBS in PD primate models[100]. However, PPN-DBS has shown mixed results, with both improvement of locomotion and motor inhibition, in the clinic. Variability in stimulus parameters, electrode location, and patient's clinical profile have been considered as explanatory factors for the mixed outcomes of PPN-DBS[24,79]. However, contrary to other typical PD-DBS targets, the PPN consists of glutamatergic, cholinergic, and GABAergic cells that are both functionally and anatomically heterogeneously distributed[51,55]. The results presented in this study show that localization and cell-specific targeted activation of caudal glutamatergic PPN neurons could provide consistently, and prolonged facilitation of a proficient locomotor output suggesting that the caudal PPN glutamatergic subpopulation should be targeted for treatment in humans. These findings shift the focus away from the cholinergic PPN neurons that traditionally have been proposed to be the target for neuromodulatory PPN approaches[25,101]. The fact that cholinergic PPN neurons exhibit a little effect on locomotor output[17,18,81] represent only 20% of the local neuronal population[52,53], and in addition degenerate during PD progression[102–105] support this conjecture. Presumably, PPN-DBS will also activate the GABAergic PPN neurons that are intermingled with the glutamatergic population within the caudal PPN. Since cell-specific stimulation of caudal GABAergic PPN neurons does not revert locomotion to a normal locomotor phenotype in our parkinsonian model it suggests that concomitant stimulation of glutamatergic and GABAergic neurons in a DBS setup may contribute to the unreliable results in clinic studies. A similar situation is present for the cuneiform nucleus which has been proposed as a possible target for alleviating locomotor deficits in PD patients[31,66,106]. Glutamatergic neurons are intermingled with GABAergic CnF neurons that with short-lasting stimulation inhibit locomotion[18]. Any electrical DBS of CnF will therefore also engage a mixed neuronal population.

A direct translation of our results into a human setting would, therefore, require cell-type-specific artificial expression of chemical or optical actuators which is currently not employed but might be possible in a not too far future. In the current absence of such possibilities, we suggest that neuromodulatory interventions in PPN should focus on the caudal PPN and take advantage of technical advancements such as closed-loop approaches[107] to employ stimulation parameters tuned to activate glutamatergic neurons when motor output is most required.

## Methods

**Experimental subjects**. All animal experiments and procedures were in accordance with the EU Directive 20110/63/EU, approved by the Danish Animal Inspectorate (Dyreforsøgstilsynet, permit: 2017-15-0201-01172, P21-326), and followed the ARRIVE guidelines[108].

The study was performed in heterozygous Vglut2[cre 18,36] or Vgat[cre 56] mice for Pedunculopontine (PPN) and Cuneiform nucleus (CnF) targeting, and Drd1[cre] (Tg(Drd1a-cre)150Gsat) mice for dorsal striatal targeting. Mice were kept on a C57BL/6 genetic background. Wild-type (WT) and heterozygous littermates were used as controls. Adult (8–10 weeks) male and female mice were group-housed (up to 5 mice) on a 12-h light cycle with food and water ad libitum (housing temperature 23–24 °C, 45–65% humidity).

## Surgical procedures

*Preparation for viral injection, probe implantation*. Mice were initially anesthetized with 3% isoflurane and maintained at 1–2.5% after placement in the stereotaxic frame (Kopf SD479, Neurostar, StereoDrive). Sterile ophthalmic ointment was applied, and the body temperature was maintained at 37 °C (with the aid of a temperature controller). The scalp was sterilized with ethanol 70% and povidone-iodine before skull exposure. For horizontal plane leveling, the relative dorsoventral displacement of Bregma and Lambda was adjusted to a maximum of 0.05 mm difference. For adjusting tilt, two points equidistant to the left and right of Bregma were used. Craniotomies were made using a hand drill (David Kopf Instruments, Model 1474, drill bit H1.104.008), followed by removal of the underlying dura using a fine needle tip. The brain surface was covered with sterile physiological saline to prevent desiccation.

*Viral injection*. Viruses were front filled into a glass pipette (NanoW) filled with mineral oil (Millipore Sigma, M3516) and connected to a nanoinjector (Neurostar, Glass capillary Nanoinjector). The glass pipettes were pulled to obtain a tip diameter of ~50–80 μm (Narishige, PP-830 puller). Viruses were infused into target regions at 50 nl/min, and the pipette was kept in place for an additional 8 min, then slowly withdrawn (dorsoventral speed 0.1 mm/s, entry and exit). Mice were given postoperative buprenorphine (0.1 mg/kg, subcutaneous) and monitored for 3 consecutive days.

*Optical fiber implantation*. Optical fibers (200 μm core, NA 0.22, 1.25 mm ceramic ferrule, Thorlabs, Germany) were implanted 7–12 days after viral infection. The surgical procedure followed the protocol described above. The skull was perforated at the desired coordinate, then a fiber was inserted (0.1 mm/s) and secured with adhesive primer (Optibond FL, Kerr) and cement filling (Tetric Evoflow #A1, Ivoclar). Mice had the external area of the head plate painted in black to allow for adequate head detection during video tracking.

*Lens implantation*. All material used for calcium imaging experiments was acquired from Inscopix (Palo Alto, USA). One month after viral injection, a gradient index (GRIN) lens was implanted using a high precision dual-arm mouse stereotaxic (0.01 mm digital resolution, David Kopf Instruments, Model 942LS). Surgical steps[109] were performed as follows. First, a head screw (Pinnacle, #8209-0.10") was placed near the opposing anterolateral corner of the frontal bone. Then, all connective tissue, muscle, and tendons within the head cap area were removed with a dissecting chisel, and air pressure was used to clear the surface from debris. Drilling was performed based on an outer lens diameter. To minimize pressure, aspiration of tissue above the region of interest was performed using a 22-gauge blunt needle attached to a vacuum pump (stop 1.5 mm dorsal to the target site). After aspiration, straight GRIN lenses were directly implanted (see the segment on stereotaxic coordinates). For prism lenses with triangular edges, we created a medial plane track parallel to the virus injection site and perpendicular to the skull using a straight-edged dissection knife (FineScience, #10055-12). The knife was lowered (0.05 mm/min) at −0.4 mm posterior from the target site until reaching the depth at which the lens would be inserted. Subsequently, the knife was moved rostrally (0.4 mm) and slowly retracted to create a path for the prism lens[110]. Once inserted, the lens (straight or prism) was secured to the skull (same as optical fibers) and covered with a plastic cap to protect its surface. Six weeks after lens implantation, the miniscope baseplate was mounted using the attached microscope to determine the field of view (FOV). Next, the FOV was inspected in the awake animal and only mice in which fluorescence variation could be observed with low LED irradiance (<1.2 mW/mm$^2$, 455 ± 8 nm) were used in behavioral analysis.

**Stereotaxic coordinates, injection volume, and virus list**. Coordinates are given in reference to Bregma. The angle for the path is given relative to the rostrocaudal axis along the sagittal plane, with 0° being vertical and negative values denoting a posterior-to-anterior path. Chemogenetic experiments were done bilaterally (exception: c-Fos expression analysis, unilateral right hemisphere) whereas optogenetic and calcium imaging experiments were done unilaterally. In optogenetic and calcium imaging experiments, increased target specificity was achieved by implanting through a path that differed from the viral delivery path by a 20° or 30° angle (see target coordinates for details). The mean (±SD) distance between Bregma and interaural Lambda was 4.44 ± 0.2 mm (n = 81) and mice with follow-up optic fiber placement had an average difference of 0.11 ± 0.1 mm (n = 37) amongst Bregma–Lambda measures from first to second surgical procedure.

*Target coordinates (mm) and volume*. For neuronal activation of PPN caudal region, virus injection (80 nl): anteroposterior (AP) = −4.72 or −4.59, mediolateral (ML) = ±1.28, dorsoventral (DV) = −3.38 or −3.45 at 0°. For DREADD-mediated neuronal inhibition or activation of the CnF, virus injection (50 nl): AP = −5, ML = ±1.3, DV = −2.9 at +5°. For optogenetic PPN fiber placement: AP = −4.84 or −4.72, ML = 1.3, DV = −2.99 or −2.96 at −20°. In CnF injected mice, fiber placement was adjusted to: AP = −5.2, ML = 1.3, DV = −3 at −30° to further restrict CnF fiber passage. For PPN calcium imaging, virus injection: AP = −4.84 and −4.72, ML = 1.22, DV = −3.61 and −3.29 at −20° (35 nl per location, 5 min wait point within), followed by either straight lens (0.6 or 1 mm diameter),

AP = −4.8, ML = 1.2, DV = −3.25, or prism lens (1 mm), AP = −4.8, ML = 1, DV = −4, at 0°. For striatal calcium imaging, virus injection: AP = 0.5, ML = 1.5, DV = −3.6 and −3.0 at 0° (150 nl per location, 10 min wait-point within) and straight GRIN lens (1 mm) AP = 0.4, ML = 1.5, DV = −2.6 at 0°.

*Viral vectors.* For channelrhodopsin (ChR2) expression: AAVdj-EF1alfa-DIO-hChR2(E123T/T159C)-P2A-mCherry-WPRE, 9x10e11 a kind gift from Dr. Deisseroth. Excitatory DREADDs (eD); AAV5-hSyn1-DIO-hM3D(gq)-mCherry-WPRE, 4x10e12, Viral Vector F-Zurich (v89-5). Inhibitory DREADDs (iD); AAV5-DIO-HA-hM4D(gi)-mCitrine-WPRE, 1.6x10e12, Viral Vector F-Zurich (v93-5). Calcium imaging: AAV1-hSyn1-DIO-GCaMP6s-WPRE-SV40, 0.5x10e13, Addgene #100845). Control vectors: for heterozygous mice (virus lacking opsin), AAV5-hSyn1-DIO-mCherry, Viral Vector F-Zurich (v116-5); wild-type mice were injected with AAVretro-Ef1a-mCherry-IRES-Cre, Addgene (#v15410) or sham surgical procedures were performed with vehicle. Viral aliquots were stored at −80 °C with no follow-up use.

**Drug preparation**. All injections were performed intraperitoneally (ip) with a volume of 10 ml/kg body weight. The experimenter was not blind to group-drug allocation. Solutions were freshly prepared on the experimental day. Clozapine-N-oxide (CNO, 1 mg/kg, Tocris, #4936) was dissolved in sterile physiological saline. The preferential D2 antagonist haloperidol ("halo", 0.5 mg/kg, Tocris, #0931) was dissolved in 5 µl of 0.5% lactic acid and saline. The D1 antagonist SCH23390 ("SCH" [R( + )-7chloro-8-hydroxy-3-methyl-1-phenyl- 2,3,4,5-tetrahydro-1H-3-benzazepine], 0.25 mg/kg, Tocris, #0925) was dissolved in saline. Control groups were injected with either CNO or saline (with or without lactic acid, accordingly).

**Behavior experiments**. Experiments were performed between 9 a.m. and 5 p.m. (dark cycle transition at 6 p.m.). Mice belong to the same cohort if they were tested sequentially (behavioral battery) and in parallel (balanced group distribution over each test day). Cohorts are presented in Supplementary Table S8. Mice were habituated to handling and injection (using saline) for 3 days leading up to their first behavioral test. Following each test, individuals remained in a clean cage until all its cage mates had been assessed, then housing groups were reassembled. Between tests, a non-experimental period of minimum 4 days was respected to diminish the risk of receptor desensitization/downregulation. All chemo- and optogenetic experiments were initiated 3–4 weeks after viral infection. Calcium imaging experiments were initiated 2 months after viral infection.

**Open Field**. Each mouse was placed in a square (50 cm × 50 cm) or circular (40 or 50 cm diameter) chamber. Basic features (color, shape, cleaning odor) were varied between experimental days to avoid drug-to-chamber associations (illumination maintained equal, 100 lux). Mice were video monitored on a multicamera setup with one above and two side-view cameras (25 fps, resolution of 1280 × 1024 per camera view; GigE monochrome acA1300-60gm Basler camera, LMVZ4411 1/1.8″ 4.4-11 mm/F1.6 aberration correcting Kowa lens). Using the above view, the positions of nose, tail, and center of mass were tracked with Ethovision 15.0 software (Noldus). Side views were analyzed in parallel to determine body contour.

*Data preprocessing and outcome measures.* A locally weighted smoothing algorithm (2-degree nonlinear polynomial fit) was applied to all tracks, based on a half-size window of eight frames before and after each x, y position[111]. To further filter out small movements of the subject's center point, a minimal distance moved threshold of 0.5 cm was applied to the entire data set. Thereafter, a series of parameters were calculated and segmented to obtain discrete behavioral categories (i.e., outcome measures). Locomotion was defined as periods when the speed of the animal's center point, averaged over four frames, remained above 2 cm/s. This moving average window size was maintained for all calculations and used to report distance moved, speed, and acceleration. Darting is defined as having high acceleration during a relatively short locomotor segment resulting in abrupt and fast bouts of movement. To estimate the propensity of an animal to dart, we divided the maximum speed reached during a locomotor bout by its duration, resulting in the variable Maximum Speed to Duration ratio (MSD ratio, cm/s/s) (for details see ref. [50]). To measure the average MSD ratio of an individual mouse, all locomotor bouts during a recording session were segmented and individual averages are presented. Stops were defined as shifts from locomotion to a continuous period during which the speed dropped below 2 cm/s (minimum three consecutive frames) and the average pixel change of the entire video image was less than 0.5% (minimum 0.2 s) accompanied by low body contour changes (below 2%). Limits to body contour pixel change in stop event detection were set so as to exclude in-place behaviors such as grooming, rearing, and stretch-attend postures. In-place behaviors are performed at low speed but accompanied by body contour alterations, such as postural changes. Hence, a locomotor period (mouse moving >2 cm/s) can be terminated with a stop and immediately followed by grooming, in which scenario only the time prior to the grooming initiation corresponds to a stop bout. If, after grooming, the mouse remains inactive and thereafter locomotes, this video fragment will contain only one-stop bout (i.e., locomotion → stop → grooming → inactive → locomotion). Validation of stop bout automatic detection was performed by manually annotating a 2 min video clip of each experimental day

(ground truth, the experimenter was blind to automated annotation, concordance 93 ± 3.2% [SEM between days]).

Open Field data are presented as raw values or normalized to facilitate comparisons between experimental conditions. Normalization was done by dividing the group average by either control group values or prior conditions. Trace example images were reconstructed using preprocessed track data exported from Ethovision and replotted using the JMP data analysis software (v14.3.0, SAS Institute Inc).

*Chemogenetics in the Open Field.* In CNO-only experiments involving chemogenetic activation, mice were injected with CNO and placed in a waiting chamber for 5 min. Subsequently, mice were moved to the Open Field and their activity was recorded for 50 min (Figs. 1, S6c–h, and 8). For the parkinsonian state challenge, we first induced an akinetic condition by injecting mice with dopamine-receptor antagonists and placing the animal in a waiting chamber for 5 min. Thereafter, mice were transferred to the Open Field and recorded for 20 min (first period). To assess motor recovery, the second injection of either saline or CNO was performed, and mice were immediately placed back into the chamber (second period, +30 min). Altogether, this procedure resulted in test sessions of ~55 min/mouse. After the session mice were placed in a waiting chamber for 3 min and then performed the Bar test. In experiments with activation of CnF an additional Bar test was performed at a midpoint (between the first and second period, Fig. S6o).

*Optogenetics in the Open Field.* For light activation of ChR2-transfected neurons, we used trains of light pulses (40 Hz, 10 ms pulse duration, 473 nm, and peak light power of 2–3.5 mW at connector tip) (Optoduet, Ikecool Corporation). Laser power settings were set for each mouse and maintained throughout all experiments. Based on (https://web.stanford.edu) predicted irradiance values for mammalian brain tissue we calculated that the light intensity was less than 0.6 mW/mm$^2$ at 1 mm distance from the fiber tip (Fig. S3h) below the value needed for activating ChR2.

All photo stimuli were automatically triggered by Ethovision and converted to TTL pulses by Master-8/9 pulse generator (AMPI). Mice were habituated to tethering the day prior to experimental start (10 min, waiting chamber) and the patch cord had an integrated rotary joint to allow for free mobility.

Each trial consisted of a laser-ON period (10 s), followed by a variable laser-OFF period (70 s average, 65–75 s range). For each trial, we extracted behavioral measures during three epochs, termed "pre", "laser", and "post" (10 s each). The "laser" epoch was the interval starting with illumination and lasting until its end. The "pre" and "post" duration was either directly preceding or following it. For reported measures of latency, we assessed the time from laser start to initiation of locomotion. The fraction of successful trials was determined by quantifying the proportion of trials in which a mouse engaged in locomotion during laser-ON.

On an experimental day mice were tethered and after 5 min transferred to the Open Field. A recording of 2–3 min baseline period without light delivery (habituation) was followed by a five-trial series (trials 1–5). Mice were then injected with either saline or a parkinsonian state-inducing drug and placed back into the chamber for 2–3 min without light delivery (habituation post-injection). The laser was then automatically triggered in a series of 15 trials (trials 6–20). At the end of this procedure, animals were further recorded for 3 min without stimulation until test end (Latency). Altogether, this procedure resulted in experimental rounds of ~35 min/mouse. In experiments involving Vglut2-PPN activation with concomitant Vglut2-CnF inhibition, we adapted this procedure to inject CNO and allow for peak effect during trials 6–20. Thus, after the fifth trial, the mouse was injected with CNO and moved to a waiting cage. Twenty minutes later, the mouse was injected with one of saline/haloperidol/SCH23390 and placed back in the Open Field chamber for recording of laser-ON trials 6–20. This adaptation extends the experimental time to ~60 min/mouse. Following the Open Field session, mice were placed in a waiting chamber for 3 min and then performed the Bar test before untethering.

*Prolonged laser test.* For Vgat$^{cre}$ mice injected with ChR2 expressing virus in the PPN and treated with SCH23390 (n = 9), a retest was done to address the possibility that prolonging laser duration could influence phenotypic recovery. Five epochs of 20 s laser-ON ('laser') with "pre" and "post" epochs adjusted accordingly were performed.

*Laser wavelength dependency test.* Laser wavelength contingency experiments were performed in all ChR2 expressing mice. Mice were tethered, placed in the Open Field and, after habituation, the laser was triggered by Ethovision in a series of 8 trials. In this case, ChR2 opsin expressing mice were exposed to the 593 nm wavelength (40 Hz, 10 ms pulse duration, 10 s duration). As with the Open Field test, the trials were followed by a variable period of laser-OFF.

**Bar test**. This test measures the latency to initiate a motor response and correct an externally imposed posture. Mice were placed with both front paws on a horizontal thin bar, 5 cm from the floor, in a half-rearing position. The time retention of this posture was measured with a 20 s cut-off. Latency to descend the bar was measured three times per animal with 1 min intertrial intervals. The descent was identified

when both paws were placed on the floor and the average descent latency of each mouse was calculated.

In chemogenetic experiments, the Bar test was used to further describe phenotypic recovery by comparing descent latency to Sham mice. In optogenetic experiments, the Bar test was considered as an exclusion criterion for Open Field data analysis. Thus, optogenetic mice which did not score on average above 15 s latency (during laser-OFF), were excluded from the Open Field data set (i.e., insufficient induced akinesia). This exclusion criterion was implemented to make sure that all mice evaluated had a robust akinetic phenotype. For a subset of mice, we performed video recordings, presented on Movies 1 and 2 (side-view, infrared range, 50 fps, USB 3.0 aCA1920-155 um Basler camera, correcting Kowa lens equipped with infrared pass filter infrarot 850 ES43, 1936 × 1216 resolution, infrared LED reporting light).

**Corridor test**. This test was implemented to measure escape velocity as mice cross a linear corridor (120 cm length, 10 cm width) to avoid an air puff. This protocol has been previously used to assess the motor effects of CnF silencing[18]. Briefly, before testing mice were allowed to explore the chamber for 3 min undisturbed. The setup lights were then turned on (400 lux) and another 3 min of habituation followed. Next, mice were moved to individual waiting chambers and given saline or CNO. Twenty minutes after injection 10 to 13 trials of the corridor test were performed with 2–3 min intertrial intervals. In each trial the mouse was gently guided to the start zone and after ~20 s an air puff was applied to the back of the mouse (pressurized gas duster, ~1 s duration, tubing fixed on apparatus wall) causing the animal to escape towards the opposing end. The test was video recorded with a high-definition side-view camera (60 fps, USB 3.0 JAI GO-5000M camera, LM12HC 1″ 12 mm 5MP wide focusing Kowa lens, 2560 × 500 resolution) and analyzed with Ethovision. For analysis, the chamber center segment (100 cm) was used to measure speed and acceleration as in the Open Field test. The software automatically identified when the animal left the start zone and when it reached the end zone (defined by a 10 cm area at the edges of the corridor). All trials were included in the analysis. Data were presented as group average instantaneous speed (traces), individual average speed, and maximal acceleration.

**Ladder test**. This test was used to evaluate skilled locomotion. Mice were placed at the lowest point of an inclining ladder (+5 degrees, 40 cm length, 8 cm wide, 2 cm between rugs, 0.3 cm rug diameter) under low light conditions (30 lux). Before testing, mice were made akinetic with an injection of haloperidol or SCH23390 (control group was given saline) and 40 min later locomotion was evoked by optogenetic activation of PPN neurons (control group tethered but non-stimulated were allowed to freely move). Laser settings were the same as in the Open Field paradigm and recording parameters were the same as in the Bar test. Each mouse did between three and five (ladder crosses) which were video recorded and later processed with TSE Motion Video Analysis Software (v9.2.2).

Two outcome measurements were taken. First, all steps were counted and classified as either functional paw placement (a weight-bearing step on a rung) or misses (a step that missed the rung completely or with initial contact followed by a slip-off with or without replacement). Second, fore-hindlimb coordination was assessed by counting how often the same rung was touched first by the forelimb and then by the ipsilateral hindlimb, a pattern that is usually observed in intact rodents[112,113]. Data were presented as a percentage of total steps or targeting attempts (accumulated over repeated trials) and stick diagrams were plotted with the JMP analysis software using exported x,y joint positions from the TSE Software.

**Obstacle test**. This test was implemented to measure the capacity of mice to ensure a smooth and adaptable locomotor movement in a complex environment. Mice were placed in the corridor (120 cm length, 8 cm width) containing three obstacles: (1) A rotating rod (2 cm diameter, 2.3 cm from floor, baseplate 6 cm length) that represented a dynamic challenge as the rod could freely rotate upon touch. (2) A slalom made of four vertical cylinders (1.7 cm diameter, 10 cm height) spaced in alternating wall sides to create an inner track space of 3 cm (baseplate 15 cm length) through which the mouse needed to perform a zig-zag motion to pass. (3) A staircase (three solid blocks with a climb of 1.5, 3.5, and 4 cm height) containing a gap (2 cm width) followed by a lower platform (2.5 cm height) and a ramp (30° decline) through which the mouse would have to climb and descend (baseplate 25 cm length). Each end of the corridor contained a small external chamber in which unknown mice from the opposing gender were placed to motivate the test subjects to perform more corridor crosses within a trial. A set of three small nose-poke holes allowed the subject to interact with the novel mice. Before testing, mice were allowed to freely explore the apparatus for two rounds of 10 min. Only the first round was performed without social exposure (habituation). Thereafter, two recording sessions were performed. For baseline recordings, subjects were injected with saline 10 min prior to the start and allowed to freely navigate the apparatus for 15 min. Twenty-four hours later, for challenge recording, the same mice were injected with CNO 20 min prior to start (to silence glutamatergic neurons in CnF) and then with SCH23390 5 min later (to induce parkinsonian state). Locomotor output was aided by optogenetic stimulation of glutamatergic neurons in the PPN. Laser settings were the same as in the Open Field paradigm. Laser-ON epochs were set to repeat until the mouse had covered

the same distance as in baseline (max of 20 epochs, 65–75 s intertrial intervals, to equalize sampling from first and second sessions). Recording parameters as in the Corridor test but camera sampling reduced to 30 fps.

For analysis with Ethovision software the chamber center segment (100 cm, containing all obstacles) was used to count the number of crosses, total distance moved, and average speed. Sub-zones were used to measure the average time needed to pass each obstacle. The software automatically identified when the animal entered/left each zone (or sub-zone). All crosses were included in the analysis and preprocessing parameters were kept as in the Open Field test. Data were presented as individual mean values measured during baseline vs challenge session. On challenge day, mice performed the Bar test at the start (to confirm akinetic state) and end (without and with laser support). Eight WT mice (2.5-month old) were used for social exposure.

**Limb Kinematics**. To perform limb kinematics we identified all instances in which test subjects from Cohort 1 were detected near the wall in the Open Field and locomoted parallel to the side-view cameras (<11 cm from the center point to wall). Once locomotor segments were identified videos were exported using ffmpeg (multimedia framework). During this step, video clips were renamed with individual codes to allow for blinded analysis (total of 3.054 video segments). For tracking, we used the markerless pose estimation software DeepLabCut[57] (version 2.2b8). First, we trained the ResNet-50 based neural network (default parameters, Imgaug augmenter type, $10^6$ iterations) to identify points of interest the hindlimb and on the body (snout, ear, tail-base, hip, knee, ankle, hind paw, hind toe tip, and front paw). The training was performed on a set of 580 frames randomly selected from 58 video-clips representatives from each experimental day (training fraction 95%). Next, we validated the network with three shuffles and found the test error was 1.38 pixels (0.09 mm). This trained network was used to analyze all video clips. The data set was filtered (median over 4 bins) and the distance between joints and joint angles were calculated using the DeepLabCut toolbox function "analyze_skeleton.py". For preprocessing we used a p-cutoff of 0.95 to condition the x, y coordinates, and only video segments in which all body parts were identified with >0.99 likelihood were included in further analysis. The average speed of the hind paw was used to determine two events that mark the step cycle: the touchdown (initiating stance phase) and lift-off (initiates swing phase) with a threshold of 5 cm/s (minimum of three consecutive frames for lift off). Identified events allowed the identification of each step cycle and only sequences containing full step cycles were further analyzed. Next, we excluded all cycles which contained features that violated a distance constraint (i.e., when a joint was detected further from other joints than it is anatomically possible). The remaining step cycles were temporally normalized and compiled (overall average of 40.69 s of compiled video segments per mouse/condition with CI 29.62 to 51.77 s). Finally, we reconstructed the sequential angular variations of joints (with a precision of 12.5 ms). Shank travel[58] angle is defined by the vector connecting the hip and knee. Group averages are presented as lines with 95% CI, and amplitude calculations of individual raw values were used for analysis.

**Calcium imaging in the Open Field**. Prior to imaging experiments, mice were habituated to the miniscope mounting procedure and added weight by using a dummy scope (no anesthesia, 4–5 training sessions, 15–30 min/session, performed within 1 week). To record freely moving animals an Inscopix commutator was used. Calcium imaging videos were recorded using Inscopix nVoke acquisition system (v2.0) and software (IDAS, v1.3.1) at 15 Hz and using between 0.5 to 1.2 mW/mm$^2$ LED irradiance (455 ± 8 nm). For each animal, the field of view and focus depth was maintained between the two experimental days (one exception shown in Fig. S2e). Ethovision was used to trigger and control the Inscopix Data acquisition (DAQ) box in synchrony with behavioral video tracking (30 fps, above view). For alignment, TTL signals were sent from the DAQ box for each frame saved, whereas Ethovision continuously sent a TTL barcode at 120 Hz to DAQ (via Noldus USB-IO box). These dual synchrony TTL signals were used as timestamps so that each behavioral tracking frame could be temporally aligned with its nearest calcium imaging video frame (23 ± 5 ms error range, done using JMP software).

During the experimental day, the miniscope was mounted and the animal was placed in its home cage for 5 min. Subsequently, the mouse was transferred to the Open Field (round, 25 cm diameter, with bedding) and the session began. After 1 min delay a 12 min continuous imaging was performed (drug-free). The mouse was then injected with a parkinsonian state-inducing drug and immediately placed back in the Open Field. Following another 5 min delay, four trials consisting of 3 min recording-ON, 3 min-OFF were performed (note: behavioral tracking data recorded continuously). Altogether, this procedure resulted in test rounds of ~42 min/mouse with a total of 24 min fluorescence imaging data per round.

**Calcium imaging data processing**. All calcium imaging movies were processed using the Inscopix Data Processing Software (v1.3.0.2723). Each experiment (day) contained five video files which were concatenated and processed as one single raw video (15 Hz, maximum FOV of 1280 × 800 pixels equivalent to 1070 × 670 μm).

First, raw video files were cropped, and defective pixels were fixed (3 × 3 pixel median filter). Then videos were temporally downsampled (factor 2, interpolation) to reduce file size. Spatial bandpass filtering was used to remove low and high

spatial frequency content from the movies. Next, we ran a motion correction algorithm[114] with frames registered to the reference image (average from min 12 to 24, weight 0.8, provisional low cut-off 0.04). Now videos were spatially downsampled (factor 2, binning) to reduce processing time. The baseline fluorescence value for each pixel $F_{baseline}(x,y)$ was based on pixel fluorescence averaged across all time frames (each frame = $t$, all frames = 24 min recording data). The dF/F value of a pixel at a given time, represented by the formula $F_{pix}'(x,y,t)$, was calculated as a function of the baseline fluorescence value and the raw fluorescence value $F(x,y,t)$ as follows:

$$F_{pix}'(x,y,t) = \frac{F(x,y,t) - F_{baseline}(x,y)}{F_{baseline}(x,y)}(x,y,t) \qquad (1)$$

Where $\forall(x,y,t)$ represents each pixel annotated through all units of time.

Thereafter, regions of interest (ROIs) were segmented. ROIs were first identified by PCA/ICA (combining Principal Component Analysis and Independent Component Analysis for spatial/temporal unmixing) with the number of independent components (ICs) estimated as 15% over the number of cells identified by visual inspection of the movie maximum projection images in ImageJ. ROI refinement was performed with the following steps. (1) The spatial footprints of each detected component were visually inspected to verify for ROI accuracy by superimposing ROIs to manually draw cell sets in ImageJ. Estimated ROIs that did not juxtapose or overlap with manually drawn cell sets were excluded to avoid false-positive [risk excluding less-active cells]. (2) ROIs smaller than 4 pixel[2] or with roundness <0.3 (i.e., circularity), were excluded to avoid instances of neuropil detection. (3) If a cell pair showed the distance between centroids <5 pixels and correlation coefficient between entire drug-free period >0.7 one of the pairs was excluded to avoid the risk of double-counting. (4) Traces that contained >2 components or signal to noise ratio <3 (median amplitude of the trace at event times divided by the median absolute deviation of the trace) were excluded. Approximately 30% of all detected ROIs were excluded from the data set. All remaining components were tagged for further analysis as curated cell sets.

Registration across experimental days was performed with curated cells only and based on a minimum correlation of 0.7[115,116]. For each ROI, a resulting file containing the temporal fluorescence trace [$F_{pix}'(x,y,t)$] was exported to JMP software (v14, SAS Institute Inc.). Within JMP, traces were individually smoothed (exponentially weighted moving average with coefficient 0.25 frames$^{-1}$, fixed interval −8 frames [1.4 s], +17 frames [3.2 s]), and Z-scored ($Z$) with a mean(dF/F) and standard deviation SD(dF/F) calculated across all frames:

$$Z(t) = \frac{dF/F(t) - \text{Mean}(dF/F)}{\text{SD}(dF/F)} \qquad (2)$$

Graphical data is presented as Z-scores for each cell, plotted as a continuum or in bins containing an average value. Heat map contains bins representing the average Z-score within 50 s windows.

**Tissue immunochemistry**. Post-experimentally brains were analyzed to validate the injection site. Mice were deeply sedated with a pentobarbital solution (200 mg/kg) and transcardially perfused with cold Phosphate Buffered Saline (PBS 0.05 M, Gibco) and heparin (10 U/ml) followed by a solution containing 4% paraformaldehyde (PFA, Histolab). The brain was isolated and postfixed for 4 h in 4% PFA. The tissue was kept in cryoprotectant solution until sectioning. For sectioning, we used a vibratome (VT 1200, Leica) at 50–60-µm thickness, and floating sections were collected in sequential order into individual wells (coronal or sagittal cuts as indicated in figures). After 1 h incubation at room temperature with blocking solution (0.5% Triton-X100 in PBS [PBS-T, Sigma Aldrich] and 2–5% donkey or goat serum [Invitrogen]), primary antibodies (anti-DsRed made in rabbit, Takara #632496; anti-GFP made in chicken, Abcam #13970 [both 1:1000]; anti-c-Fos made in rabbit, Cell Signaling #9F6 [1:250]) were applied in solution (PBS-T, 1% serum) together with nuclear counterstaining (NeuroTrace 640/660, Invitrogen #N21483 [1:750]) and incubated overnight, at room temperature, under gentle shaking. Fluorophore-coupled secondary antibodies (all from Invitrogen; Alexa-568 anti-rabbit #A10042 [1:500], Alexa-488 anti-chicken #A11039 [1:750], Alexa-405 anti-rabbit #A31556 [1:500]) were applied after washing (3x in PBS-T, 20 min/step) and incubated for 3–4 h at room temperature. Sections were then washed in PBS and mounted on microscope slides (TOM-11, Matsunami) with antifade preservative medium (prolong diamond, Invitrogen #P36961). Mounting was done in sequential order along the rostrocaudal or mediolateral axis and imaging was performed 24 h after.

Within each mice cohort, all samples were stained simultaneously. To confirm primary and secondary antibody specificity, parallel control staining was performed. For primary antibodies, using anti-GFP in sections with mCherry expression or anti-DSRed in sections with Citrine expression. For secondary antibodies, using samples lacking primary antibody incubation. In our control staining, no unspecific immunolabeling was observed.

**Imaging and mapping**. All animals belonging to the same cohort were imaged in a single round using the same microscopy settings. For overview images. Full sections were scanned using a ZEISS Axio Observer Z1/7 microscope (10x/0.30 M27

EC Plan-Neofluar air-objective, 1 pixel = 0.57 µm, equipped with Colibri 7 light system and Axiocam 702 monocrome camera, 14 bit). In coronal slices, we imaged sections spanning from the caudal border of the substantia nigra pars reticulata (SNR) towards the end of the superior cerebellar peduncle (scp) (from Bregma, in mm: −3.88 to −5.20, [12 sections/mouse, spaced 120 µm apart]). Moreover, specific rostral and caudal segments which are known to receive PPN afferents were imaged (most rostral 0.62, most caudal −7.08 mm). In sagittal slices, we imaged every section from 0.90 to 1.50 mm (lateral from Bregma [10/mouse, continuous]) and extra sections spanning from 0.48 to 2 (1:3 serial). The resulting tile images were then stitched by overlapping (15%) and merged automatically using built-in functions from the Zen software (Zen 3.0 pro). Using the NeuroTrace-stained channel we identified each section in relation to Bregma[117]. Thereafter, sections of each mouse cohort were processed separately.

*Assessment of implant position and viral targeting.* For all implanted mice the optical fiber/lens track itself was used to locate the mediolateral and anteroposterior implant end-position. For coronal sections we did scale rotation on top of the mouse brain atlas[117] panel with Adobe Illustrator (v2020), using NeuroTrace staining as reference. The now superimposed images contained PPN anatomical boundaries defined by lines from the atlas. Thereafter, the viral-staining signal was exposed and mice with injections/implantations outside the target area were excluded from all data sets. Mapping of injection sites (8–9 sections/mouse spanning from Bregma −4.24 to −5.20, coronal) was performed using ImageJ (v1.53J) scripting and batch processing with the following steps. First, for each section, the image containing the NeuroTrace-signal (blinded analysis) was opened and the experimenter placed two square ROIs (1.5 mm side squares) centered over the PPN and/or CnF (one ROI per hemisphere). Thereafter ImageJ analysis steps were automated and followed a sequence of built-in functions (now applied to the channel containing the stained viral signal): Auto threshold, subtract background (rolling ball = 100 pixels), recall ROIs, and perform image crop. Finally, per ROI image, get intensity gray value within each grid of 100 µm$^2$ (shifts over image-making measurements at every x,y-grid position) and export results as x,y-grid intensity values (.csv) for further analysis. Thereafter, exported.csv files were mapped back to their respective Bregma distance, mouse group, and ID to be further processed using JMP software. For each mouse the ROIs corresponding to the same segment were averaged, resulting in a final data file of x,y-grid value (i.e., per segment for each mouse). Individual x,y values were used to create a contour map containing the 90% quantile of highest intensity. Contour maps were superimposed and used to create a congruence heat map line in which the intensity range (color tint) corresponds to the number of animals (within the group) that show infection within borders (Figs. S5a, S6b). For sagittal sections, we adapted the approach from[52] by first selecting 7–8 sections/mouse (mm lateral from Bregma; 2x/0.96, 3x/1.20, and 2x/1.44). Within the Zen software (Zen blue, v3.2), and using the NeuroTrace-signal (blinded analysis), we placed a rectangular ROI (3 × 0.5 mm) at an angled position, starting on SNR center, and ending in alignment with the lobule 2 of the cerebellar vermis. The ROI angle was adjusted to lay over the scp as indicated in Figs. S3g and S9c. We then transferred the ROI to the viral channel and extracted intensity values using batch processing and the "rectangular profile" tool within the Zen software. For each section, the raw intensity gray values were standardized with min = 0 and max = 100. Thereafter, all sections were plotted together to observe the relationship between viral-channel intensity in the rostrocaudal direction across the different mediolateral levels.

*Assessment of c-Fos.* To confirm that eD can induce neuronal activity a cohort of mice was injected unilaterally with eD in the PPN and compared to WT mice. These mice were injected with CNO 90 min prior to perfusion and placed back in their home cage during waiting time. Only a subset of coronal sections (from Bregma, in mm: −4.60 to −4.72, 5–6 sections/mouse, sequential) were stained for both c-Fos and the viral tag mCherry. Analysis was performed using ImageJ software (v1.53J) as follows. For each section, a square ROI (1.2 mm side) was placed over the PPN on each hemisphere using the NeuroTrace channel. Next, the ROI was transferred to the c-Fos channel, and the image was cropped to this region of interest. The x,y location of each c-Fos$^+$ nucleus was extracted using background subtraction (rolling ball, 50 µm), transformed to a grayscale 8-bit image, maxEntropy thresholded from 30 to 255, and then analyzed using the ImageJ function "analyze particles". Data from each ROI (containing x,y nuclear positions) were exported and further processed in JMP software. For each mouse, the average number of c-Fos$^+$ nuclei per section and in each hemisphere was used for graphing. Mapping of c-Fos$^+$ nuclei distribution was made by calculating the average nuclear count per group on grids of 100 µm$^2$.

Higher resolution images of selected brain regions were taken for visualization of fiber tracks or staining co-localization (20x/0.8 M27 air-objective Plan-Apocromat, with Z-stack three to five planes, 1 pixel = 0.312 µm, LSM900 ZEISS confocal microscope, 16 bit). These images were Z-projected using max-intensity.

## Statistics

*Power analyses.* The effect size was unknown beforehand, and the minimum group size was initially defined based on the assumption that the mean of control and

treated groups would need to differ by at least 20% (δ) to be considered a meaningful effect whereas SD would remain within a 10% (σ) margin regardless of treatment. These parameters were selected based on previous publications[64] taking into consideration the distance moved in the Open Field by control animals when handled by the experimenter. With $\alpha = 0.05$ and a power level of 95%, we estimated a minimum group size to be seven mice. Those same parameters with increased SD to 13% (σ) resulted in group sizes of 11 individuals. Thus, we defined that the minimum group size for a behavioral experimental round should be set to seven mice and that the second batch of animals would be generated for the completion of groups aiming at $n = 10$.

*Experimental blinding and exclusion events.* The experimenter was not blind to experimental group allocations, yet all animals were monitored with automated scoring of behaviors. A total of 119 mice were used. In five instances subjects were excluded from a data set based on akinesia-state as defined by the Bar test (their Bar-test score is graphed in supplementary data, Fig. S9e). Furthermore, 12 mice were excluded from all data sets due to injection sites/fiber placement targeting errors.

*Statistical methods.* See the detailed statistical report and critical values on Tables S1–S7 as recommended by guidelines[118]. All data were analyzed with GraphPad Prism (v8.4.1) or JMP analysis software (v14, SAS Institute Inc.). For two-group comparisons, two-tailed Student's *t*-test (paired or unpaired as appropriate, referred to as *t*-test and only cases with paired analysis are indicated in the text) and its nonparametric analogs Wilcoxon's signed-rank (paired, where the critical value W is the sum of signed ranks) or Mann–Whitney *U*-test (unpaired) were used. For three or more groups and for experiments with longitudinal data One-way ANOVA or Two-way ANOVA factorial or by repeated measures (RM) as appropriate were used. Multiple comparisons were done with Dunnett's or Bonferroni. When the assumptions of normality were not met (D'Agostino–Pearson normality test and outlier check by ROUT, Q = 1% [chance of falsely identifying an outlier]), Kruskal–Wallis and Friedman test followed by Dunn's T3 multiple comparison (a test that can handle unequal variances) were used instead. Note that for the Friedman test the reported critical values T and S correspond to the number of treatments and number of subjects, respectively. When assumptions of normality were met, but groups showed unequal variance we used Welch's (W) and/or Brown–Forsythe (B–F) corrections. In such instances, instead of reporting F-critical values, which are less reliable, we report B–F or W. The degrees of freedom (df) reported in tests with correction are adjusted to control the risk of type-1 error and may contain decimal cases. To measure the strength and direction of association between variables we used the Spearman rank-order correlation coefficient (nonparametric, denoted by the symbol $r_s$). Statistical significance in text is reported as exact values (up to four decimal cases can be seen in statistical tables) and in the figures reported as follows: $*p < 0.05$, $**p < 0.01$, $***p < 0.001$, $****p < 0.0001$. Graphical data were presented as mean with either ±SEM or ±SD, min-to-max boxplots (extend from the 25th to 75th percentiles, with whisker reaching the full range of points and the middle line representing the median), violin plots (interquartile ranges), or confidence interval bands (CI 95%), as indicated on legends/y-axis. The exact number of animals/replicates are stated in figure/legends and reported on supplementary statistical tables (S1–S7).

**Reporting summary**. Further information on research design is available in the Nature Research Reporting Summary linked to this article.

## Data availability
The data generated in this study are available in the source data provided with this paper. Raw data files (i.e., videos and images) are not permanently deposited in an open access depository, yet example videos are available from the corresponding author upon reasonable request. Source data are provided with this paper.

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

## Acknowledgements

This research was supported by the European Research Council under the European Union's Horizon 2020 Research and innovation program (grant agreement no. REP-SCI-693038), the Lundbeck Foundation (R276-2018-183), and the Novo Nordisk Foundation Laureate Program (NNF15OC0016016), and The Swedish Research Council. We thank members of the Kiehn lab for their comments on the manuscript. We acknowledge the Core Facility for Integrated Microscopy, Faculty of Health and Medical Sciences, University of Copenhagen.

## Author contributions

OK conceptualized and supervised all aspects of the study. DM and OK planned and designed experiments. DM performed experiments, analyzed the data. OK and DM wrote the manuscript.

## Competing interests

The authors declare no competing interests.

## Additional information



