## [Peer Review File · Nature Communications]

Reviewers' Comments:

Reviewer #1:

Remarks to the Author:

The study of Masini and Kiehn provides interesting *in vivo* evidence that glutamatergic neurons in the caudal pedunculopontine nucleus (PPN) increase locomotor activity when activated in Vglut2-cre animals injected with a cre-dependent AAV coding for a channelrhodopsin (optogenetics) or for an excitatory DREADD (chemogenetics). Such increase in locomotor activity was seen in mice showing made akinetic following application of a typical antipsychotic drug, the D2 antagonist haloperidol (0.5 mg/kg ip); and in mice made akinetic following application of a D1 antagonist, SCH 23390 (0.25 mg/kg ip). In contrast, optogenetic and chemogenetic stimulation of GABAergic neurons in the caudal PPN of VGAT-cre animals only increased locomotor activity in mice made akinetic by the D2 antagonist, but not in mice made akinetic by the D1 antagonist. Such dopaminergic-receptor dependent effect was not observed in the bar test, where the animal's forelimbs lie on an elevated bar. Mice made akinetic by the D1 or D2 antagonist showed an increased latency to start moving away from the bar. Optogenetic and chemogenetic stimulation of PPN glutamatergic or GABAergic neurons both decreased the latency to move. The authors interpret their results in the context of Parkinson's disease (PD) and propose that the caudal glutamatergic PPN neurons are a more reliable target than the GABAergic ones to alleviate the locomotor deficits induced by insufficient dopaminergic signaling in PD.

This highly interesting study comes from an excellent laboratory working on the neural control of locomotion. The authors take advantage of powerful genetic tools to activate different types of cell types in the Mesencephalic Locomotor Region (MLR). Recently, the Kiehn lab has provided evidence that PPN glutamatergic neurons evoke locomotion, that GABAergic neurons stop locomotion, and that cholinergic neurons decrease locomotion (Caggiano et al. Nature 2018 doi: 10.1038/nature25448). Here they test the relevance of targeting some of these cell types in pathological conditions. The present study is highly relevant to the motor control and clinical neuroscience communities, because the MLR is an exploratory target in PD and data is needed to identify the best target to improve locomotor function.

I only have a series of minor comments that would need to be addressed to strengthen the paper and to better understand the unexpected observation that optogenetic and chemogenetic stimulations of GABAergic PPN neurons evoked locomotion (relative to the inhibitory effect reported by Caggiano et al. Nature 2018 doi: 10.1038/nature25448 and Roseberry et al. Cell 2016 doi: 10.1016/j.cell.2015.12.037). Additional experiments are requested. I think that the authors could perform at least some of those. The ones that will not be done should at least be mentioned in the discussion.

MINOR COMMENTS.

1. Specify of the transgene expression. Could the authors provide anatomical evidence of the specificity of the expression of their transgenes following AAV injections? The authors could illustrate and quantify whether the infected cells are neurons, and whether such neurons are Vglut2- or VGAT-positive using RNA or immunofluorescence experiments.
2. Effect of chemogenetic activation on the targeted cells. To my knowledge, this is the first series of experiments using excitatory chemogenetics to activate MLR cells. I think that some control data could be added. Maybe the authors already have those data. In Fig. 5, the authors show that D1 and D2 antagonism reduce the calcium activity in PPN glutamatergic neurons. Do the authors have the calcium imaging data of the same animals when CNO was administered? If yes, it could be used to determine whether CNO increased the frequency of calcium transients up to pre-D1 or D2 antagonist levels. The same recordings could be examined in VGAT-eD mice, if possible. Alternatively, the authors could use other approaches, if possible (patch on slice, *in vivo* electrophysiology, calcium imaging *in vitro*, c-fos).
3. Is the locomotion evoked normally coordinated? Could the authors illustrate whether locomotion evoked by their optogenetic and chemogenetic stimulations is normally coordinated? The authors could use limb kinematics or the footfall patterns. Such data would be useful to compare the locomotor patterns evoked by glutamatergic vs. GABAergic PPN neurons. This would also be important in the context of the clinical relevance.
4. How to explain the pro-locomotor effect of PPN VGAT-positive neurons?
 - 4.1. Unexpectedly, the authors show that optogenetic (Fig. S6) and chemogenetic (Fig. 7) stimulation of PPN GABAergic neurons (VGAT-positive) increase locomotor activity. This finding

contrasts with Caggiano et al. Nature 2018 doi: 10.1038/nature25448 and Roseberry et al. Cell 2016 doi: 10.1016/j.cell.2015.12.037, who showed that optogenetic activation of GABAergic PPN neurons decreases locomotor activity. Therefore, I was surprised that the authors did not underline in the result section that this was unexpected. I think this should be more clearly discussed in a dedicated paragraph about GABAergic neurons.

4.2. To better understand the respective roles of glutamatergic and GABAergic PPN neurons in the control of locomotion, it would be interesting to examine, if possible, the effect of D1 and D2 antagonism on PPN VGAT-positive neuron activity recorded with a GRIN lens. The authors did the same experiment for PPN glutamatergic neurons and showed that their calcium activity decreased after D1 or D2 antagonism (Fig. 5).

5. The number of observations and the post hoc tests after the ANOVAs

The statistics are well reported in the paper. I only have two small comments.

5.1. The number of observations is not always clear in the text. Could the authors please define the number of trials or at least the number of animals tested when reporting their results in the text?

5.2. I had some troubles finding the result of post hoc tests in some cases. For the ANOVAs, when the ANOVA is not significant, it is OK of course not to specify the results of the post hoc tests. However, when the ANOVA is significant, I would expect a value for each post hoc test between groups, time point by time point. Either more stars should be drawn in the graph, or more information should be given in the legend. Let's take for instance Figure 1B, there is a "ns" for t=5 min, and then a single "****" for all the other time points. Does this mean that the post hoc test was applied to each time point between the two groups, and that the significance in "ns" for 5 min and then "****" for all the other time points (10 min, 15 min50 min)? Two other examples. In Fig. 2c and Fig. 3a. Is there a "ns" just for t=20 min, but a "*" difference at time = 10, 30, 40 and 50 min? Same for Fig. 6c, b, e... It would be good to specify to which time points the significance applies.

DETAILS

Title

"Targeted activation of midbrain neurons restores locomotor function in mouse models of Parkinson's disease"

Please consider replacing "in mouse models of Parkinson's disease" with "in mice made akinetic by D1 or D2 antagonism".

Running title

"Pedunculopontine activation in Parkinson's disease!"

Please consider replacing "in Parkinson's disease" with "in akinetic mice".

Abstract

Line 5-6 "in two pharmacological PD mouse models"

Please consider replacing with "in mice made akinetic by D1 or D2 antagonism".

Line 7-8 "GABAergic PPN population is not sufficient to support motor recovery"

Please consider modifying this sentence to be closer to your results, with something like "glutamatergic cells evoked sustained locomotor activity in mice made akinetic by D1 or D2 antagonists, whereas GABAergic cells only evoked sustained locomotor activity in mice made akinetic by D2 antagonism". This can be adjusted of course.

Introduction

Line 13-20. I understand that the authors highlight that abnormal BG activity is considered as an important part of the PD pathophysiology. However, the cause is the degeneration of DA cells. It should be stated in the first paragraph, even if the approach of the study does not rely on DA cell death.

Line 26. "increases the relative contribution to D2-MSMNs"

Should the authors also mention the decreased inputs to D1-MSN in PD? This might be useful to increase the relevance of the decreased striatal calcium activity they recorded in mice made akinetic by D1/D2 antagonism in Figure 4 and Figure S2?

Line 47-48 "two distinct PD mouse models"

Please consider replacing with "mice made akinetic by systemic administration of D1 or D2 antagonists".

Line 51-52 "self-paced locomotion"

Maybe remove "self-paced". How do we know that such locomotion is self-paced?

Line 53-54 "PPN caudal GABAergic neurons could not restore locomotor capabilities".

To avoid confusion in the mind of the reader, this should be modified because it contradicts the results of the present study, which shows that their activation increases locomotion in control mice (chemogenetics Fig. 7; optogenetics Fig S6) and in mice made akinetic by D2 antagonism (chemogenetics Fig. 7; optogenetics Fig S6). However, it is true that their activation does not induce locomotion in mice made akinetic by D1 antagonism (chemogenetics Fig. 7; optogenetics Fig S6).

Line 64-65 "known innervation targets"

Please specify those targets.

Line 70 "locomotion"

Please define what you quantified here as locomotion.

Line 71 See minor comment 5 about the stats.

Figure 1 Legend

Line 88. Please define what appears in white/grey: is it the neuronal staining (black is no signal, right?). Please specify the neuronal marker. Also, please see comment 1. A higher magnification of this should be added maybe as a supplementary figure to see which neurons were infected. I cannot see any cell on this picture.

Line 101 "PD phenotype" [...] "with dopamine receptor antagonist haloperidol or SCH 23390".

The specification of the pharmacological tools used should come much earlier in the text.

Line 106 "halo 0.5 mg/kg"

Could the authors please add a reference or two to justify this dose?

Line 116 "vs med of 20s"

Please explain, I do not understand.

Line 120 "profile max difference" [...] "35% for Sham" "5% for Vglut2_eD"

Could the authors please explain with sentences?

Figure 2.

Panel A

Above "Injection 1", could the authors add "sal/halo" and above Injection 2, "sal/CNO" to improve clarity?

Line 149 "SCH 23390 (0.25 mg/kg)"

Could the authors please add a reference or two to justify this dose?

Line 179

Please replace "basal ganglia" with BG if already abbreviated

Line 185-186 "PD1-4, 3 min REC/3min OFF"

Could the authors please make sentences to make this clear please?

Line 189 "high calcium dynamics that declined "

Could the authors please add a statistical analysis to support this?

Line 190 "81%"

Could the authors please add a SD or SEM?

Line 191 "78.9%"

Could the authors please add a SD or SEM?

Figure 4.

Line 199-200 Panel d.

Could the authors please add a statistical analysis for the decrease in locomotor activity?

Line 201-202 Panel e. "maximum deviation from average fluorescence through the 24 min recording)."

Could the authors please define what this means? Maybe in the methods.

Line 207-208 Panel g.

Could the authors please add a statistical analysis for the decrease in calcium activity?

Line 214 "72.9% and 80%"

Could the authors please add a SD or SEM?

Line 214 "reduced activity"

As for the legend, could the authors please add a statistical analysis for the decrease in calcium activity?

Line 215 "with the remaining neurons increasing their activity"

Could the authors please add a statistical analysis for the increase in calcium activity?

Line 216 "the data shows"

Please replace with "show"

Fig 5 panel b

Do the authors have the same recordings under CNO in vglut2-eD to show the increase in calcium transients?

Fig 5 panels c and D

Could the authors please add a statistical analysis for the decrease and increases in calcium activity? (Cf comments on Fig 4)

Line 239

Please specify the duration of the epochs

Line 242 "2-3.5 mW"

How was this measured?

Line 243 Please delete "random"

Line 246 "laser reliability"

Please define.

Line 251

Please define T and S.

Line 252-253.

This is not a sentence. Could the authors please rephrase?

Line 258 "light"

Please specify wavelength

Line 263 Please define W

Line 273 "postural corrections"

Please make clear whether this test allows you to test locomotion initiation or postural corrections. If the authors were looking at postural corrections, please define what the postural correction is.

Line 274 "median of 20s"
Please explain.

Line 292.
Please specify \pm SD or SEM

Line 293-294 Please specify \pm SD or SEM

Line 299 "triad"
Please check whether this is the good word.

Line 304 "cholinergic represent the minority"
Please specify the proportion.

Line 305-307.
Please specify clearly that their optogenetic activation decreases locomotion in Caggiano et al. 2018 and Roseberry et al. 2016. Please make clear that their tonic excitation with chemogenetics was not tested.

Line 311.
Please add "Unexpectedly" at the beginning of this sentence. It is important for the reader to understand that this is a surprise (at least, to me, it was)

Line 313. "129.2 \pm 9.8"
Please specify the unit

Line 319 "The difference in distance travelled vs speed profile between Sham, Vgat_eD and Vglut2_eD mice..."
Can the authors please rephrase this sentence?

Line 322 "stop bouts"
Could the authors please define those here?

Line 324 "Brown-Forsythe and Welch ANOVA"
Could the authors please define in the methods when this one was used?

Line 325
Please define "B-F" and "W"

Line 323-324 "Yet the average stop bout duration of VGAT-ED was nearly half of that observed in sham and Vglut2 -ED"
This should be discussed, see comment 4.

Figure 7.
Panel e. Lines 344-347. "as identified by the tracking software" sounds mysterious. Could the authors please define the stop bouts because it is also weird to see the stop in the middle of the vglutED recording not to be counted as a stop, despite the fact that it looks longer than the ones counted in the vgat-ed recording.

Line 350
Please define "T3"

Line 368. I would recommend rephrasing this sentence to highlight more clearly the specific difference with the D2 antagonism. Moreover "CNO treatment" could mean "by itself on Sham animals". Please specify in VGAT-ED mice.

Line 391-393.

This is where the authors are describing a result that contrasts with Caggiano et al 2018 and Roseberry et al. 2016. This should be clearly highlighted. It is not a problem to say it, it is interesting that the result is not the same. Stimulation of MLR cholinergic neurons also led to some surprises from study to study.

Line 393-396.

Could the authors please specify whether there was a wavelength control with 593 nm light for these results?

Line 401 "reliability"

Please define

Line 403 "light"

Please specify wavelength

Line 407-408. Please specify whether this was observed in presence of a D1 or D2 antagonist. Please specify whether a 593 nm laser control was done for these observations.

Line 409-410. Please specify that this is unexpected relative to Caggiano et al. 2018, Roseberry et al. 2016.

Line 411 "IN accord"

Please rephrase.

Line 411-412

Please rephrase to make the sentence more faithful relative to the experiments. The sentence should be more balanced. It should state the absence of effect of VGAT neuron activation when PD was induced with a D1 antagonist, but should mention that the activation of VGAT neurons counteracted the motor deficits in the bar test evoked by D1 or D2 antagonists, and counteracted the motor impairments evoked in the open field by a D2 antagonist.

Discussion

Line 425-427.

I do not understand why the authors increase the contrast between the effects of the stimulation of vglut and vgat neurons, considering that the only difference is relative to D1 evoked akinesia specifically in the open field. This is also surprising relative to the end of the discussion, where the authors say that stimulating the two cell types should not be a problem. Maybe rephrase and homogenise in the whole paper.

Line 432" Simulate the neurochemistry"

Please replace with "simulate part of the neurochemistry"

Line 441 "further supports"

Please replace with support

Line 442 "pedunculopontine nucleus"

Please replace with PPN

Line 470-471

Could the authors please provide single cell controls? See comment 2.

Line 475. "We suggest"

Space missing before We

Line 480" can recover motor function without causing unusual behavior"

Please provide controls showing that the optogenetic and chemogenetic stimulation of vglut or vgat neurons induced normal locomotion (see minor comment 3).

Line 483 to 506. Please name a full dedicated section on GABAergic neurons.
Please highlight the difference with Caggiano et al. 2018 and Roseberry et al. 2016.
Please explain how you could interpret these three studies as a whole to make sense of the contradiction. If the authors performed additional experiments as suggested (see minor comments) , please discuss the new results here.

Line 489-490 "previously not reported"
Please expand relative to previous results of Caggiano and Roseberry.

Line 492-493. "was insufficient"
Again, I do not understand why the authors increase the contrast between the effects of the stimulation of vglut and vgat neurons, considering that the only difference is relative to D1 evoked akinesia specifically in the open field. Please rephrase. See also other comments relative to this point.

Line 496 This statement contradicts the data in Fig S6 c, d, e, where optogenetic activation of PPN gabaergic neurons improves locomotion in the open field. Please rephrase.

Line 496-500. If new data is added, this section should be modified to interpret the new data.

Line 506. Could the authors please rephrase to specify the sensory inputs that the authors are mentioning here?

Line 508-509. This sentence is increasing the contrast between vglut2 and vgat neurons but I do not feel it is justified. Please see the other comments relative to this point and maybe rephrase to be more specific.

Line 534-538.
This interpretation sounds a bit strange. The point of the unreliability of the clinical studies might specifically be related to the fact that DBS does not segregate glutamatergic from GABAergic neurons. The supplementary experiments suggested in the minor comments could help deciding how to phrase this key point.

Supplementary figure

Fig S1. I do not get why the insets in a are not in color similar to the neighbouring panels?

Panel C please add the open field raw data as for the panel b. In Fig S5b, open field data is provided as in panel S5a.

Higher magnifications are needed to illustrate that neurons are labelled. See comment 1 relative to the anatomical controls.

Fig S2

Panel c please add statistical analysis for the locomotor deficits

Panel d Please define SD (dF/F)

Line 913 "Paxinos"

Please be more precise on this reference

Line 914 Please define "PCA/ICA"

Figure S3

Panel A, D, E, F Please give a name and a unit to the Y axis here and in similar figures elsewhere

Panel b. Please specify wavelength

Line 937 "before and after saline"

Should saline be replaced with "drug"?'`

Panel f

Please define CnF, PrCnF, MiTg in the legend. Please check similar figures.

Fig S5. See comment 1. Magnification is needed to see whether the transgenes were expressed in

neurons, in vgat-positive or vglut-positive neurons.

Line 970 Please be more precise on this reference here and elsewhere in the text ("Brain atlas of Paxinos" at least)

Fig S6.

Panel E. I am not sure this panel is needed. Maybe consider removing it.

Figure S7

Line 1005 Please specify wavelength

Line 1029

Please correct SCH 23390

Line 1039

Please correct SCH 23390

Methods

Line 579. GRIN LENS

Have the authors evaluated whether the implantation had any effect on locomotor activity? (speed, but also pattern, please see comment 3). Please specify. The hole is spectacular in the brain.

Line 601. Could the authors specify how stable was the field of view using the GRIN LENS during a single day and from one day to another? This is interesting to know.

Line 619-622.

Why WT mice were not injected with the virus for optogenetics or chemogenetics, but with other viruses?

Line 661." The accuracy of this definition was validated by a ground proof scoring of a 2 min video".

Could the authors please rephrase?

Line 663 "strange ` before stops"

Line 740-741. Is there a reference to justify the exclusions?

Line 474 Duration individually tailored

Could the authors please specify the range?

Line 774

Motion correction algorithm

Could the authors please define and where to find it? This could be interesting to know.

Line 785-786

Could the authors please precisely define the exclusion criteria.

Line 824 "ROUT, Q= 1%"

Could the authors please explain ?

Line 836-843. Could the authors please specify the host species for each antibody?

Reviewer #2:

Remarks to the Author:

In this manuscript, Masini and Kiehn examine the role of the caudal pedunculopontine nucleus (PPN) in movement, specifically whether its activation can improve motor function in drug-induced parkinsonian mice. Using a combination of approaches including chemo- and opto- genetic approaches of either glutamatergic or gabaergic neurons in this region, they conclude that, in particular, activation of excitatory neurons in this region can improve motor function.

I will say from the outset that this is an impressive piece of work (and I don't just mean it's length, I mean it's quality too!), and is beautifully illustrated. I have no doubt that it will be of great interest to the community, in particular those involved in movement neuroscience from either a fundamental or clinical viewpoint.

I think that some straightforward modifications would strengthen the manuscript, and accordingly suggest the following. In my view, no further experiments are needed to convince me of the results.

1. Deep brain stimulation. The authors take a DBS angle. While this is worthwhile in the Discussion, there are so many issues with clinical DBS that it's barely worth mentioning in the Introduction.... But okay, if that's what they want. So I will focus on the Discussion: in DBS, a large diameter (>1mm) electrode with cylindrical surfaces of 1.5 mm are placed... somewhere. In the case of PPN targeting, the electrode is placed in the dorsolateral mesopontine tegmentum, so they are close to the PPN (and injuring it), but lots of variability. Then a whole bunch of current is passed through these large targets, and then the current ... does what? Who knows? Excites axons? Produces depolarising block? Switches neuronal state/responsiveness? The fact is, we don't know despite all the poorly-supported papers about low and high frequency stimulation that have been published. DBS is crude at best. The STN is more homogeneous and well defined than the PPN, so it's an easier target. The PPN? In fact, the point of this paper is that other therapies – more targeted – are needed. So line 529 “the prime target for therapy in humans” rather than “for DBS.” But the point is, this terrific manuscript defines that PPN DBS is fraught with even more problems than we thought, and new therapies are needed. Electrical therapies? Seems unlikely to me. (I would suggest modifying the last sentence of the Intro, for example, as well as that bit in the Discussion.) In short: no need to say we need better DBS... we need more targeted therapies!

2. PD mouse models. The authors use drugs to induce parkinsonism in mice. That is fine as there is no really good rodent model of PD (that I know about), but the models presented here are not models of PD and the distinction should be made throughout the paper. (Paragraph at 431 could be improved.) Other than MPTP-induced PD in humans, PD is a relatively slow neurodegenerative process, during which neurons in nuclei such as the PPN can adapt. In fact, we don't even know that we have PD until at least 1/3 of our nigral DAergic neurons are gone. We are homeostatic beasts. So I would urge caution throughout about calling this a PD model – it's a model of parkinsonism. The degree to which activating these neurons will benefit a slowly developing disease is not known – and does not detract from the novelty of this manuscript. But the authors should discuss this in that paragraph.

3. Having said all that, what's really call is the differential result in locomotion and the bar test. PD presents differently in different people, with different proportions of the cardinal symptoms/signs. Here, the authors report differential effects (with Gaba expts) on bar test and locomotion. That's really cool, and could be discussed – different symptoms/signs have different circuits.

4. Control mice. In many of the experiments, the control mice (Sham, e.g. Fig 1D, Fig 2D, Fig 3B) are running faster (quite fast, in fact) than the experimental mice at the outset. This is a bit disturbing, as it implies the groups are not equivalent. Does their reduced activity later on result from them being tired out? Why are the groups not equivalent? Do the viral vectors damage some neurons in the PPN for example?

5. Statistics. There a lot of statistics in here, and they seem appropriate although I am far from expert. However, the use of the SEM is not appropriate. The SEM may be appropriate when one is trying to quantify experimental error. But in biology, we're interested in biological variability. Hence, a measure of variance – such as standard deviation – should be used. There are a few instances in the ms where the authors switch to sd, but it's not clear why. I would strongly suggest completely eliminating SEM throughout. (Even better than showing SD would be to show all data points, but I recognise that this may clutter some of the graphs.)

6. Calcium imaging plots – these are not showing deltaF/F but rather the z-scores. The labeling and legends can be changed to indicate this. Also, it's not clear to me how the baseline was determined.

7. Experimental unit. Although the authors state that ARRIVE guidelines were followed, they are not always clear. This is especially true when it comes to the experimental unit. For example, in the calcium imaging experiments, there are n mice. But then there are about 11 neurons recorded per day in these mice. It's not clear how many days they were then recorded, and how many of these neurons were the same day after day. This creates confusion. For example, the authors state that about 80% of neurons showed reduced activity – but were these evenly spread across mice, or were some mice more “affected” than others? This could be analysed, and in fact Fig 4F and 5C could be divided into mice. And if a cell recorded more than once, that could be indicated as well.

Really minor:

1. Line 2: a neuron isn't a “who”, even if you develop a close relationship to it during your experiments.
2. Lines 29-30, “These findings suggest...” I do not see how this sentence follows from the preceding sentences.
3. Scale bar in Fig 5b not defined.
4. Line 243 – what does random mean? How were these intervals randomised? And why?
5. Figs 2B, 3B, 8B, 8E – the *'s should be defined as in reference to the WTsal group. I now see some light gray lines, but indicating in legend might be helpful.
6. Line 445 – inbuilt???
7. Line 461: “experimental confounders” not clear in this sentence.. a confounder affects both the dependent and independent variables and could thus lead to the appearance of an association that's not there. I don't think this is what the authors mean.
8. 517 – tremor treatment is Vim – ventral intermediate nucleus of thalamus
9. 518 – “with some success” downplays the success of DBS, which has changed treatment and lives of patients with PD. It is terrific success. But not all symptoms and signs are treated, only the DOPA-responsive ones.
10. Line 537 – “activation” instead of “stimulation” as DBS sometimes inactivates

Methods:

1. 555-556 – when in this diurnal cycle were experiments done?
2. 580 – source of GRIN lenses?
3. 647 – confirm frame rate only 25 fps, and indicate camera type and source
4. 748 – what are the details of the microscope?
5. 779 – what is the last term in the equation? Should be defined.

Rob Brownstone

Reviewer #3:

Remarks to the Author:

This study aims to identify the role of glutamatergic neurons of the pedunculopontine nucleus (PPN) in reversing the motor impairment produced by acute dopamine depletion. The authors accurately reflect on the controversial benefits of deep brain stimulation in the PPN of Parkinson's disease patients and propose that the neurochemical heterogeneity of this structure is likely playing a role in the inconsistency of the results in the clinics. While this idea has been around for some time, it had not been experimentally tested until now. Following a recent publication by the same group, where they show that optogenetic activation of PPN glutamatergic neurons increase exploratory locomotion (although see comments below), in this study the authors use this previous knowledge to manipulate the activity of these same neurons and counteract the decrease in motor activity that follows the administration of D1 and D2 dopamine receptor antagonists. The results show that the activation of PPN glutamatergic neurons by means of chemogenetics or optogenetics successfully compensates for the motor impairment. The authors also tested the same manipulation on PPN GABAergic neurons and observed a much more limited effect. In neither case (glutamatergic or GABAergic) is clear in the manuscript the connectivity that would underlie such effects, or lack of effects. This study uses a wide array of techniques and an adequate use of

controls. Overall, it is a carefully executed work addressing important and timely questions. However, I have a series of concerns regarding the interpretation of the data and I am skeptical about the advancement that these data represent for PD research or basal ganglia function.

Major concerns:

1) Acute pharmacological dopamine depletion is not a model of Parkinson's disease. Haloperidol is widely used to treat a variety of conditions including mood disorders and aggression. I am sure that the authors do not consider a dose of haloperidol as a temporary parkinsonian state. PD is much more beyond dopamine depletion, it involves a series of chronic compensatory changes that involve alterations in the firing properties of neurons across the basal ganglia circuit and up/downregulation of a variety of receptors. The authors should interpret their results accordingly and rewrite their manuscript focusing on the acute dopamine depletion rather than presenting it as a model of PD.

2) Another important concern relates to the effect of stimulating PPN neurons. In their previous publication on the MLR, the authors showed that stimulation of the cuneiform nucleus (a structure that is adjacent to the PPN in the most caudal region) produced a strong effect on locomotion, whereas stimulation of the PPN produced a milder effect. In the same paper, the authors showed that PPN and CnF are interconnected through glutamatergic connections. Their results, however, contrast with other publications showing either no effect or an opposite effect on locomotion following the stimulation of PPN glutamatergic neurons (i.e. inhibition of motor activity; see Kroeger et al., 2017, J Neurosci; Josset et al., 2018 Curr Biol; Dautan et al., 2020, Biorxiv). In this context, it is concerning to see the spread of the transduction into the CnF in Figure S1-a5. From the evidence provided it is then possible that optogenetic or chemogenetic stimulation of the PPN also activated CnF neurons, which produce a robust effect on locomotion. Another possibility is that, given the connectivity between PPN and CnF, activation of PPN neurons recruit CnF neurons through excitatory synaptic transmission. In line with this, the long latencies for movement induction reported for PPN stimulation (~ 1.5 s) fit with the idea that PPN neurons are recruiting other circuit components to produce increased locomotion. To avoid such ambiguity, authors should provide convincing evidence that the observed effects are not mediated by CnF activation.

3) Dopamine depletion was achieved by the systemic administration of haloperidol and SCH23390. It is therefore expected that dopamine transmission will be blocked across all brain regions. One of such regions is the PPN, which receives direct projections from dopamine neurons of the substantia nigra pars compacta (Ryczko et al., 2016, PNAS) and the zona incerta (Sharma et al., 2018, Sci Rep). However, the interpretation of the data given by the authors is exclusively based on the effects over the direct and indirect striatal pathways. To be able to maintain these conclusions, the authors need to provide evidence that the effects of dopamine antagonists on PPN activity are not a consequence of direct binding on PPN neurons.

4) The manuscript fails to cite previous studies that have made important contributions to research on the role of the PPN in PD. While there are plenty of these studies to include here, one of them in particular stands out for its relevance to the present data. In a 2015 publication, Gut and Winn showed that DBS across different regions of the PPN in the 6-hydroxydopamine model of nigrostriatal degeneration has different consequences for motor activity. They report that DBS in the caudal PPN improved different gait parameters. Authors should include some of these references in their Introduction and/or Discussion to better reflect the current knowledge in the field.

5) The rationale for targeting the caudal PPN (as opposed to the rostral PPN or the whole PPN) is missing. Please include this in the Introduction.

6) I find the following statement particularly problematic: the "caudal PPN glutamatergic subpopulation should be the prime target for DBS in humans" (lines 528-529). The data presented in this study does not address the complexity of PD. Making such a statement is likely to mislead clinical approaches where a rigorous assessment of targets is necessary before any interventions in PD patients takes place.

Minor comments:

Fig. 1b: For how long does the increased locomotor effect last?

Fig. 2d: What do the 2 sets of asterisks in the speed range from 5-10 show?

Fig. 5: The title does not reflect the data, not all glutamatergic neurons decreased their activity.

Fig. 5c: The effects of D1/D2 antagonists on the PPN seem to depend on the previous level of activity of the neuron (i.e. active neurons decrease their activity whereas inactive neurons increase their activity). Is this dependent on the ongoing motor behavior? Could some of these neurons be in the CnF?

Fig. S1b-c: Data should be normalized to account for the different recording lengths.

L2: "is controlling" should read "controls"

L71: How did the authors get 6.059 degrees of freedom for a t-test?

L146-147: "raising the possibility that the direct pathway has unique access to the PPN" Are the authors suggesting that direct pathway MSNs synapse onto PPN neurons? Moreover, given that both antagonists had the same effect on PPN neurons, this statement seems unsupported.

We would like to thank all the reviewers. We understand the time and effort you placed on reviewing this study. The feedback given is greatly appreciated and your suggestions helped strengthen the work. Below you find a point-to-point answer to all questions raised, including outline of additional experiments and analyses performed. The major changes and additions to the manuscript have been marked in yellow.

Reviewer #1 (R#1)

The study of Masini and Kiehn provides interesting *in vivo* evidence that glutamatergic neurons in the caudal pedunculopontine nucleus (PPN) increase locomotor activity when activated in Vglut2-cre animals injected with a cre-dependent AAV coding for a channelrhodopsin (optogenetics) or for an excitatory DREADD (chemogenetics). Such increase in locomotor activity was seen in mice made akinetic following application of a typical antipsychotic drug, the D2 antagonist haloperidol (0.5 mg/kg ip); and in mice made akinetic following application of a D1 antagonist, SCH 23390 (0.25 mg/kg ip). In contrast, optogenetic and chemogenetic stimulation of GABAergic neurons in the caudal PPN of VGAT-cre animals only increased locomotor activity in mice made akinetic by the D2 antagonist, but not in mice made akinetic by the D1 antagonist. Such dopaminergic-receptor dependent effect was not observed in the bar test, where the animal's forelimbs lie on an elevated bar. Mice made akinetic by the D1 or D2 antagonist showed an increased latency to start moving away from the bar. Optogenetic and chemogenetic stimulation of PPN glutamatergic or GABAergic neurons both decreased the latency to move. The authors interpret their results in the context of Parkinson's disease (PD) and propose that the caudal glutamatergic PPN neurons are a more reliable target than the GABAergic ones to alleviate the locomotor deficits induced by insufficient dopaminergic signaling in PD.

This highly interesting study comes from an excellent laboratory working on the neural control of locomotion. The authors take advantage of powerful genetic tools to activate different types of cell types in the Mesencephalic Locomotor Region (MLR). Recently, the Kiehn lab has provided evidence that PPN glutamatergic neurons evoke locomotion, that GABAergic neurons stop locomotion, and that cholinergic neurons decrease locomotion (Caggiano et al. Nature 2018). Here they test the relevance of targeting some of these cell types in pathological conditions. The present study is highly relevant to the motor control and clinical neuroscience communities, because the MLR is an exploratory target in PD and data is needed to identify the best target to improve locomotor function.

I only have a series of minor comments that would need to be addressed to strengthen the paper and to better understand the unexpected observation that optogenetic and chemogenetic stimulations of GABAergic PPN neurons evoked locomotion (relative to the inhibitory effect reported by Caggiano et al. Nature 2018 and Roseberry et al. Cell 2016). Additional experiments are requested. I think that the authors could perform at least some of those. The ones that will not be done should at least be mentioned in the discussion.

We note that the reviewer suggested additional experiments and that we could do at least some of them. We have followed this route and indeed included a substantial amount of new data as outlined below. We complemented those new results by adding discussion points to the text.

MINOR COMMENTS

- 1. Specify of the transgene expression.** Could the authors provide anatomical evidence of the specificity of the expression of their transgenes following AAV injections? The authors could illustrate and quantify whether the infected cells are neurons, and whether such neurons are Vglut2- or VGAT-positive using RNA or immunofluorescence experiments.

This is valid comment, and we welcome the opportunity to better present the mouse lines used in this paper.

Regarding Vglut2-cre specificity: The BAC-Vglut2Cre mouse line used was produced in Kiehn lab (Borgius ... Kiehn. BMC 2010) and has been used in multiple studies from our lab and others (*e.g.*, Nature 2013; PNAS 2013; J. Neuroscience 2014; Nature 2018; Nature Neuroscience 2020). It shows specific expression of Cre in Vglut2⁺ cells, without ectopic expression in inhibitory neurons (GABAergic or glycinergic). This pattern is described within multiple brain structures such as thalamus, hypothalamus, superior colliculi, inferior colliculi and deep cerebellar nuclei, together with nuclei in the midbrain and hindbrain all the way to the spinal cord (Borgius et al. J. Neuroscience 2014). We used this mouse line for our previous MLR study and showed that Cre expression within the MLR is observed in neurons containing the Vglut2-mRNA and Cre expression in this line is indeed absent in the local Chat⁺ subpopulation (Fig.1a-c of Nature 2018 - Caggiano... Masini,...Kiehn).

Regarding Vgat-cre specificity: This BAC-VgatCre mouse line has been directly assessed for specificity in comparison with the *Vglut2^{cre}* line using two complementary approaches in a previous publication (supplementary Fig1 of Haaglund... Kiehn, PNAS 2013). First, with *in situ* hybridization for Vgat or Vglut2 mRNA in *Vgat^{cre}* mice crossed with the reporter line R26-YFP we confirmed that only in the *Vgat^{cre}* mouse line Vgat-mRNA is found within YFP⁺ neurons. Second, with immunohistochemistry, we showed that 98% of the Cre-expressing cells were also positive for either GABA or glycine as revealed in *Vgat^{cre}* crossed with GAD67-GFP mice.

Anatomical evidence: The projection patterns of Vglut and Vgat neurons in the PPN have been previously described and is in line with the patterns shown in the manuscript (Fig.S1a and S7c).

These data strongly support the specificity and selectivity of the transgenic lines used in this manuscript and we therefore have not added experimental data.

2. Effect of chemogenetic activation on the targeted cells. To my knowledge, this is the first series of experiments using excitatory chemogenetics to activate MLR cells. I think that some control data could be added. Maybe the authors already have those data. In Fig. 5, the authors show that D1 and D2 antagonism reduce the calcium activity in PPN glutamatergic neurons. Do the authors have the calcium imaging data of the same animals when CNO was administered? If yes, it could be used to determine whether CNO increased the frequency of calcium transients up to pre-D1 or D2 antagonist levels. The same recordings could be examined in VGAT_{eD} mice, if possible. Alternatively, the authors could use other approaches, if possible (patch on slice, *in vivo* electrophysiology, calcium imaging *in vitro*, c-fos).

Regarding excitatory chemogenetics to activate MLR: We are not the first to use chemogenetics in the PPN. Kroeger et al. (J. Neuroscience 2017) performed selective activation of each PPN neuronal subtype in a study focused on wakefulness vs sleep. Their work contains a validation of eDREADD induced activation of Vglut2 and Vgat mouse lines with CNO, both *in vivo* and *in vitro*. Their report supports that chemogenetic activation can be performed in all main neuronal subtypes within the PPN. In our dataset, the behavioral effects observed in ChR2 and eDREADD mice are aligned, indicative that both methods activate the neurons targeted. Still, we agree with the

reviewer that further confirmation is important for the strength of our claims. Therefore, c-Fos experiments were added.

Data added ①: We added a c-Fos nuclear count (**Fig.S1b-f**) in a new batch of mice injected unilaterally with eD (5 Vglut_eD, 5 Vgat_eD, 2 WT) and treated with CNO. This analysis shows that for both mouse lines, the highest number of c-Fos⁺ nuclei can be found within the PPN in the injected hemisphere (ipsi). Furthermore, in injected animals the contralateral hemisphere contains a higher number of c-Fos⁺ nuclei than in WT mice indicating that unilateral activation engages the opposing hemisphere. Staining is also seen in other brainstem areas caused by the activity during the experiment. Note that prior to perfusion mice were tested in the Open Field to confirm increased locomotor output upon CNO treatment (**Fig.S1f**). Colocalization analysis, performed with higher magnification images, shows that approx. 70% of eD_mCherry neurons contain c-Fos. These experiments confirm that eD expression can effectively activate both PPN neuronal populations targeted.

3. Is the locomotion evoked normally coordinated? Could the authors illustrate whether locomotion evoked by their optogenetic and chemogenetic stimulations is normally coordinated? The authors could use limb kinematics or the footfall patterns. Such data would be useful to compare the locomotor patterns evoked by glutamatergic vs. GABAergic PPN neurons. This would also be important in the context of the clinical relevance.

We agree with the reviewer that this analysis would bring valuable information. We performed 3 new evaluations of locomotor function, focusing on distinct aspects of locomotor performance.

Data added ②: We added **limb-kinematics** data (**Fig.10a-c** and **S10a-e**) extracted from the side-view cameras of Open Field recordings (using DeepLabCut) in Vglut_eD and Vgat_eD injected with haloperidol or SCH and treated with CNO, in comparison to Sham mice and CNO-only conditions (all seven groups were analyzed). We show that limb-kinematics after locomotor recovery reported in our chemogenetic experiments is indeed equivalent to what is observed in healthy mice. The only exception occurs with SCH-Vgat_eD mice which show prolonged stance phase and hyperextension during the step cycle.

Data added ③: To describe skilled locomotion we used data from ladder crossings (**Ladder test**). We evaluated paw placement and fore-hindlimb coordination in Vglut2_ChR2 and Vgat_ChR2 mice injected with haloperidol or SCH and rescued with PPN-Optogenetics, in comparison to WT mice. Step performance data is presented on **Fig.10d-e**. These data show that the phenotypic rescue driven by optogenetic stimulation Vglut2_ChR2 leads to proficiency in a skilled locomotor task but with less fore-hindlimb coordination in SCH-Vgat_ChR2 mice.

Data added ④: To evaluate locomotor adaptation to a complex external scene, we tested our Optogenetic mice in an obstacle course (**Obstacle corridor**). We evaluated crossing times for 3 different types of obstacles (moving-rod, slalom-turns and stair-climb and descent). A new group of mice was generated for this experiment (n=10, Vglut2-CnF neuron inhibited by iDREADDs and concomitant activation of Vglut2-PPN neurons with ChR2, these mice also took part in other experiments requested by R#3). Mice were allowed to move freely, and individual data was compared in two-states:

saline injected vs SCH+CNO+opto. We observed no differences in the average time to cross each obstacle type before and after interference (**Fig.10f-i** and **S10f-h**).

Taken together, these 3 new locomotor proficiency analyses show that locomotion resulting from cauda Vglut2-PPN neuron activation is like that exhibited naturally while there are deviations in coordination resulting from caudal Vgat-PPN neuron activation in SCH challenged mice. Furthermore, it demonstrates that locomotor output is under the subject's control and can be adapted in a complex environment. The conjoint of this data is presented on a new segment within results and addressed in the discussion segment.

4. Unexpectedly, the authors show that optogenetic (Fig. S6, **now Fig.S8**) and chemogenetic (Fig. 7, **now Fig.8**) stimulation of PPN GABAergic neurons (VGAT-positive) increase locomotor activity. This finding contrasts with Caggiano et al. Nature 2018 and Roseberry et al. Cell 2016 (DOI: 10.1016/j.cell.2015.12.037), who showed that optogenetic activation of GABAergic PPN neurons decreases locomotor activity. Therefore, I was a surprised that the authors did not underline in the result section that this was unexpected. I think this should be more clearly discussed in a dedicated paragraph about GABAergic neurons.

Regarding the “unexpected results”: The reviewer is correct in asking for clarifications. We failed to describe how nuanced aspects of descriptive vocabulary can cause the results obtained here to seem divergent from other reports **when in fact the results can be aligned**. We have improved the text (**489-494 and 583-588**) and mentioned the differences in a dedicated segment within the discussion(**780-785**). We were careful to properly interlink the results between articles.

The nuances of GABAergic-PPN neuronal stimulation effect: Rosberry et al. reports the effect of MLR-Vgat (not PPN specific) optogenetic activation (20-Hz, 5s duration, 5s observation window) in head-fixed mice as follows: *“stimulation of the GABAergic population during running caused deceleration (Figures 2H and S2F) but no change when the mouse was stationary at the beginning of stimulation (Figures 2E and 2J).”* Note that their graphs show how locomotor speed drops from 20cm/s, but not necessarily reaches and stays at zero. When stationary, mice seem to engage in slow locomotion 3s after stimulus start. Thus, our interpretation of the results are that mice which are moving will reduce speed and can stop, whereas mice which are still can move, but will do so at slow speed and not continuously. In our Nature paper (2018) we report data from mice freely moving in a corridor. Stimulation was performed in PPN-Vgat neurons (PPN specific, 20-50Hz, 1s duration, 2.5s observation window). Stationary mice did not initiate movement within the first 2s of stimulus start (borderline to the measured latency to initiate locomotion reported here, **Fig.S8b, d, f**). When moving (26.7cm/s on average) animals decreased speed to 8.6cm/s and show occasional stops that were not quantified (Figure S3b of Nature article). Note that here, we show how chemogenetic stimulation of PPN-GABAergic neurons causes a preference for speed ranges between 5-10cm/s intermingled with stops (**Fig.8d**) and that with optogenetics those features remain.

The text clarifications will help readers identify the alignment of results reported by different labs and/or facilitate discussion and interpretation of present/future data.

- 4.2 To better understand the respective roles of glutamatergic and GABAergic PPN neurons in the control of locomotion, it would be interesting to examine, if possible, the effect of D1 and D2

antagonism on PPN VGAT-positive neuron activity recorded with a GRIN lens. The authors did the same experiment for PPN glutamatergic neurons and showed that their calcium activity decreased after D1 or D2 antagonism (Fig. 5).

Although we agree that this analysis would be of interest, we have refrained from doing such experiment because it would require a large amount of time. We are certain the PPN-GABAergic neurons play a role in the motor function as described here and we address this role more clearly in writing in the results and discussion.

5. The number of observations and the post hoc after the ANOVAs. The statistics are well reported in the paper. I only have two small comments.

We have made further adjustments to stats reporting, please see R#2 point 5.

- 5.1 The number of observations is not always clear in the text. Could the authors please define the number of trials or at least the number of animals tested when reporting their results in the text?

We indeed did not write explicitly the number of animals within the main text. Mostly due to space concerns. Yet, we did a series of adjustments to help the reader find the information on group numbers. In this reviewed version, the number of animals is specified in each Figure (within the image) or in the legend (by panel or at the end of the legend if n is the same in all reports). Cases where replicates are analyzed have further description within Methods and the replicate values are stated in legends. We also made sure to guide readers towards our statistical tables (in supplementary) which contain detailed information and include the number of mice tested. In conclusion, we adjusted our reports in the revised version of the manuscript. We hope it is okay for the reviewer.

- 5.2 I had some troubles finding the result of post hoc tests in some cases. For the ANOVAs, when the ANOVA is not significant, it is OK of course not to specify the results of the post hoc tests. However, when the ANOVA is significant, I would expect a value for each post hoc test between groups, time point by time point. Either more stars should be drawn in the graph, or more information should be given in the legend ("more stars in the graph"; this is how we report now, we use connecting lines to avoid drawing too many stars when significance is the same). Let's take for instance Figure 1B, there is a "ns" for t=5 min, and then a single "*****" for all the other time points. Does this mean that the post hoc test was applied to each time point between the two groups and that the significance in "ns" for 5 min and then "*****" for all the other time points (10 min, 15 min50 min)? (report corrected to show value at each time bin) Two other examples. In Fig. 2c and Fig. 3a. Is there a "ns" just for t=20 min, but a "*" difference at time = 10, 30, 40 and 50 min? (yes, here we report group effect rather than point-by-point. First vs second period of the test. Stated in legend) Same for Fig. 6c, b, e... It would be good to specify to which time points the significance applies.

Upon re-reading the text we understand how this was unclear and as pointed out by R#2 (point 5), this work has a lot of statistical reporting. We have amended those cases (those are marked in the details, point-by-point notes). In short, no changes to the statistical analysis were made during the review, though more stats are added in response to reviewers' requests, and in relation to new experiments. We focused on performing modifications in the reporting to facilitate the overall reading.

Regarding point-by-point reports in figures such as 2c, 3a, 9a,d and S6i. Reporting post-hoc results will lead to a higher significance over time bin, not lower (lines diverge as time passes). Yet we focus on the improvement over the test period, we believe that

our reporting approach (ANOVA group effect) is representative in these cases. It also makes it easier to discuss the recovery profile by considering the full period instead of a time point (for example 133-134).

Reviewer #2 (R#2)

In this manuscript, Masini and Kiehn examine the role of the caudal pedunculopontine nucleus (PPN) in movement, specifically whether its activation can improve motor function in drug-induced parkinsonian mice. Using a combination of approaches including chemo- and opto- genetic approaches of either glutamatergic or gabaergic neurons in this region, they conclude that, in particular, activation of excitatory neurons in this region can improve motor function.

I will say from the outset that this is an impressive piece of work (and I don't just mean it's length, I mean it's quality too!), and is beautifully illustrated. I have no doubt that it will be of great interest to the community, in particular those involved in movement neuroscience from either a fundamental or clinical viewpoint.

I think that some straightforward modifications would strengthen the manuscript, and accordingly suggest the following. In my view, no further experiments are needed to convince me of the results.

We thank the reviewer for the positive comments.

1. Deep brain stimulation. The authors take a DBS angle. While this is worthwhile in the Discussion, there are so many issues with clinical DBS that it's barely worth mentioning in the Introduction... But okay, if that's what they want. So I will focus on the Discussion: in DBS, a large diameter (>1mm) electrode with cylindrical surfaces of 1.5 mm are placed... somewhere. In the case of PPN targeting, the electrode is placed in the dorsolateral mesopontine tegmentum, so they are close to the PPN (and injuring it), but lots of variability. Then a whole bunch of current is passed through these large targets, and then the current ... does what? Who knows? Excites axons? Produces depolarizing block? Switches neuronal state/responsiveness? The fact is, we don't know despite all the poorly-supported papers about low and high frequency stimulation that have been published. DBS is crude at best. The STN is more homogeneous and well defined than the PPN, so it's an easier target. The PPN? In fact, the point of this paper is that other therapies – more targeted – are needed. So line 529 “the prime target for therapy in humans” rather than “for DBS.” But the point is, this terrific manuscript defines that PPN DBS is fraught with even more problems than we thought, and new therapies are needed. Electrical therapies? Seems unlikely to me. (I would suggest modifying the last sentence of the Intro, for example, as well as that bit in the Discussion. In short: no need to say we need better DBS... we need more targeted therapies!

We agree with the interpretation by R#2 and we have done modifications to the discussion (834-839) which better emphasize the need for new therapeutic approaches.

2. PD mouse models. The authors use drugs to induce parkinsonism in mice. That is fine as there is no really good rodent model of PD (that I know about), but the models presented here are not models of PD and the distinction should be made throughout the paper. (Paragraph at 431 could be improved). Other than MPTP-induced PD in humans, PD is a relatively slow neurodegenerative process, during which neurons in nuclei such as the PPN can adapt. In fact, we don't even know that we have PD until at least 1/3 of our nigral DAergic neurons are gone. We are homeostatic beasts. So I would urge caution throughout about calling this a PD model – it's a model of parkinsonism. The degree to which activating these neurons will benefit a slowly developing disease is not known – and does not detract from the novelty of this manuscript. But the authors should discuss this in that paragraph.

We have adjusted our vocabulary throughout the manuscript (including title) making no direct references to our model as a “Parkinson’s disease” model but rather referring to, ‘dopamine signaling interference/depletion’, ‘acute dopamine depletion’, ‘parkinsonism’, ‘made akinetic’. Note that these amendments also follow the feedback given by the other reviewers. We have extended the discussion of the models in the discussion including the chronic changes (706 ‘Methodological considerations’; 775-777). Also see further notes on R#3, point1.

3. Having said all that, what’s really call is the differential result in locomotion and the bar test. PD presents differently in different people, with different proportions of the cardinal symptoms/signs. Here, the authors report differential effects (with Gaba expts) on bar test and locomotion. That is really cool and could be discussed – different symptoms/signs have different circuits.

We thank the reviewer for pointing out that differences in recovery profile are indicative of variation in circuitry access given by the neuronal population targeted. We have added a segment that extends our dialogue on this topic (537-538; 801-811). To this discussion we also included the new results obtained when targeting Vglut2⁺ neurons within the CnF.

4. Control mice. In many of the experiments, the control mice (Sham, *e.g.*, Fig 1D, Fig 2D, Fig 3B) are running faster (quite fast, in fact) than the experimental mice at the outset. This is a bit disturbing, as it implies the groups are not equivalent. Does their reduced activity later on result from them being tired out? Why are the groups not equivalent? Do the viral vectors damage some neurons in the PPN for example?

We believe the R#2 might have misunderstood the reading of those graphs. We have done some adjustments to the text (different locations, small changes) to aid the reader. We discuss our points here. The graphs refer to speed range (x-axis; 2-5cm/s, 5-10cm/s...) used by the mice when locomoting (>2cm/s). The y-axis (% of time) represents the proportion of locomotor time spend in each speed-range. **Figure 1** (as example, below) has Sham and Vglut2_eD which have been injected with CNO. Excitation of PPN-VGlut2 neurons promotes locomotion and mice show preference for speed ranges above 10cm/s, therefore a drop in percent of time at 2-5cm/s (red arrows). Thus, the first min-max bar (blue arrow) represents Sham^{CNO} group (grey) spending near 40% of their locomotor time at speeds between 2-5cm/s.

To verify if outset conditions are equal, it is best to see the graphs of distance moved. Figure 1b (as example above) shows no difference in distance moved in the first 5min of the Open Field (yellow highlight).

Regarding Figures 2 and 3, the interpretation relies on the level of immobility prior to CNO injection. Let's take an example again (**Figure 3**, where the difference you noted is bigger, image placed below). In panel **b** all animals were injected with SCH and then CNO. The Sham (grey) group is bradykinetic. Note that if they move, they do so at 2-5cm/s (60% of the time, black arrow). The outset condition is seen in **panel a**, where both points superimpose at min 20 (but note that the ns refers to group comparisons within the first period [stated in legend]).

Regarding the question of animals being tired out. As mice explore the chamber, it is expected that they reduce activity over time (habituation). Tiredness is a possibility and can be hard to distinguish it from habituation (time frame here is long). Here is what we have described: Vglut_eD (**Fig.1b**) have a peak at bin 15 and show no habituation (**Table S1 row 2**). Vgat_eD (**Fig.8b**) maintain activity throughout the 50min session (no habituation, **Table S2 rows 1-2**). On the new batch used for c-Fos analysis [R#1] we also performed an Open Field test (to confirm motor output, **Fig.S1f**), for this group we had a longer recording session (up to 150min) please see our notes regarding effect wear out on R#3>Minor Points>the full graph is there.

5. Statistics. There are a lot of statistics in here, and they seem appropriate although I am far from an expert. However, the use of the SEM is not appropriate. The SEM may be appropriate when one is trying to quantify experimental error. But in biology, we're interested in biological variability. Hence, a measure of variance – such as standard deviation – should be used. There are a few instances in the ms where the authors switch to sd, but it's not clear why. I would strongly suggest completely eliminating SEM throughout. Even better than showing SD would be to show all data points, but I recognize that this may clutter some of the graphs.

We have not completely eliminated SEM in the text. But we performed major changes in panels and graphs. Graphs of distance moved over time are maintained with SEM as common in similar publications. All other graph types have one of the following: min-to-max bars, individual data points, Bars without data points have SD, Line graphs with group average have confidence interval (CI, 95%) and in some instances the confidence interval is plotted per mouse to indicate individual variability.

As an overall rule we did the following. 1) when reporting an effect size, we use -fold or % of difference. Yet cumulative graphs contain SD (*e.g.*, **280**). 2) when reporting data range, we use SD (*e.g.*, **439**). 3) when reporting involves interpretation of response variability, we use CI (**439**). Note that if the reader checks the stats table the report there will include further measures. The new data which has been added follows this same patterns.

- 6. Calcium imaging plots** – these are not showing $\Delta F/F$ but rather the z-scores. The labelling and legends can be changed to indicate this. Also, it's not clear to me how the baseline was determined.

We have adjusted the axis. The Standard Deviation of dF/F [SD, as presented in **Fig.4e-f** and **5b-c**] is the unit of measure. The reviewer is correct in noting that we calculated Z-scores (as indicated in methods formula [2]). The calculation we used places the score over a normal distribution curve ranging from -3 to +3 (99% of the values should lie between these two numbers) (for example: **Fig.4f** and **5c**). In conclusion, what is measured, and the unity of measure are clarified.

The baseline calculation is clarified in legend (236, 261) and methods (1144-1145).

- 7. Experimental unit.** Although the authors state that ARRIVE guidelines were followed, they are not always clear. This is especially true when it comes to the experimental unit. For example, in the calcium imaging experiments, there are n mice. But then there are about 11 neurons recorded per day in these mice. It's not clear how many days they were then recorded, and how many of these neurons were the same day after day. This creates confusion. For example, the authors state that about 80% of neurons showed reduced activity – but were these evenly spread across mice, or were some mice more “affected” than others? This could be analyzed, and in fact Fig 4F and 5C could be divided into mice. And if a cell recorded more than once, that could be indicated as well.

R#1 also pointed out this. We have adjusted multiple instances within the main body of the text, in which the number of mice was unclear (R#1, point 5.2).

Regarding calcium Imaging. We have added **new analysis (Fig.S2d-g)**. In short, each mouse is recorded twice (SCH or Halo). The Z-score shows overall decrease even if we look at individual mice (**Fig.S2d**). The neurons can be followed over the two experimental days (**Fig.S2e**) and if neurons are grouped by response type one realizes that neurons that increased their response do not necessarily reach a drug-free Z-Score (**Fig.S2f**). Finally, some neurons can change response depending on drug (**Fig.S2g**) but general proportions (thickness of line indicates the proportion of all neurons) are maintained with the vast majority showing a reduced activity after drug. These swap cases are very few and there is no specific animal accounting for their occurrence.

Reviewer #3 (R#3)

This study aims to identify the role of glutamatergic neurons of the pedunclopontine nucleus (PPN) in reversing the motor impairment produced by acute dopamine depletion. The authors accurately reflect on the controversial benefits of deep brain stimulation in the PPN of Parkinson's disease patients and propose that the neurochemical heterogeneity of this structure is likely playing a role in the inconsistency of the results in the clinics. While this idea has been around for some time, it had not been experimentally tested until now. Following a recent publication by the same group, where they show that optogenetic activation of PPN glutamatergic neurons increase exploratory locomotion (although see comments below), in this study the authors use this previous knowledge to manipulate the activity of these same neurons and counteract the decrease in motor activity that follows the administration of D1 and D2 dopamine receptor antagonists. The results show that the activation of PPN glutamatergic neurons by means of chemogenetics or optogenetics successfully compensates for the motor impairment. The authors also tested the same manipulation on PPN GABAergic neurons and observed a much more limited effect. In neither case (glutamatergic or GABAergic) is clear in the manuscript the connectivity that would underlie such effects, or lack of effects This study uses a wide array of techniques and an adequate use of controls. Overall, it is a carefully executed work addressing

important and timely questions. However, I have a series of concerns regarding the interpretation of the data and I am skeptical about the advancement that these data represent for PD research or basal ganglia function.

We thank the reviewer for the detailed suggestions. We have performed a range of new experiments to address the concerns raised and the inclusion of these new experiments has emphasized the importance of this study.

MAJOR CONCERNS:

1. **Acute pharmacological dopamine depletion** is not a model of Parkinson's disease. Haloperidol is widely used to treat a variety of conditions including mood disorders and aggression. I am sure that the authors do not consider a dose of haloperidol as a temporary parkinsonian state. PD is much more beyond dopamine depletion, it involves a series of chronic compensatory changes that involve alterations in the firing properties of neurons across the basal ganglia circuit and up/downregulation of a variety of receptors. The authors should interpret their results accordingly and rewrite their manuscript focusing on the acute dopamine depletion rather than presenting it as a model of PD.

The reviewer has made an important consideration which is in line with both R#1 and R#2. Major changes in vocabulary/description have been done throughout the manuscript. On our **notes to R#2, point 2**: "We have adjusted our vocabulary throughout the manuscript (including title) making no direct references to our model as a 'Parkinson's disease' model but rather referring to, 'dopamine signaling interference/depletion', 'acute dopamine depletion', 'parkinsonism', 'made akinetic'. Furthermore, we included a dedicated section in the discussion that deals with this issue (**706-739 'Methodological considerations'; 775-777**).

We reinforce that we understand and agree with the 3 reviewers. Thus, changes were made in the text. We note that no animal model replicates the full spectrum of human PD disease. In strict terms, there is no PD-model: An ideal model of PD should consist of pathological and clinical features, involving both dopaminergic and non-dopaminergic systems, the central and peripheral nervous systems, plus motor and nonmotor symptoms. Moreover, display such features with an age-dependent onset and progressive nature. None of the available models encompasses all of that. Yet, the use pharmacological approaches that cause acute dopamine signal depletion has several advantages. In particular, they produce strong motor suppression, characterized by akinesia, bradykinesia and delayed motor response, which is reversible over time and allow for pre-post comparisons. Moreover, we could compare perturbations of both indirect and direct pathways on the same cohort of mice. We agree that positive and negative modelling-factors must be presented to the reader, and we extended this discussion in text. Nonetheless, *we maintain our view that the present study presents a significant advancement for PD research by reproducing some of the motor symptoms seen in the disease and speaks to basal ganglia function.*

2. Another important concern relates to the effect of stimulating PPN neurons. In their previous publication on the MLR, the authors showed that stimulation of the cuneiform nucleus (a structure that is adjacent to the PPN in the most caudal region) produced a strong effect on locomotion, whereas stimulation of the PPN produced a milder effect. In the same paper, the authors showed that PPN and CnF are interconnected through glutamatergic connections. Their results, however, contrast with other publications showing either no effect or an opposite effect on locomotion

following the stimulation of PPN glutamatergic neurons (*i.e.*, inhibition of motor activity; see Kroeger et al., 2017, J Neurosci; Josset et al., 2018 Curr Biol; Dautan et al., 2020, Biorxiv). In this context, it is concerning to see the spread of the transduction into the CnF in Figure S1-a5. From the evidence provided it is then possible that optogenetic or chemogenetic stimulation of the PPN also activated CnF. Neurons, which produce a robust effect on locomotion.

Another possibility is that, given the connectivity between PPN and CnF, activation of PPN neurons recruit CnF neurons through excitatory synaptic transmission. In line with this, the long latencies for movement induction reported for PPN stimulation (~1.5s) fit with the idea that PPN neurons are recruiting other circuit components to produce increased locomotion. To avoid such ambiguity, **authors should provide convincing evidence that the observed effects are not mediated by CnF activation.**

We have added additional experiments to exclude those possibilities (**new data ⑤ and ⑥**). The animals composing the added experimental groups had injection sites carefully mapped. Importantly, we do NOT argue against Vglut2-CnF stimulation as a possible treatment as it can induce locomotion and therefore alleviate some of the motor deficits in PD. Yet, we now show that Vglut2-CnF is not the cause of the effect we see from Vglut2-PPN stimulation and that a chemogenetic approach that activates the glutamatergic CnF neuronal population has a different locomotor behavior (see below).

Prior behavioral evidence supporting that stimulation in this study activates PPN and not CnF: In our Nature paper we showed that optogenetic activation Vglut2-Cnf neurons produced a wide range of speeds and all gaits: the alternating gaits of walk and trot and the synchronous gaits of gallop and bound. At 40Hz stimulation frequency, gallop or bound was evoked. The latency for onset of CnF-induced locomotion was in the range of 100-150ms (as R#3 described), similar to what was reported in Josset et al. 2020. In contrast stimulation of Vglut2-PPN neurons always produced walk and trot and never gallop and bound, even at 40Hz stimulation. These divisions in gait type were also seen after selective inactivation of PPN or Vglut2-CnF neurons. In the present study, we never observed gallop or bound driven by Vglu2-PPN stimulation, although we consistently stimulated with 40Hz. Note how mice always moved with slow speeds in the walk and trot range when stimulated (**Fig.6d and f**). Moreover, the latency to move (ranged between $2.5 \pm 1s$; SD) just as previously reported. Finally, in the Nature article, the maximum speed when stimulating Vglut2 PPN-ChR2 neurons was 19cm/s, compared with 56cm/s for Vglut2 CnF-ChR2 neurons. Thus, from within lab comparisons, if our targeting strategy had substantially hit the CnF, we would have expected to observe escape-like locomotion. The analysis of **inralimb kinematics** (R#1, point 3) that we have added also confirms that PPN-stimulated mice show preferably walking (chemogenetics, **Fig.S10c-e**).

Regarding anatomical connectivity between CnF and PPN: As referred by the Reviewer we indeed described that PPN and CnF are interconnected through glutamatergic connections in our Nature paper. One important detail, however, is that the *dominant projections arise from the CnF towards the PPN, not from PPN to CnF*. It is, therefore, less likely that PPN neurons recruit CnF neurons through excitatory synaptic transmission although some projections are present (see next point).

Regarding viral spread on Fig.S1a: The apparent ‘spread’ of staining in the CnF border to PPN is mostly from fibers, not infected cells. We have adjusted this figure (**S1a-inset5**), which was also a request from R#1, and the image now is presented in larger

format. This should clarify that what appears to be labeled neurons are in fact mostly fibers or dendrites.

Since this is an important point, we wanted to further address it. Thus, **additional experiments with inactivation of Vglut2-CnF when stimulating Vglut2-PPN neurons in the drug model** have been added to the manuscript.

New data added ⑤: We have performed bilateral chemogenetic inhibition of **Vglut2-CnF neurons** with concomitant optogenetic activation of the **Vglut2-PPN** in *Vglut2^{cre}* mice (n=10, Saline, Halo and SCH experiments). The site of injections for this batch has been mapped and show restricted expression (by viral type) in CnF or PPN (**Fig.S5a**). We show that Vglut2-CnF inhibition blocks fast locomotor speeds indicating that the CnF inhibition is effective (Corridor test; **Fig.7b-c**). We show that in the Open field, where speed ranges are generally low, mice can locomote when Vglut2-CnF neurons are inhibited (**Fig.S5c-e**). Finally, in the situation where CnF is inhibited PPN activation is still able to promote locomotion with and without dopamine depleting drugs (**Fig.7d-i** and **S5f-j**).

New data added ⑥: To further address the effect of stimulation of Vglut2-CnF we did extra experiments with targeted eD in the CnF (n=5, **Fig.S6**). Then compared CNO-only effect or CNO-akinetic condition. We describe how CNO injected mice move at very high speeds, faster than control and faster than Vglut2-PPN excitatory DREADD animals. We identify the development of a ‘darting’ behavior. The abrupt changes to high-speed locomotion are present in Haloperidol challenge and the main factor contributing to the increase in distance covered. For this new group we also show a detailed mapping of injection site. *In conclusion, by specifically targeting the CnF we see a very different phenotype than in our PPN experiments.*

3. Dopamine depletion was achieved by the systemic administration of haloperidol and SCH23390. It is therefore expected that dopamine transmission will be blocked across all brain regions. One of such regions is the PPN, which receives direct projections from dopamine neurons of the substantia nigra pars compacta (Ryczko et al., 2016, PNAS) and the zona incerta (Sharma et al., 2018, Sci Rep). However, the interpretation of the data given by the authors is exclusively based on the effects over the direct and indirect striatal pathways. To be able to maintain these conclusions, the authors need to provide evidence that the effects of dopamine antagonists on PPN activity are not a consequence of direct binding on PPN neurons.

The reviewer correctly points out that there are dopaminergic projections from SNc and zona incerta to PPN although neither Ryczko nor Sharma showed the specific neuronal target in PPN. Our own transsynaptic labelling data supports that there are connections to Vglut2-PPN neurons from SNc and Zona Incerta (Fig.6 of Nature 2018.). Both pathways may have a locomotor promotion effect through D1 receptors on PPN neurons. Therefore, in Parkinson’s Disease, where SNc cells are lost, the locomotor phenotype will be affected through this pathway (this is also what both Ryczko and Sharma conclude). Thus, the D1-antagonist effect we see might clearly involve some effect on PPN neurons directly. We do not think this is the case for D2 antagonism though, since D2 receptors have not been found in PPN. We have now tuned the text so that the beneficial effects of caudal-PPN activation in D1 antagonism are not only linked to its interconnectivity with the Basal ganglia but also to local receptors (**723-731**). We also clarify that the akinetic and bradykinetic state observed in our dopamine-receptor

antagonized mice has been described in rodents with striatal cannula infusions of these drugs (716). We therefore do not see that this point jeopardizes our conclusions.

4. The manuscript fails to cite previous studies that have made important contributions to research on the role of the PPN in PD. While there are plenty of these studies to include here, one of them in particular stands out for its relevance to the present data. In a 2015 publication, Gut and Winn showed that DBS across different regions of the PPN in the 6-hydroxydopamine model of nigrostriatal degeneration has different consequences for motor activity. They report that DBS in the caudal PPN improved different gait parameters. Authors should include some of these references in their Introduction and/or Discussion to better reflect the current knowledge in the field.

We truly apologize and we corrected this error. We are citing this work with special reference to electrical stimulation in the caudal PPN (citation #30, 47-50). The work from Gut and Winn has been a source of information for this manuscript and should be rightfully included in our citations. Presently we have many citations, and we did not want to leave any great mind uncited. We apologize now, and will need to apologize again in the future, for those who we could not cite.

5. **The rationale for targeting the caudal PPN** (as opposed to the rostral PPN or the whole PPN) is missing. Please include this in the Introduction.

This information is now added to the Introduction and also brought out in the discussion (45-50, 745-749).

6. I find the following statement particularly problematic: the “caudal PPN glutamatergic subpopulation should be the prime target for DBS in humans” (lines 528-529). The data presented in this study does not address the complexity of PD. Making such a statement is likely to mislead clinical approaches where a rigorous assessment of targets is necessary before any interventions in PD patients takes place.

We changed the sentence fully and instead added a discussion segment (706-739).

R#1 DETAILS:

Title “Targeted activation of midbrain neurons restores locomotor function in mouse models of Parkinson’s disease”. Please consider replacing “in mouse models of Parkinson’s disease” with “in mice made akinetic by D1 or D2 antagonism”. **Title adjusted. Now reads: “Targeted activation of midbrain neurons restores locomotor function in mouse models of parkinsonism”.**

Running title “Pedunculopontine activation in Parkinson’s disease” (!). Please consider replacing “in Parkinson’s disease” with “in akinetic mice”. **Adjusted. Now reads: “Activation of neuronal subpopulations within the Mesencephalic locomotor region in mouse models of parkinsonism”.**

Abstract

Line 5-6 “in two pharmacological PD mouse models”. Please consider replacing with “in mice made akinetic by D1 or D2 antagonism”. **Done.**

Line 7-8 “GABAergic PPN population is not sufficient to support motor recovery”. Please consider modifying this sentence to be closer to your results, with something like “glutamatergic cells evoked

sustained locomotor activity in mice made akinetic by D1 or D2 antagonists, whereas GABAergic cells only evoked sustained locomotor activity in mice made akinetic by D2 antagonism". Done.

Introduction

Line 13-20. I understand that the authors highlight that abnormal BG activity is considered as an important part of the PD pathophysiology. However, the cause is the degeneration of DA cells. It should be stated in the first paragraph, even if the approach of the study does not rely on DA cell death. Done.

Line 26. "increases the relative contribution to D2-MSNs". Should the authors also mention the decreased inputs to D1-MSN in PD? This might be useful to increase the relevance of the decreased striatal calcium activity they recorded in mice made akinetic by D1/D2 antagonism in Figure 4 and Figure S2? Sentence adjusted.

Line 47-48 "two distinct PD mouse models". Please consider replacing with "mice made akinetic by systemic administration of D1 or D2 antagonists". Done.

Line 51-52 "self-paced locomotion". Maybe remove "self-paced". How do we know that such locomotion is self-paced? Sentence adjusted.

Line 53-54 "PPN caudal GABAergic neurons could not restore locomotor capabilities". To avoid confusion in the mind of the reader, this should be modified because it contradicts the results of the present study, which shows that their activation increases locomotion in control mice (chemogenetics Fig. 7; optogenetics Fig S6) and in mice made akinetic by D2 antagonism (chemogenetics Fig. 7; optogenetics Fig S6). However, it is true that their activation does not induce locomotion in mice made akinetic by D1 antagonism (chemogenetics Fig. 7; optogenetics Fig S6). Adjustments made (Fig. 8-9, S7-9).

Results

Line 64-65 "known innervation targets". Please specify those targets. Done.

Line 70 "locomotion". Please define what you quantified here as locomotion. Sentence adjusted. Defined on methods as speeds above 2cm/s.

Line 71 See minor comment 5 about the stats. Solved. Stats on graphs are made clear by connecting lines to indicate which group is used as reference. Number of animals is present on figures or legends and restated on statistical tables.

Figure 1 Legend. Line 88. Please define what appears in white/grey: is it the neuronal staining (black is no signal, right?). Please specify the neuronal marker. Also, please see comment 1. A higher magnification of this should be added maybe as a supplementary figure to see which neurons were infected. I cannot see any cell on this picture. Color code changed to: NeuroTrace = green (as in supplementary). The images of Fig.1a are made bigger and atlas reference lines reduced, facilitating section visualization. Supplementary figure 1 contains images with higher magnification, and the anatomical panels were made bigger.

Line 101 "PD phenotype" [...] "with dopamine receptor antagonist haloperidol or SCH23390". The specification of the pharmacological tools used should come much earlier in the text. Done. Now it is clear in the first sentence of this paragraph (114-116).

Line 106 "halo 0.5 mg/kg". Could the authors please add a reference or two to justify this dose? Done. The reference article we added shows plasma levels of D2 occupancy and behavioral

catalepsy for different dosages of Haloperidol as performed in Rats (PMID: 10779880). In the cited article the following is stated: “With 0.5mg/kg the D2, occupancy was 94% at 1 hour, 91% at 2 hours, 88% at 4 hours, 80% at 8 hours, and 39% at 24 hours. As expected from the threshold for catalepsy observed at the 2-hour mark, animals showed catalepsy only at times when their D2 occupancy was above 80%.”

Line 116 “vs med of 20s”. Please explain, I do not understand. Med is spelled out as “median”. Overall explanation: the statistical approach applied in the Bar-test is comparing Sham^{halo+CNO} and Vglut_{eD}^{halo+CNO} to test if median values differ between groups. We report the median difference using a 2-tailed Mann Whitney. To test if animals remained in the bar until the end of the 20s cut-off (comparing Sham^{halo+CNO} to expected median of 20s) we used the Wilcoxon test, and a significant p value indicates the group median is statistically equal to the cut-off time.

Line 120 “profile max difference” [...] “35% for Sham” “5% for Vglut_{eD}”. Could the authors please explain with sentences? This is part of the stats report and relates to the distribution of time spend in each speed range by group. The use of a speed range differs from healthy (WT^{sal+sal}) by 35% in Sham^{halo+CNO}. Whereas Vglut_{eD}^{halo+CNO} differ by only 5%. Adjustments were made to clarify the sentence in text.

Figure 2. Panel A. Above “Injection 1”, could the authors add “sal/halo” and above Injection 2, “sal/CNO” to improve clarity? Image adjusted. Note that we added text to the box on the right panel instead. Indeed, improves clarity.

Line 149 “SCH 23390 (0.25 mg/kg)”. Could the authors please add a reference or two to justify this dose? Done.

Line 179 Please replace “basal ganglia” with BG if already abbreviated. Done.

Line 185-186 “PD1-4, 3 min REC/3min OFF”. Could the authors please make sentences to make this clear please? Done. Calcium imaging recording intervals were clarified in text.

Line 189 “high calcium dynamics that declined “. Could the authors please add a statistical analysis to support this? Done. We used Spearman’s correlation.

Line 190 “81%”. Could the authors please add a SD or SEM? **Line 191** “78.9%”. Could the authors please add a SD or SEM? Note that we are referring to the proportion of all neurons recorded, not per/mice (pie-graphs, **Fig.4g** and **5d**). This report approach is common in calcium imaging and is valid here because no individual mouse accounts for the effect, as seen on the supplementary (**Fig.S2d**) where we plot Z-scores per animal/drug. In the text we have added the count to facilitate the interpretation. It now reads as: “Locomotor suppression was associated with silencing of 81% (128/158) of the D1-MSN population under haloperidol and 78.9% (131/166) under SCH23390”. To address this request, we also added a new panel (**Fig.S2f**). The new graph shows the effect size for neurons classified based on their response to the drug (reduced vs increased firing) and is reported with standard deviation [SD] together with the confidence interval (95%) for each group of neurons.

Figure 4. Line 199-200 Panel d. Could the authors please add a statistical analysis for the decrease in locomotor activity? Done. **Fig.4b** and **S2c**, see legends.

Line 201-202 Panel e. “maximum deviation from average fluorescence through the 24 min recording).” Could the authors please define what this means? Maybe in the methods. Within the legend we have added a short support sentence to aid interpretation: “pixels with large differences in intensity become brighter in this projection” (**Fig.4e**).

Line 207-208 Panel g. Could the authors please add a statistical analysis for the decrease in calcium activity? This panel (**Fig.4g**) contains pie-charts showing a categorical classification of all neurons. It is an absolute number with no stats. We have added a new panel to **Fig.S2f**. This panel shows the effect in each group (neurons that decrease and neurons that increase) for each experiment. Main stats are added in text and multiple comparisons described in legend. Same is valid for **Fig.5d**.

Line 214 “72.9% and 80%” Could the authors please add a SD or SEM? Refers to **Fig.5d**. Please, see note above.

Line 214 “reduced activity”. As for the legend, could the authors please add a statistical analysis for the decrease in calcium activity? Done. Please see legends in figures **4f** (STR) and **5c** (PPN).

Line 215 “with the remaining neurons increasing their activity”. Could the authors please add a statistical analysis for the increase in calcium activity? Done, reported on new **Fig.S2f**.

Line 216 “the data shows”. Please replace with “show”. Done.

Fig 5 panel b. Do the authors have the same recordings under CNO in Vglut2_eD to show the increase in calcium transients? Unfortunately, the data was lost to covid19 restrictions.

Fig 5 panels c and D. Could the authors please add a statistical analysis for the decrease and increases in calcium activity? (Cf comments on Fig 4). Stats added with overall effect in text and new panel **Fig.S2f**, discussed above.

Line 239. Please specify the duration of the epochs. Epochs (pre, laser, post) are described in the end of the paragraph (10s each).

Line 242 “2-3.5 mW”. How was this measured? Described in Methods (connector tip). Please see new panel **S3h**.

Line 243 Please delete “random”. Done.

Line 246 “laser reliability”. Please define. The output measure is now referred to as: Fraction of successful trials. Adjustment done throughout the text.

Line 251. Please define T and S. These are statistical report values. All are now described in a paragraph within Methods>Statistics>Statistical methods. In short, T is number of data sets (in this case T=3 -> pre, laser, post), S is the number of subjects (in this case S=10 because there were 10 mice in the experiment).

Line 252-253. This is not a sentence. Could the authors please rephrase? Good point. Was missing the subject. Correction done.

Line 258 “light”. Please specify wavelength. Done. Note that wavelength used to excite ChR2 is also described in the first paragraph of the results segment describing the protocol for the Open field experiments.

Line 263 Please define W. These are statistical report values. All are now described in a paragraph within Methods>Statistics>Statistical methods (1269). The Wilcoxon test is nonparametric, comparing paired values by computing their difference and ranking those differences. In short, W is the sum of signed ranks. If the paired values have no difference (the two data sets basically have the same median) W will equal zero.

Line 273 “postural corrections”. Please make clear whether this test allows you to test locomotion initiation or postural corrections. If the authors were looking at postural corrections, please define what the postural correction is. Sentence adjusted. Note that a definition for postural correction is detailed on Methods>Bar test.

Line 274 “median of 20s”. Please explain. The test used here (One sample Wilcoxon test) is a non-parametric test that compares the median of the sampled data (latency to descent bar) versus a hypothetical median (in this case 20s, the threshold that ends the test procedure). The p value is significant and therefore indicates that there is no discrepancy between the latencies we measured and the hypothetical median. The sentence has been adjusted to aid interpretation.

Line 292. Please specify \pm SD or SEM. Refers to Fig.6d. Done.

Line 293-294. Please specify \pm SD or SEM. Refers to Fig.6f. Done.

Line 299 “triad”. Please check whether this is the good word. Changed.

Line 304. “cholinergic represent the minority”. Please specify the proportion. Done (1/3 of the neurons are cholinergic and GABAergic outnumbers those), citation added (459).

Line 305-307. Please specify clearly that their optogenetic activation decreases locomotion in Caggiano et al. 2018 and Roseberry et al. 2016. Please make clear that their tonic excitation with chemogenetics was not tested. Done. Discussed on main points.

Line 311. Please add “Unexpectedly” at the beginning of this sentence. It is important for the reader to understand that this is a surprise (at least, to me, it was). Done.

Line 313. “129.2 \pm 9.8”. Please specify the unit. Done (meters, m).

Line 319. “The difference in distance travelled vs speed profile between Sham, Vgat_eD and Vglut2_eD mice...”. Can the authors please rephrase this sentence? [refers to results report on Fig.7e] Paragraph adjusted.

Line 322. “stop bouts”. Could the authors please define those here? Done. A precise definition used for the analysis is described in the methods section.

Line 324. “Brown-Forsythe and Welch ANOVA”. Could the authors please define in the methods when this one was used? Done. Methods>Statistics>Statistical methods (1276).

Line 325. Please define “B-F” and “W”. These are statistical report values. All are now described on a paragraph within Methods>Statistics>Statistical methods (1276-1279).

Line 323-324. “Yet the average stop bout duration of Vgat_eD was nearly half of that observed in sham and Vglut2_eD”. This should be discussed, see comment 4. Done. Results and Discussion segments adjusted.

Figure 7. Panel e. Lines 344-347. “as identified by the tracking software” sounds mysterious. Could the authors please define the stop bouts because it is also weird to see the stop in the middle of the Vglut2_eD recording not to be counted as a stop, despite the fact that it looks longer than the ones counted in the Vgat_eD recording. These data are now Fig.8e. We adjusted the legend and included a reference to the methods section. In place behaviors, such as rearing and grooming, are not accounted as stop. Please see the methods segment ‘Data preprocessing and outcome measures’.

Line 350. Please define “T3”. These are statistical report values. All are now described on a paragraph within Methods>Statistics>Statistical methods (1274).

Line 368. I would recommend rephrasing this sentence to highlight more clearly the specific difference with the D2 antagonism. Moreover “CNO treatment” could mean “by itself on Sham animals”. Please specify in Vgat_eD mice. Done.

Line 391-393. This is where the authors are describing a result that contrasts with Caggiano et al 2018 and Roseberry et al. 2016. This should be clearly highlighted. It is not a problem to say it, it is interesting that the result is not the same. Stimulation of MLR cholinergic neurons also led to some surprises from study to study. Solved. Discussed on main points.

Line 393-396. Could the authors please specify whether there was a wavelength control with 593 nm light for these results? Yes. We performed the wavelength test on all optogenetic mice (1014). (including new groups added). In this instance, the text was missing a reference to the supplementary figure (Fig.S9a). The reference has been added and the issue is corrected.

Line 401. “reliability”. Please define. Done previously in the text (on Vglu2_ChR2 segment). The output measure is now referred to as: Fraction of successful trials.

Line 403. “light”. Please specify wavelength. Done.

Line 407-408. Please specify whether this was observed in presence of a D1 or D2 antagonist. Please specify whether a 593 nm laser control was done for these observations. Done.

Line 409-410. Please specify that this is unexpected relative to Caggiano et al. 2018, Roseberry et al. 2016. Sentence adjusted.

Line 411. “IN accord”. Please rephrase. Sentence adjusted.

Line 411-412. Please rephrase to make the sentence more faithful relative to the experiments. The sentence should be more balanced. It should state the absence of effect of VGAT neuron activation when PD was induced with a D1 antagonist, but should mention that the activation of VGAT neurons counteracted the motor deficits in the bar test evoked by D1 or D2 antagonists, and counteracted the motor impairments evoked in the open field by a D2 antagonist. Done.

Discussion

Line 425-427. I do not understand why the authors increase the contrast between the effects of the stimulation of vglut and vgat neurons, considering that the only difference is relative to D1 evoked akinesia specifically in the open field. This is also surprising relative to the end of the discussion, where the authors say that stimulating the two cell types should not be a problem. Maybe rephrase and homogenize in the whole paper. Result-Claim adjusted throughout.

Line 432. "Simulate the neurochemistry". Please replace with "simulate part of the neurochemistry".
Sentence adjusted.

Line 441. "further supports". Please replace with support. Done.

Line 442. "pedunculo pontine nucleus". Please replace with PPN. Done.

Line 470-471. Could the authors please provide single cell controls? See comment 2. This sentence was changed. The claim was based on reports by others, we did not perform such experiment.

Line 475. "We suggest". Space missing before We. Done.

Line 480. "and recover motor function without causing unusual behavior". Please provide controls showing that the optogenetic and chemogenetic stimulation of vglut or vgat neurons induced normal locomotion (see minor comment 3). We have added new experiments to address this issue (Ladder cross, Obstacle corridor and Kinematics, Fig.10 and S10).

Line 483 to 506. Please name a full dedicated section on GABAergic neurons. Please highlight the difference with Caggiano et al. 2018 and Roseberry et al. 2016. Please explain how you could interpret these three studies as a whole to make sense of the contradiction. If the authors performed additional experiments as suggested (see minor comments), please discuss the new results here. Segment added. Discussed on R#1 point 4.

Line 489-490. "previously not reported". Please expand relative to previous results of Caggiano and Roseberry. Sentence fully changed.

Line 492-493. "was insufficient". Again, I do not understand why the authors increase the contrast between the effects of the stimulation of vglut and vgat neurons, considering that the only difference is relative to D1 evoked akinesia specifically in the open field. Please rephrase. See also other comments relative to this point. This sentence has been rewritten to better fit with the results.

Line 496. This statement contradicts the data in Fig S6 c, d, e, (now S8) where optogenetic activation of PPN gabaergic neurons improves locomotion in the open field. Please rephrase. This sentence has been rewritten to better fit with the results.

Line 496-500. If new data is added, this section should be modified to interpret the new data. Done.

Line 506. Could the authors please rephrase to specify the sensory inputs that the authors are mentioning here? Done. Interpretation supported by new data on Fig.10 and S10.

Line 508-509. This sentence is increasing the contrast between vglut2 and vgat neurons, but I do not feel it is justified. Please see the other comments relative to this point and maybe rephrase to be more specific. This sentence has been rewritten to better fit with the results.

Line 534-538. This interpretation sounds a bit strange. The point of the unreliability of the clinical studies might specifically be related to the fact that DBS does not segregate glutamatergic from GABAergic neurons. The supplementary experiments suggested in the minor comments could help deciding how to phrase this key point. This sentence has been rewritten.

Supplementary figures

Fig S1. I do not get why the insets in a are not in color similar to the neighbouring panels? We have adjusted the legend and the insets are shown in bigger format to aid visualization. The reason why only the viral signal is shown (on black vs white background) is to facilitate visualization (in print and color blind).

Panel C please add here the open field raw data as for the panel b. In Fig S5b, open field data is provided as in panel S5a. We understand the request, but we refrain from it because the locomotor pattern in Vglut2_eD challenged with halo or SCH is similar. In the case of Vgat_eD the locomotor output is different between D1 and D2 antagonized mice, warranting the need to show raw data to the reader. Note that the article has substantially increased in number of figures and panels, and we end up having to limit panel number.

Higher magnifications are needed to illustrate that neurons are labelled. See comment 1 relative to the anatomical controls. We have added higher magnification images together with the c-Fos data set (**Fig.S1b**). Note that virus type/lot is exactly the same that we used in Vglut_eD bilaterally injected. Higher magnification was performed with the LSM900 (a more powerful microscope). Procedure is described on Methods.

Fig S2 (Calcium Imaging supplementary).

Panel c please add statistical analysis for the locomotor deficits. Done. **Fig.S2c**.

Panel d Please define SD (dF/F). Done – see above R#2 notes (this is Z-score).

Line 913. “Paxinos”. Please be more precise on this reference. Done here and throughout the text reference added.

Line 914. Please define “PCA/ICA”. Done in methods (1150-1151) and the acronym is explained in the legend.

Figure S3 (Optogenetics Vglut2_ChR2).

Panel A, D, E, F Please give a name and a unit to the Y axis here and in similar figures elsewhere. [Fig.S3a,d,e] Due to space the y-axis name and unit is placed above each graph. We have increased the text size to aid the viewer. Also, in all other instances.

Panel b. Please specify wavelength. Done.

Line 937. “before and after saline” Should saline be replaced with “drug”? Corrected. Thanks!

Panel f. Please define CnF, PrCnF, MiTg in the legend. Please check similar figures. Done.

Fig S5 (now Fig.S7). See comment 1. Magnification is needed to see whether the transgenes were expressed in neurons, in vgat-positive or vglut-positive neurons. Please see our notes on mouse line specificity: R#1, point 1.

Line 970. Please be more precise on this reference here and elsewhere in the text (“Brain atlas of Paxinos” at least). Done.

Fig S6 Panel E (now Fig.S8). I am not sure this panel is needed. Maybe consider removing it. Has been removed.

Figure S7 Line 1005 (now Fig.S9a). Please specify wavelength. Done, legend.

Methods

Line 1029. Please correct SCH 23390. Done.

Line 1039. Please correct SCH 23390. Done.

Line 579. GRIN LENS. Have the authors evaluated whether the implantation had any effect on locomotor activity? (speed, but also pattern, please see comment 3). Please specify. The hole is spectacular in the brain. We have not evaluated the behavior further than what we show. But we observed that upon recovery and habituation to the miniscope, mice could walk on grids, run corridors and use running wheels. What we can say is that the approach is similar to what others are using for striatum and PPN.

Line 601. Could the authors specify how stable was the field of view using the GRIN LENS during a single day and from one day to another? This is interesting to know. A panel showing neurons identified across experimental days (Halo, SCH tests) has been added (Fig.S2e).

Line 619-622. Why WT mice were not injected with the virus for optogenetics or chemogenetics, but with other viruses? We used different control groups depending on paradigm. Mice that express Cre were injected with the corresponding virus lacking the opsin (AAV5-hSyn1-DIO-mCherry). WT mice, which do not express Cre, were injected (during stereotaxic surgery) with either saline or a virus that induces cre expression (AAVretro-Ef1a-mCherry-IRES-Cre). The animals injected with retro-cre were part of an anatomical check we wanted to perform. They went on to be used on trial experiments after serving as controls on this project. This approach complies with the 3Rs.

Line 661. “The accuracy of this definition was validated by a ground proof scoring of a 2 min video”. Could the authors please rephrase? This sentence was a fragment regarding the Stops definition in the segment Open Field>Data processing. We clarified the issue with the following “Validation of stop bout automatic detection was performed by manually annotating a 2min video clip of each experimental day (ground truth, experimenter was blind to automated annotation, concordance \$93 \pm 3.2\%\$ [SEM between days]).”

Line 663. “strange ‘ before stops”. Solved.

Line 740-741. Is there a reference to justify the exclusions? Probably, but not that we know of. Selecting >15s latency on the Bar test was a decision by the experimenter (made a priori). The reasoning being that the akinetic state can vary, and injections can go wrong (consider how many were done in this article). By probability, there would be errors, thus we decided to be conservative and only add data from mice that were ‘heavily’ akinetic.

Line 747. “Duration individually tailored”. Could the authors please specify the range? We have specified the duration of training sessions using the “dummy” miniscope. Methods>Calcium imaging in the Open Field (1118).

Line 774. “Motion correction algorithm”. Could the authors please define and where to find it? This could be interesting to know. We have added a citation. Note that this algorithm is imbedded in the Inscopix analysis software (citation added, 1142).

Line 785-786. Could the authors please precisely define the exclusion criteria. Done. See the first paragraph after the mathematical formula in the Methods>Calcium imaging data

processing. A paragraph now describes the exclusion criteria as a series of steps 1 to 4 (1153-1161).

Line 824 “ROUT, Q= 1%”. Could the authors please explain? These are statistical report values. All are now described in a paragraph within Methods>Statistics>Statistical methods (1273). This is the statistical approach to check for outliers that uses a nonlinear regression (test named ROUT). The ROUT method is based on the False Discovery Rate specified by Q. Q=1% means we aim for no more than 1% of the identified outliers to be false. Note that this test was not used to exclude data. It was only used to help identify when the data broke with normality assumptions.

Line 836-843. Could the authors please specify the host species for each antibody? Done. In methods (1179) and in the reporting summary.

R#2- REALLY MINOR:

Line 2. A neuron isn't a “who”, even if you develop a close relationship to it during your experiments. Fixed. Mindset has been adjusted as well.

Line 29-30. “These findings suggest...” I do not see how this sentence follows from the preceding sentences. Sentence adjusted.

Scale bar in **Fig 5b** not defined. Done. We also checked all other instances of scale bars.

Line 243. What does random mean? How were these intervals randomised? And why? This was also noted by R1. It is solved. Intertrial intervals were not random but rather between 65 to 75s.

Figs 2D, 3B, 8B, 8E. The *'s should be defined as in reference to the WT^{sal} group. I now see some light gray lines, but indicating in legend might be helpful. Those are the graphs showing percent of time in each speed range (min-to-max boxes). The stats is vs WT^{sal} group (green). We have enhanced the contrast of connecting lines and adjusted the legend.

Line 445. inbuilt??? Word deleted.

Line 461. “experimental confounders” not clear in this sentence. a confounder affects both the dependent and independent variables and could thus lead to the appearance of an association that's not there. I don't think this is what the authors mean. Sentence corrected. “not due to experimental confounders”.

Line 517. tremor treatment is Vim – ventral intermediate nucleus of thalamus. Our mistake. Sentence corrected (814-816). Thank you.

Line 518. “with some success” downplays the success of DBS, which has changed treatment and lives of patients with PD. It is terrific success. But not all symptoms and signs are treated, only the DOPA-responsive ones. Sentence adjusted.

Line 537. “activation” instead of “stimulation” as DBS sometimes inactivates. Corrected.

Methods

Line 555-556. when in this diurnal cycle were experiments done? Between 9am and 5pm. We have added this statement on a new segment Methods>Behavior experiments (928).

Line 580. source of GRIN lenses? Stated. The sentence containing this information is now at the start of Methods>Surgical procedure>Lens implantation (872).

Line 647. confirm frame rate only 25 fps, and indicate camera type and source. Yes, it was 25 fps. This sampling rate is sufficient to acquire data based on center point tracking (above view) and is the rate recommended by Ethovision when tracking 3 body points (nose, center, tail base). If the rate is too high, the noise caused by small trunk movements during gait will be picked up and give an overestimate of dependent variables such as the distance moved (especially if the recording is long). A higher frame rate is needed if the mouse is expected to do fast accelerations (such as in the ‘escape’ Corridor test, there we used 60fps, new Fig.7b-c). In calcium imaging we used 30fps because it facilitates the alignment between Ethovision and the Inscopix data.

Frame rate, camera types and lenses are now stated within methods for each experimental setup.

Line 748. what are the details of the microscope? We adjusted the paragraph by bringing the information closer to this segment of the text. It now reads: “Calcium imaging videos were recorded using Inscopix nVoke acquisition system (v2.0) and software (IDAS, v1.3.1) at 15Hz and using between 0.5 to 1.2mW/mm² LED irradiance”

Line 779. what is the last term in the equation? Should be defined. We believe you refer to $\forall(x,y,t)$, this is to indicate the formula is repeated for every pixel (x,y coordinate) over the entire video (t=frames). The inverted A is a mathematical symbol typically read as “for all”. We adjusted the text (1149).

R#3 MINOR COMMENTS:

Fig. 1b. For how long does the increased locomotor effect last? We have not previously measured this. But in our new experiment for the c-Fos data (see notes on R#1), both Vglut2_eD and Vgat_eD mice were first tested in the Open field (to confirm we could see a locomotor effect, Fig.S1f) and a few days later they were used in the terminal experiment (CNO->perfusion). The graph shows only up to 55min after CNO injection (in line with other data sets) but in fact we recorded this group for up to 2h after CNO (equivalent to time bin 150, shown below) and we can see that the effect starts to wear out after 1.5h from the injection. At this point the difference to WT mice seems to be reducing steadily. Mice are probably ‘tired’ because the CNO effect has been reported to last longer (4h or so). Since we could not address this carefully (we would need a bigger batch), we prefer not to report this in the main text.

Fig. 2d. What do the 2 sets of asterisks in the speed range from 5-10 show? R#2 also had this question. They show the post hoc comparison to WT^{sal+sal} (for Vglut_eD and Sham, treated with halo, then CNO). We have adjusted the legend. Moreover, throughout the figures we added connecting lines that help the reader identify which group is used as reference on the statistical comparisons.

Fig. 5. The title does not reflect the data, not all glutamatergic neurons decreased their activity. Title adjusted (refers to overall population effect).

Fig. 5c. The effects of D1/D2 antagonists on the PPN seem to depend on the previous level of activity of the neuron (*i.e.*, active neurons decrease their activity whereas inactive neurons increase their activity). Is this dependent on the ongoing motor behavior? Could some of these neurons be in the CnF? Yes, you are correct. The maps you see on **Fig.4f** and **5c** are color coded so that each line has in average the value zero (grey color). This allows the reader to better appreciate that some neurons are relatively inactive when the mouse is in drug-free condition (lower lines of each square panel). We agree with your interpretation, active neurons seen in a freely moving animal become inactive after drug injection. We interpret it as being dependent on ongoing motor behavior. It is very interesting that some neurons which were rather silent, engaged after the animal became akinetic (seen in STR and PPN). As an observation, these neurons are not specific to one animal and might act 'in reverse' when we change the drug used. Please see the new panels in **FigS2e-h**. Regarding the possibility that neurons imaged can also be from the CnF, we cannot fully exclude this scenario.

Fig. S1b-c (now Fig.S1g-h). Data should be normalized to account for the different recording lengths. Good point. We have re-graphed here and in all other instances. The data has been normalized to control group (Sham) and is presented as fold change in total distance moved.

Line 2. "is controlling" should read "controls". Corrected. Thank you.

Line 71. How did the authors get 6.059 degrees of freedom for a t-test? The report is correct. The value seems off because we did not state that the test was run with Welch's correction (unequal variance). This is now corrected. Also note that we have added details to the Methods>Statistics>Statistical methods (see notes for R#1). In methods a new sentence reads: "Note also that the degrees of freedom (df) reported in tests with correction are adjusted to control the risk of a type error 1 and may contain decimal cases" (1278-1279).

Line 146-147. "raising the possibility that the direct pathway has unique access to the PPN" Are the authors suggesting that direct pathway MSNs synapse onto PPN neurons? Moreover, given that both antagonists had the same effect on PPN neurons, this statement seems unsupported. We are not considering a direct connection between MSNs and PPN, but rather that the output regions of the BG can maintain a segregation. The sentence has been adjusted.

Reviewers' Comments:

Reviewer #1:

Remarks to the Author:

I would like to congratulate Masini and Kiehn for the excellent revised version of their manuscript. They have addressed all my concerns. There is a considerable amount of interesting new experimental material and new text. Mechanically, I have some new, but very minor comments and suggestions, which should be very easily addressed by the authors (mostly references to add or rewording).

The authors have performed new experiments. C-fos experiments were added to show that chemogenetic activation increased cellular activity specifically in VGAT+ neurons. DeepLabCut-based analyses of hindlimb joint angles were added to describe the intralimb kinematics during chemogenetic stimulation of PPN Vglut2+ or VGAT+ neurons. In vivo experiments requested by another reviewer were added to show that deactivation of CnF Vglut2+ neurons did not prevent the motor benefits associated with PPN Vglut2+ neuron activation. Behavioral data were added to show that mice could adapt their locomotion to a complex environment during MLR stimulation, as recently shown during optogenetic stimulation of CnF Vglut2+ neurons by van der Zouwen et al. 2021 Front Neural Circuits 15:639900. They added reference to previous papers to describe the specificity of the mice used. They added reference to a published control of the effects of chemogenetics at the cellular level (all my apologies to the authors, I did not know Kroeger et al. 2017 J Neurosci). The statistics are now much clearer. They also now clearly discuss the unexpected effects of the activation of PPN VGAT+ neurons in a dedicated paragraph. Altogether, I think that this revised version is superb. There is absolutely no doubt that the paper will be of major interest for the motor control community.

Very minor comments and suggestions:

- Line 213. "follow up analysis revealed that in some instances neuronal response (decrease or increase activity upon drug challenge) could swap between days (Fig.S2h)"
There is no Fig. S2h. Could the authors please add the panel cited? Or maybe it should be Fig. S2g cited in the main text? By the way I do not completely understand the curved line plots in Fig. S2g. It is not clear to me how many neurons are illustrated for instance. Maybe add such information in the legend to improve clarity?
- Line 244, "Fig. S2e-h". Similar comment. There is no Fig. S2h. Could the authors please check the references to other new panels just in case other mistakes need to be corrected?
- Lines 337-338 "CnF activity supports the entire range of speeds both slow gait and fast-paced locomotion"
Could the authors please consider adding reference to Josset et al. 2018 Current Biol 28(6):884-901 and van der Zouwen et al. 2021 Front Neural Circuits 15:639900? These papers strengthen the point of the authors.
- Line 353 "We observed only a mild effect on total distance covered"
The authors show a significant decrease in the distance covered when the CnF was deactivated. To be closer to the data, could the authors please consider rewording as follows: "We observed a mild but significant decrease of the total distance covered".
- Line 384 "493 nm" Could the authors please replace with 593 nm? (I guess this is a typo)
- Lines 353, 367-368, 371-373. The authors have added the "fold increase in distance traveled" in many locations in the paper (e.g. lines 133, 168, 292, etc.). To be homogeneous, it would be informative if the authors could maybe specify i) the fold decrease in distance travelled following CnF deactivation (please consider adding this somewhere in line 353); ii) the fold increase in distance traveled when PPN Vglut2+ are activated while the CnF Vglut2 are deactivated (please consider adding this somewhere in the lines 367-368); iii) the fold increase in distance when activating the PPN while deactivating the CnF under SCH23390 or haloperidol (please add this somewhere in the lines 371-373). This is just a suggestion.

- Line 419 "251%"

Could the authors please convert in "fold increase in distance traveled" to be homogeneous with the rest of the paper?

- Line 428: space missing between "with" and "Vglut2-PPN activation"

- Lines 429-430 "(Speed average when above 20cm/s: for PPN targeted mice was 34.70 ± 2.7 , whereas for CnF it was 55.67 ± 9.2 cm/s [SD])".

Could the authors please specify whether the difference was significant or not?

- Lines 431-432 Were the stops recorded during chemogenetic stimulation of the CnF Vglut2+ neurons longer than those recorded during chemogenetic stimulation of PPN VGAT+ neurons? Could the authors please provide the answer to this question in the results or discussion?

- Line 584 Could the authors please add "what" before "was"?

- Lines 592-596 (especially lines 594-595) and lines 637-652.

The authors added measurements to check whether the animals can navigate complex environment during PPN stimulation. Could the authors please consider adding citation to van der Zouwen et al. 2021 *Frontiers in Neural Circuits* 15:639900, the first study that showed that navigation can be adaptable when encountering an obstacle during optogenetic stimulation of CnF Vglut2+ neurons?

For instance, line 594-595 "if the locomotor movement is performed naturally with the same coordination as non parkinsonian animals and if it is adaptable to environmental changes or needs", could the authors please consider adding ", as was recently demonstrated during optogenetic stimulation of CnF Vglut2+ neurons in freely moving mice (van der Zouwen et al. 2021)".

- Line 594 "Short lasting" please define the duration of the stim, e.g. "a few seconds"

- Line 595. Please correct "To address this issue"

- Line 597. When introducing the analysis of intralimb kinematics, I think it would be fair to add reference to van der Zouwen et al. 2021 *Frontiers in Neural Circuits* 15:639900 10.3389/fncir.2021.639900, who introduced the exact same measurements using DeepLabCut as well, to determine whether hindlimb movements evoked by optogenetic stimulation of CnF Vglut2+ neurons were normal. Also please note that van der Zouwen et al. 2021 was published, so it would be appreciated if the authors could cite the *Frontiers* paper rather than the *Biorxiv* in the present manuscript. Please also note that Fougère et al. 2021 *Biorxiv* <https://doi.org/10.1101/2021.06.13.448213> also used the same analysis in 6-OHDA mice during optogenetic stimulation of CnF Vglut2+ neurons, and thus might deserve to be cited as well, if the authors agree.

- Line 717 Please correct "Our imaging data support"

- Line 723 "For instance, dopaminergic cells in SNc and Zona Incerta project to PPN and may have their locomotor promoting effect through D1 receptors present 723 on PPN neurons"

Could the authors please consider citing Ryczko et al. 2013 *PNAS* 110(34):E3235-42 and Ryczko et al. 2016 *PNAS* 113(17):E2440-9? These articles constitute the first demonstration that descending projections of DA neurons increase MLR activity through D1 receptors in lamprey and that DA neurons of the SNc project down to the PPN in mammals.

- Line 749. Please correct the spacing "Here, we further adjusted"

- Lines 766-769. It would be fair to more explicitly say that optogenetic stimulation of the CnF Vglut2+ neurons also allow mice to perform smooth and adaptable navigation, as first measured and discussed in van der Zouwen et al. 2021 *Front Neural Circuits* 15:639900.

- Line 771 "Finally, the recovery of motor proficiency observed in the obstacle course highlights

that the optical activation of glutamatergic PPN neurons does not lead to automatic or 'robotic' movement, but rather releases movement that is adaptable to a complex environment"

I think it would be fair to more explicitly say in this sentence that such adaptability of MLR-evoked locomotion was first shown in van der Zouwen et al. 2021, who showed that optogenetic stimulation of CnF Vglut2+ neurons controlled locomotor speed without preventing mice from adapting their navigation when encountering an obstacle.

- Line 773: "To the best of our knowledge, no reports of recovery of such strong akinetic phenotype exist in the literature, although there are publications showing that chronic treatments⁷³⁻⁷⁶ can reduce the severity of the phenotype, those are mostly dependent upon pre-treatment approaches".

I think that this could be reworded to be a bit more balanced. The authors may have missed the preprint of Fougère et al. 2021 Biorxiv <https://doi.org/10.1101/2021.06.13.448213>, in which it was found that optogenetic stimulation of glutamatergic neurons in the CnF controls locomotor movements in a 6-OHDA mouse model of Parkinson's disease. Could the authors please consider rewording to mention this study?

- Line 781 and rest of the paragraph. Concerning the slow speed induced by the stimulation of GABA neurons, the authors did a great job at modifying the discussion relative to this point. However, in their Caggiano et al. 2018 Nature paper, they also were speaking about "slow locomotion" when describing the locomotion evoked by PPN glutamatergic neurons. Is the slow locomotion induced by PPN VGAT+ neurons slower than the slow locomotion induced by PPN Vglut2+ neurons? Could the authors please clarify the speed ranges maybe somewhere in these lines?

- Line 794 Please correct "stimulation"

- Lines 823-826 : "The results presented in this study show that localization and cell-specific targeted activation of caudal glutamatergic PPN neurons could provide consistent, and prolonged facilitation of a proficient locomotor output suggesting that this caudal PPN glutamatergic subpopulation should be the prime target for treatment in humans."

I think this sentence should be reworded to be a bit more balanced. The authors may have missed a recent preprint in which control of locomotor speed was obtained using optogenetic stimulation of CnF glutamatergic neurons in 6-OHDA mice (Fougere et al. 2021 Biorxiv <https://doi.org/10.1101/2021.06.13.448213>). Moreover, a recent study in humans based on careful analysis of DBS electrode positioning has shown that the CnF may be responsible for the locomotor benefits observed in good responders (Goetz et al. Neurosurgery 2019 84:506-518). And a recent preliminary clinical trial has shown that DBS of the CnF increases locomotor performance in a patient with levodopa resistant freezing of gait (Chang et al. 2021 Front Hum Neurosci 15:676755).

- Line 958 Please correct "below"

Again, congratulations to the authors for this excellent study.

Best regards

Dimitri Ryczko

Reviewer #2:

Remarks to the Author:

This is a resubmission of a manuscript that I previously reviewed. I applaud the authors for their impressive responses, the text revisions, the new figures, and the new analyses. Together, my previous concerns have been well answered.

Two minor points:

1. Regarding my previous misinterpretation of the figures: It might be helpful to label the colour scale bars to the right of these figures with "% Time" - totally up to the authors.

2. While the authors have done a great job with confidence intervals, effect sizes, etc, they still use SEM. The fact that "everyone uses SEM" is taken for justification for using it does not make it the optimal error to show. I appreciate there is a strong culture behind using SEM (likely based on the fact that the bars are smaller!), but we need to move away from this useless figure. I will leave that up to the journal and its editors.

Reviewer #3:

Remarks to the Author:

The authors have made a remarkable job in addressing the comments by the reviewers with additional experiments and changes throughout the manuscript. I commend this effort and believe that the revised version has been significantly strengthened. The CnF experiments are elegant and increase the confidence of the role of the PPN in the induction of motor activity during dopamine depletion.

There are, however, two main aspects that remain to be properly addressed.

1. Authors and reviewers seem to agree that an acute dopamine manipulation is not a model of PD. Even though the term "PD model" has been removed and the wording has been toned down, the same idea concerning the reproduction of a motor state that resembles the diseased state remains. I am convinced by the authors' arguments in favor of a predominantly striatal effect following the systemic administration of dopamine antagonists, but how this mimics the motor impairment in PD, which develops over decades of adjustments to chronically decreasing levels of dopamine, is far-stretched. A transitory reduction in dopamine release cannot be considered as a "disease phenotype" (line 714). The authors should refer to this experimental state as "acute dopamine depletion" or "reversible dopamine depletion", and not as a "model of parkinsonism" (just in the same way as anesthesia is not a model of sleep).

2. There are important differences in the literature between the data provided here and studies from other groups that must be addressed. While the present study provides clear and robust data indicating a role for glutamatergic and GABAergic PPN neurons in the induction of locomotion, other studies with similar clarity and rigor show contrasting results. Such contrasts are only addressed superficially in one line (744-745). The authors should make a better work at reconciling the current literature with their own data (e.g. Kroeger et al 2017; Josset et al 2018; Carvalho et al 2020; Dautan et al 2021); this growing evidence cannot be simply ignored (see also SfN 2021 abstract by Kaur et al). In fact, Dautan et al showed that the performance of mice in a grid test is impaired during PPN stimulation, which is in opposition to the data of the ladder test where authors show improvement in performance (the grid and ladder tests evaluate similar outcomes). Some clues about the disparity may be explained by the functional heterogeneity among glutamatergic neurons, as recently reported in a paper by the Arber lab (Ferreira-Pinto et al 2021, Cell). Given that glutamatergic PPN neurons innervate and control the firing of dopamine neurons (Galtieri et al 2017), an increased influx of striatal dopamine driven by PPN inputs may counteract the effects of the dopamine antagonists and restore motor activity. In contrast, activation of the SN pars reticulata (the inhibitory output of the basal ganglia) would have the opposite effect by increasing the inhibitory basal ganglia output, as suggested by Ferreira-Pinto et al. These references are central to the present study but notably absent from the discussion. With such divergence in the field it becomes imperative to bring together these studies and create some consensus, particularly considering the implications of these results for the clinics and the need for an integrative approach to future DBS therapies.

Similarly, the discussion about the contrasting results following the manipulation of GABAergic neurons when compared to previous studies is not straightforward and the overall message about the role of GABAergic neurons is ambiguous. The authors explain that motor inhibition occurs only if stimulation is brief, whereas long stimulation promotes locomotion through a local mechanism. If there is evidence for such a mechanism occurring within the PPN please provide a reference. Or

even better, if authors have the data to back this up, please provide them.

Last but not least, it is worth pointing out a recently published study (Fougere et al 2021, PNAS) describing the effects of CnF stimulation in mice with chronic dopamine depletion (6-hydroxidopamine model). It would be pertinent to compare the data between studies in the context of the differences between acute and chronic dopamine depletion.

Other:

Please add data of D1 and D2 antagonist effects to the kinematic analysis of Figure 10. Currently it is only shown that controls and halo/SCH animals with PPN stimulation are not significantly different, but it is important to show the impairment in these tasks following dopamine antagonists administration to appreciate how they improved with PPN stimulation.

I don't understand what Fig. S2e is showing or what is its purpose. Is this purely to represent the number of neurons recorded?

I was unable to access to the video files.

Reviewer #1 (R#1)

I would like to congratulate Masini and Kiehn for the excellent revised version of their manuscript. They have addressed all my concerns. There is a considerable amount of interesting new experimental material and new text. Mechanically, I have some new, but very minor comments and suggestions, which should be very easily addressed by the authors (mostly references to add or rewording).

The authors have performed new experiments. C-fos experiments were added to show that chemogenetic activation increased cellular activity specifically in VGAT+ neurons (experiments also performed in Vglut2-cre mouse line). DeepLabCut-based analyses of hindlimb joint angles were added to describe the intralimb kinematics during chemogenetic stimulation of PPN Vglut2+ or VGAT+ neurons. *In vivo* experiments requested by another reviewer were added to show that deactivation of CnF Vglut2+ neurons did not prevent the motor benefits associated with PPN Vglut2+ neuron activation. Behavioral data were added to show that mice could adapt their locomotion to a complex environment during MLR stimulation, as recently shown during optogenetic stimulation of CnF Vglut2+ neurons by van der Zouwen et al. 2021 Front Neural Circuits 15:639900. They added reference to previous papers to describe the specificity of the mice used. They added reference to a published control of the effects of chemogenetics at the cellular level (all my apologies to the authors, I did not know Kroeger et al. 2017 J Neurosci). The statistics are now much clearer. They also now clearly discuss the unexpected effects of the activation of PPN VGAT+ neurons in a dedicated paragraph. Altogether, I think that this revised version is superb. There is absolutely no doubt that the paper will be of major interest for the motor control community.

Very minor comments and suggestions:

We note these suggestions only involved minor editorial changes. Many of the points were already implemented in the previous version (clarifications are given in those cases) and therefore do not require any further text adjustment. Below we have outlined a point-by-point response.

- **Line 213:** “follow up analysis revealed that in some instances neuronal response (decrease or increase activity upon drug challenge) could swop between days (Fig.S2h)” There is no Fig. S2h. Could the authors please add the panel cited? Or maybe **it should be Fig. S2g** cited in the main text? By the way I do not completely understand the curved line plots in Fig. S2g. It is not clear to me how many neurons are illustrated for instance. Maybe add such information in the legend to improve clarity?

Thanks for noting this. We were **indeed referring to Fig. S2g**. Corrections to the text have been made. The line thickness in this figure represents the proportion of all neurons within each response type. In text the exact numbers are given (**lines 208-210 and 241-243**). To facilitate this interpretation, we have made adjustments to the legend in S2g (**lines 1421-1422**).

- **Line 244:** “Fig. S2e-h”. Similar comment. There is no Fig. S2h. Could the authors please check the references to other new panels just in case other mistakes need to be corrected?

References are corrected; **now reads as Fig.2e-g** (**lines 213, 235, 244, 260**). All other references to figures have been checked.

- **Lines 337-338:** “CnF activity supports the entire range of speeds both slow gait and fast-paced locomotion” Could the authors please consider adding reference to Josset et al. 2018 Current Biol 28(6):884-901 and van der Zouwen et al. 2021 Front Neural Circuits 15:639900? These papers strengthen the point of the authors.

Review round 2 (received 5th November 2021, resubmitted 16th November 2021)

Both papers were already cited within the text (citations: 17 -> Josset and 85-> Zouwen). As suggested, we have added the reference to Josset et al. 2018 in this sentence within the text, especially because Josset's paper specifically addresses the issue of speed (line 339).

- **Line 353:** "We observed only a mild effect on total distance covered" The authors show a significant decrease in the distance covered when the CnF was deactivated. To be closer to the data, could the authors please consider rewording as follows: "We observed a mild but significant decrease of the total distance covered".

Refers to **Fig. S5c-e, iDREADDs in CnF**. We originally left out the word 'significant' intentionally (though stats are given to the reader immediately after the sentence ends and graph shows the stats report). That was because although there is a statistical significance to total distance moved (over a 50min test), the difference in values observed were minor (reduction of 0.24-fold) and the other variables measured do not accumulate evidence towards emphasizing the change to the reader (distance moved over time bins, locomotor time and speed ranges). We thus wanted to be careful on the interpretation for the biological significance of one variable vs all others. We were also careful in other instances (see for example line 572, regarding Vgat_ChR2: "led to a modest..." followed by stats, lines 574-575).

Nonetheless, we comply with the reviewer's request, please see lines 354-358.

- **Line 384:** "493 nm" Could the authors please replace with 593 nm? (I guess this is a typo).

Corrected. Thank you.

- **Lines 353, 367-368, 371-373:** The authors have added the "fold increase in distance traveled" in many locations in the paper (e.g. lines 133 Vglut2_eD^{halo+CNO}, 168 Vglut2_eD^{SCH+CNO}, 292 Vglut2_ChR2^{halo+opto}, etc.). To be homogeneous, it would be informative if the authors could maybe specify.

- i) the fold decrease in distance travelled following CnF deactivation (please consider adding this somewhere in line 353);
Added (line 356, reduction of 0.24-fold).
- ii) the fold increase in distance traveled when PPN Vglut2+ are activated while the CnF Vglut2 are deactivated (please consider adding this somewhere in the lines 367-368);
Stats for this comparison is now added to text (lines 372-375).
- iii) the fold increase in distance when activating the PPN while deactivating the CnF under SCH23390 or haloperidol (please add this somewhere in the lines 371-373). This is just a suggestion.
We have included a statistical analysis comparing mice with PPN-glutamatergic activation, with or without CnF contribution. See lines 378-385.

- **Line 419:** "251%" Could the authors please convert in "fold increase in distance traveled" to be homogeneous with the rest of the paper?

Refers to **CNO mediated activation of CnF glutamatergic neurons**. We maintain the reported difference as percentage here as it is homogeneous with the rest of the article. We used reports of relative change (%) whenever measures of total distance come from long observation windows (50min, see lines 426-428 and 480-483). Both reports (relative/absolute) are directly translatable and easy for the reader to interpret. No changes needed.

- **Line 428:** space missing between “with” and “Vglut2-PPN activation”.

Corrected. Thank you.

- **Lines 429-430:** “(Speed average when above 20cm/s: for PPN targeted mice was 34.70 ± 2.7 , whereas for CnF it was 55.67 ± 9.2 cm/s [SD])”. Could the authors please specify whether the difference was significant or not?

The difference is highly significant. The data was already graphed in **Fig.S6f** but previously we only compared groups to Sham^{CNO}. We have added stats comparing PPN and CnF targeted mice in text (**lines 439-440**) and added the stats result to **Fig.S6f (line 1519)**.

- **Lines 431-432:** Were the stops recorded during chemogenetic stimulation of the CnF Vglut2+ neurons longer than those recorded during chemogenetic stimulation of PPN VGAT+ neurons? Could the authors please provide the answer to this question in the results or discussion?

Yes, stop bouts observed in chemogenetic experiments with CnF-Vglut2+ neurons were longer than those observed in mice with DREADD mediated excitation of PPN-Vgat+ neurons. These data were already displayed and mentioned in the text: a decrease (-49.6%) in stop bout duration for Vgat_eD mice (**Fig.8f, right panel, line 495 and 525**) vs the increase (+67.2%) in stop bout duration observed in CnF-Vglut2_eD (**Fig. S6g, right panel**)”, **line 440-442 and 1522**. No changes needed.

- **Line 584:** Could the authors please add “what” before “was”?

Corrected.

- **Lines 592-596 (especially lines 594-595) and lines 637-652:** The authors added measurements to check whether the animals can navigate complex environment during PPN stimulation. Could the authors please consider adding citation to van der Zouwen et al. 2021 *Frontiers in Neural Circuits* 15:639900, the first study that showed that navigation can be adaptable when encountering an obstacle during optogenetic stimulation of CnF Vglut2+ neurons? For instance, line 594-595 “if the locomotor movement is performed naturally with the same coordination as non-parkinsonian animals and if it is adaptable to environmental changes or needs”, could the authors please consider adding “as was recently demonstrated during optogenetic stimulation of CnF Vglut2+ neurons in freely moving mice (van der Zouwen et al. 2021)”.

A citation to the paper is already in the text elsewhere (**citation 85**). The work by van der Zouwen et al. 2021 (a work by this reviewer) regards CnF activation in healthy mice and analysis of turns in the corners of the Open field. The fact that CnF or PPN activation in healthy mice promotes controlled locomotion in the Open Field was already shown by us in our *Nature* 2018 and also addressed by kinematic and EMGs in Josset et al. 2018.

The point here regards adaptable locomotion aided by optogenetic activation of PPN *without* CnF contribution. This is a unique scenario that has not been reported elsewhere.

- **Line 594:** “Short lasting” please define the duration of the stim, e.g. “a few seconds”.

Done (**line 593**).

- **Line 595:** Please correct “To address this issue”.

Corrected.

- **Line 597:** When introducing the analysis of intralimb kinematics, I think it would be fair to add reference to van der Zouwen et al. 2021 *Frontiers in Neural Circuits* 15:639900 10.3389/fncir.2021.639900, who introduced the exact same measurements using DeepLabCut as well, to determine whether hindlimb movements evoked by optogenetic stimulation of CnF Vglut2+ neurons were normal. Also please note that van der Zouwen et al. 2021 was published, so it would be appreciated if the authors could cite the *Frontiers* paper rather than the Biorxiv in the present manuscript. Please also note that Fougère et al. 2021 Biorxiv also used the same analysis in 6-OHDA mice during optogenetic stimulation of CnF Vglut2+ neurons, and thus might deserve to be cited as well, if the authors agree.

We fixed the reference to Zouwen et al. as it was published (citation 85).

Yet the use of kinematics analysis is common to many labs and the measurements extracted/graphed date back to our own publications (Carmelo and Kiehn 2015, *Current Biology* and Bouvier...Kiehn, *Cell* 2015) and more recently (Allodi ...Kiehn 2021 *Nature Com* – published in May 2021 but submitted august 2020) where we also used DeepLabCut (for the exact same analysis) in an ALS mouse model. We also used DeepLabCut in Cregg ...Kiehn *Nature Neuroscience* 2020. Furthermore, Josset et al. 2018 also used kinematic for MLR analysis of gaits. Thus, we believe it is a stretch to claim those measures were introduced by Zouwen et al. Moreover, the data we analyzed here is unique as they are extracted from animals moving *at will* and in the Open Field, not in a corridor or a treadmill.

- **Line 717:** Please correct “Our imaging data support”.

Corrected.

- **Line 723:** “For instance, dopaminergic cells in SNc and Zona Incerta project to PPN and may have their locomotor promoting effect through D1 receptors present on PPN neurons” Could the authors please consider citing Ryczko et al. 2013 *PNAS* 110(34):E3235-42 and Ryczko et al. 2016 *PNAS* 113(17):E2440-9? These articles constitute the first demonstration that descending projections of DA neurons increase MLR activity through D1 receptors in lamprey and that DA neurons of the SNc project down to the PPN in mammals.

We already cited this work. Please see citation 28, the mini-review article entitled “Dopamine and the Brainstem Locomotor Networks: From Lamprey to Human” by Ryczko.

- **Line 749:** Please correct the spacing “Here, we further adjusted”.

Corrected. Thank you.

- **Lines 766-769:** It would be fair to more explicitly say that optogenetic stimulation of the CnF Vglut2+ neurons also allow mice to perform smooth and adaptable navigation, as first measured and discussed in van der Zouwen et al. 2021 *Front Neural Circuits*.

See response above. The work from Zouwen ...Ryczko 2021 is already cited (citation 85).

- **Line 771:** “Finally, the recovery of motor proficiency observed in the obstacle course highlights that the optical activation of glutamatergic PPN neurons does not lead to automatic or ‘robotic’ movement, but rather releases movement that is adaptable to a complex environment” I think it would be fair to more explicitly say in this sentence that such adaptability of MLR-evoked locomotion was first shown in van der Zouwen et al. 2021, who showed that optogenetic stimulation of CnF Vglut2+ neurons

Review round 2 (received 5th November 2021, resubmitted 16th November 2021)

controlled locomotor speed without preventing mice from adapting their navigation when encountering an obstacle (Open Field corner turns).

We don't think Zouwen ...Ryczko 2021 can take credit or priority for this test. This obstacle course was designed specifically for our experiments. It has never been published before and there is nothing like that in the literature. The text states: "For this purpose, mice were exposed to complex environment containing 3 obstacles that it needed to pass (i.e., Obstacle test): a rod that rotates over a free base, a slalom that requires a zig-zag motion, and a set of stairs with ascending and descending steps separated by a central gap (Fig.10f-g)."

- **Line 773:** "To the best of our knowledge, no reports of recovery of such strong akinetic phenotype exist in the literature, although there are publications showing that chronic treatments can reduce the severity of the phenotype, those are mostly dependent upon pre-treatment approaches". I think that this could be reworded to be a bit more balanced. The authors may have missed the preprint of Fougère et al. 2021, in which it was found that optogenetic stimulation of glutamatergic neurons in the CnF controls locomotor movements in a 6-OHDA mouse model of Parkinson's disease. Could the authors please consider rewording to mention this study?

Our sentence refers to the fact that extensive recovery of movement in mice with pharmacologically induced acute dopamine signaling depletion (via haloperidol and SCH23390) is known to be remarkably difficult ...no acute approach can rescue mobility to the extent we did in these animals. To recover mobility in these mice, scientists rely in prolonged treatment approaches (citations 86 to 89). The requested change to our sentence is therefore not adequate as we are clearly referring to our pharmacological approach to induce akinesia.

Note that do cite Fougère... Ryczko et al. 2021 work in several places now: citation 66, lines 795-799.

- **Line 781 and rest of the paragraph:** Concerning the slow speed induced by the stimulation of GABA neurons, the authors did a great job at modifying the discussion relative to this point. However, in their Caggiano et al. 2018 Nature paper, they also were speaking about "slow locomotion" when describing the locomotion evoked by PPN glutamatergic neurons. Is the slow locomotion induced by PPN VGAT+ neurons slower than the slow locomotion induced by PPN Vglut2+ neurons? Could the authors please clarify the speed ranges maybe somewhere in these lines?

In this article we compared the speed ranges of Vglut_eD and Vgat_eD with the same experiments and analysis method. We were careful to highlight the speed range differences between these groups and WT mice (Fig. 1d and 8d). PPN-glutamatergic activation favored speeds between 10-20cm/s (line 92) and GABAergic favored 5-10cm/s (as already stated in lines 501-502 and 515-516). In the Nature article the "slow locomotion" sentence is used as a comparison between PPN and CNF stimulation, concluding that PPN drives exploratory speed ranges.

- **Line 794:** Please correct "stimulation".

Corrected.

- **Lines 823-826:** "The results presented in this study show that localization and cell-specific targeted activation of caudal glutamatergic PPN neurons could provide consistent, and prolonged facilitation of a proficient locomotor output suggesting that this caudal PPN glutamatergic subpopulation should be the prime target for treatment in humans." I think this sentence should be reworded to be a bit more balanced. The authors may have missed a recent preprint in which control of locomotor speed was obtained using optogenetic stimulation of CnF glutamatergic neurons in 6-OHDA mice (Fougere

Review round 2 (received 5th November 2021, resubmitted 16th November 2021)

et al. 2021). Moreover, a recent study in humans based on careful analysis of DBS electrode positioning has shown that the CnF may be responsible for the locomotor benefits observed in good responders (Goetz et al. Neurosurgery 2019 84:506–518). And a recent preliminary clinical trial has shown that DBS of the CnF increases locomotor performance in a patient with levodopa resistant freezing of gait (Chang et al. 2021 Front Hum Neurosci 15:676755).

We would like to point out that the sentence refers to the specific results presented in this paper. namely that: caudal glutamatergic PPN neurons could provide consistent, and prolonged facilitation of a proficient locomotor output suggesting that this caudal PPN glutamatergic subpopulation could be the prime target for treatment in humans. Again, we do not say that CnF stimulation will not work, but we limited our statements only to what we have tested, and that is that CnF is not involved on the results we observed.

We have a feeling that referee is overstating what the human literature regarding PD has actually shown:

The Goetz et al. Neurosurgery paper includes 11 patients and in their own words: *“Most effective DBS electrode contacts to treat FOG in PD patients were located in the posterior part of the cMRF (encompassing the posterior PPN and cuneiform nucleus) at the level of the pontomesencephalic junction.”* Moreover, there was uncertainty about the precise site, which was discussed in a letter to the editor (10.1093/neuros/nyy516).

The Chang et al. 2021 Front Hum Neurosci paper does not show any clear locomotor improvement in the one patient investigated with CnF stimulation, they only report that rhythmic activation of muscles was obtained. In their own words: *“DTI-based targeting and intraoperative stimulation to evoke limb EMG activity may be useful methods to help target the CnF accurately and safely in patients. Long term follow-up and detailed gait testing of patients undergoing CnF stimulation will be necessary to confirm the effects on FOG.”*

Given the limited evidence on CnF targeting and in light of the much larger literature focusing on PPN stimulation in humans. We have included a statement citing Fougere, Goetz and others, but in relationship to the diversity of neurons in CnF (glutamatergic and GABAergic) and the problem with functional activation (line 876-879).

- **Line 958:** Please correct “below”.

Corrected.

Again, congratulations to the authors for this excellent study.

Best regards, Dimitri Ryczko

Reviewer #2 (R#2)

This is a resubmission of a manuscript that I previously reviewed. I applaud the authors for their impressive responses, the text revisions, the new figures, and the new analyses. Together, my previous concerns have been well answered.

Two minor points:

1. Regarding my previous misinterpretation of the figures: It might be helpful to **label the colour scale bars to the right of these figures** with "% Time" - totally up to the authors.

Done (Fig. 1d, 2d, 3d, S5e, S6e, 8d, 9b,e). We make an apology for not having taken this initiative earlier. We have highlighted the changes in yellow on all figures.

2. While the authors have done a great job with confidence intervals, effect sizes, etc, they still use SEM. The fact that "everyone uses SEM" is taken for justification for using it does not make it the optimal error to show. I appreciate there is a strong culture behind using SEM (likely based on the fact that the bars are smaller!), but we need to move away from this useless figure. I will leave that up to the journal and its editors.

We apologize if we were not clear in the previous response. We did indeed change from SEM to SD/CI in all graphs except one: distance moved over time bins/epochs. However, this graph is always accompanied by a graph with total distance or fold differences, which contains all data points. The statistical tests are the measure used to evaluate differences and they were carefully selected to include intra-individual and group variations. If the Journal insists that we change this specific graph type, we will do it. But please consider that it will not change the conclusions we draw, since the most influential step to fair data reporting is the correct choice of statistics.

Reviewer #3 (R#3)

The authors have made a remarkable job in addressing the comments by the reviewers with additional experiments and changes throughout the manuscript. I commend this effort and believe that the revised version has been significantly strengthened. The CnF experiments are elegant and increase the confidence of the role of the PPN in the induction of motor activity during dopamine depletion. There are, however, two main aspects that remain to be properly addressed.

1. Authors and reviewers seem to agree that an acute dopamine manipulation is not a model of PD. Even though the term "PD model" has been removed and the wording has been toned down, the same idea concerning the reproduction of a motor state that resembles the diseased state remains. I am convinced by the authors' arguments in favor of a predominantly striatal effect following the systemic administration of dopamine antagonists, but how this mimics the motor impairment in PD, which develops over decades of adjustments to chronically decreasing levels of dopamine, is far-stretched. A transitory reduction in dopamine release cannot be considered as a "disease phenotype" (line 714, text was adjusted). The authors should refer to this experimental state as "acute dopamine depletion" or "reversible dopamine depletion", and not as a "model of parkinsonism" (just in the same way as anesthesia is not a model of sleep).

We do refer to our model as an acute dopamine depletion and not a chronic PD model. We have further adjusted lines 719-740 (model discussion) and a few other instances to be completely consistent (marked in yellow throughout the text). We also highlighted the text within figures that refers to the drug used (now text such as "**D1 antagonism**" are in bold [Figs. 2, 3, 6, 7, 9], marked in yellow on the figures).

The first author of this article has worked for 5 years with 6-OHDA and MPTP models in Gilberto Fisone's lab at Karolinska Institute (1. Masini et al 2021, Biomedicines 10.3390/biomedicines9060598; 2.

Review round 2 (received 5th November 2021, resubmitted 16th November 2021)

Masini et al 2018, *Frontiers in Neur.* 10.3389/fneur.2018.00208; 3. Masini et al 2017, *Translational Psych.* 10.1038/tp.2017.58; 4. Val Cervo... Masini, 2017 *Nature Biotechnology* 10.1038/nbt.3835; 5. Bonito-Oliva, Masini 2014 *Front. Behav. Neurosci.* 10.3389/fnbeh.2014.00290; and 6. PhD thesis).

We did therefore seriously consider these model approaches for the present study.

The MPTP model was excluded because of the high variability in DA-depletion and low reproducibility of motor symptoms, which tend to fade-off as compensatory changes occur.

The 6-OHDA mouse with bilateral injection in STR or MFB, upon reaching a stable and chronic depleted state (3-4 weeks after injection, 70-80% neuronal depletion) does not show a clear locomotor phenotype and is used to study non-motor symptoms in PD (for example: Masini, D., et al. *Biomedicines* 2021). We know that the disease late stage encompasses a bilateral and severe depletion of neurons in the SNc. However, bilateral lesions larger than 80% in mice are nearly incompatible with survival (for example: Ferro, M. M., et al. *J Neurosci Methods*, 2005). Mice die before the full degenerative takes place. Due to these 2 constrains, the bilateral 6-OHDA approach could not be used for the purpose of testing locomotor recovery in our study.

The remaining option would be to use the unilateral 6-OHDA full lesion approach, which leads to a well characterized set of motor imbalances (*i.e.*, rotational behavior). This unilateral model is a hallmark in PD research and continues to shine light on nuanced changes in circuitry caused by SNc degeneration (see for example the preprint from Linda Kim 2021, *BioRxiv* 10.1101/2021.09.01.458438). Yet, the asymmetric motor output, to our judgment, is not adequate for studying PPN induced motor recovery, because the mouse would most likely be displaying rotational bias, due to lesion imbalance, as they do when they are pharmacologically treated.

An increasing number of studies is using 6-OHDA injected mice - either injected in STR or MFB – and investigated during the first week after the injection (Fougere et al. 2021; Watson et al. 2021 *Science Advance*). However, studies performed this early after 6-OHDA injection are dealing with an ACUTE depletion, where degenerative processes are still taking place (citations 59-62). Moreover, it is well known that bilateral 6-OHDA lesions have high mortality rates (especially during the first week, citations 68, 69) and require intense post-surgical care. Upon lesion, mice show a syndrome strikingly similar to the '*hypothalamic syndrome*', which is not present in PD patients. Testing animals at this early-stage is therefore not a chronic model and may be confounded by other factors.

The pharmacological signaling depletion we use here is an acute state and we have fairly described the limitations of it in our text.

In summary, we would like the reviewers to acknowledge these considerations and our modeling choice. We have carefully discussed the options and abandoned the 6-OHDA as a useful model for studying recovery of chronic akinetic motor symptoms. It is likely that future PD models will show clear motor symptoms in the chronic phase but right now they are not available to us.

2. There are important differences in the literature between the data provided here and studies from other groups that must be addressed. While the present study provides clear and robust data indicating a role for glutamatergic and GABAergic PPN neurons in the induction of locomotion, other studies with similar clarity and rigor show contrasting results. Such contrasts are only addressed superficially in one line (744-745). The authors should make a better work at **reconciling the current literature with their own data** (e.g. Kroeger et al 2017; Josset et al 2018; Carvalho et al 2020; Dautan et al 2021); this growing evidence cannot be simply ignored (see also SfN 2021 abstract by Kaur et al,). In fact, Dautan et al showed that the performance of mice in a grid test is impaired during PPN

stimulation, which is in opposition to the data of the ladder test where authors show improvement in performance (the grid and ladder tests evaluate similar outcomes). Some clues about the disparity may be explained by the functional heterogeneity among glutamatergic neurons, as recently reported in a paper by the Arber lab (Ferreira-Pinto et al 2021, Cell). Given that glutamatergic PPN neurons innervate and control the firing of dopamine neurons (Galtieri et al 2017), an increased influx of striatal dopamine driven by PPN inputs may counteract the effects of the dopamine antagonists and restore motor activity. In contrast, activation of the SN pars reticulata (the inhibitory output of the basal ganglia) would have the opposite effect by increasing the inhibitory basal ganglia output, as suggested by Ferreira-Pinto et al. These references are central to the present study but notably absent from the discussion. **With such divergence in the field, it becomes imperative to bring together these studies and create some consensus**, particularly considering the implications of these results for the clinics and the need for an integrative approach to future DBS therapies.

We thank the reviewer for bringing this topic forward. We agree that it is important to try to reconcile the PPN stimulation and we have extended the discussion. We fully agree that the disparities may be explained by the functional heterogeneity among glutamatergic neurons related to different target strategies and stimulation protocols. We apologize for not being clear about this issue in the first revision. We further stress that it is the **caudal PPN** that should be the target for stimulation.

Please see **lines 762-786 (text below)**, we added a new paragraph focused on the discussion on rostral vs caudal PPN targeting. See also legend on **Fig.S3h**.

“Recent studies in healthy rodents by us and others have shown that activation of glutamatergic PPN neurons can initiate and maintain locomotion primarily within the exploratory speed range^{18,19,29} which would make this cell type an ideal candidate for locomotor recovery in PD. However, other studies have been unable to see locomotor initiation by optogenetic or chemogenetic stimulation of PPN-Vglut2 neurons^{17,79,80}. In these studies, the optogenetic stimulation instead elicited phasic¹⁷ or tonic muscle activity⁷⁹ in resting animals or decreased locomotor speed during ongoing locomotion¹⁷ while chemogenetic activation of PPN-Vglut2 neurons did not change locomotor distance travelled in open field but did increase the amount of wheel running⁸⁰. These discrepancies may seem difficult to reconcile but likely reflect activation of functional heterogeneous glutamatergic PPN neurons. A recent study in mice has indeed shown that subpopulations of glutamatergic neurons in PPN with axonal projection to either spinal cord, medulla or the substantia nigra reticularis may be related to diverse motor actions, including body extension, locomotion or rearing⁸¹. Moreover, we have shown that optogenetic stimulation of glutamatergic PPN neurons that express the transcription factor Chx10 produce instantaneous full body motor arrest including arrest of locomotion^{33,82}. These PPN Vglut2-Chx10 neurons are located in the rostral most part but absent from the caudal PPN region. Activation of the PPN Vglut2-Chx10 neurons never induces locomotion. Vglut2 targeting of neurons in PPN that involve the rostral and caudal part of PPN will therefore produce motor responses that depend on the site of stimulation with rostral inducing motor arrest

Review round 2 (received 5th November 2021, resubmitted 16th November 2021)

and caudal promoting locomotion (see²⁹ with data from rats). Targeted caudal PPN DBS has also been shown to have the best clinical results^{26-28,50}. In the present study we therefore aimed at targeting the viral expression to the caudal PPN similar to what we did in our previous study using optogenetics¹⁸. Here, we further adjusted the fiber position so that light would only reach the caudal PPN (see Fig.S3f-h) and used 40 Hz stimulation to activate PPN neurons. With this approach we observed that prolonged (10s) activation of caudal glutamatergic PPN neurons consistently promotes a sustained increase in locomotor output and that optogenetic stimulation increases locomotion only during stimulation. The prolonged stimulation caused locomotor initiation in nearly every trial, even more reliable than we observed with shorter stimulation tested previously¹⁸, possibly because of the long stimulation duration (10s) and high stimulation frequency”

Similarly, the discussion about the contrasting results following the manipulation of GABAergic neurons when compared to previous studies is not straightforward and the overall message about the role of GABAergic neurons is ambiguous. The authors explain that motor inhibition occurs only if stimulation is brief, whereas long stimulation promotes locomotion through a local mechanism. **If there is evidence for such a mechanism occurring within the PPN please provide a reference. Or even better, if authors have the data to back this up, please provide them.**

We did include a long segment about the GABAergic effect in the discussion as requested by this reviewer in the previous revision. **We have expanded this section** and added a clear statement that we do not completely understand the cellular or network mechanism for this effect. However, motor inhibition occurs only if stimulation is brief which leads to stops, possibly mediated by activating of local GABA to glutamatergic PPN neuron circuits. It is well known that there are local GABAergic projections in PPN so that could explain the short stop. The long-lasting locomotor promoting effect was also a surprise to us – and we say that now – and we are unable provide a precise mechanism for this effect, at the moment. We have an idea which we now mention directly (regarding STN connectivity). We could have chosen not to report this data – but find that this would be right. We currently reported all that we do know. Please see **lines 818-838 (text below)**.

“In healthy mice, prolonged activation of caudal GABAergic PPN neurons increases locomotor distance travelled albeit constrained to slow ranges of speed and interrupted by frequent brief stops. The increased distance travelled was surprising since previous reports from us and others showed that short-lasting light activation^{18,19} of GABAergic PPN was unable to evoke locomotion or that it decreased ongoing speeds. However, here we found that the locomotor initiation from rest had a long latency which explains why it was not detected with short-lasting stimuli. Moreover, the prolonged locomotion was interrupted by frequent stops and slow speed (5-10cm/s) of locomotion. Therefore, it appears that the prolonged stimulation initiates a mixed effect of long-latency locomotor initiation superimposed by short stops. The network mechanism for the effects of GABAergic PPN neuron activation are not easily explained but they might originate from activation of intrinsic local-PPN⁹⁰ and/or PPN-BG connectivity⁵¹. Brief stops could arise from local GABAergic inhibition of the

Review round 2 (received 5th November 2021, resubmitted 16th November 2021)

glutamatergic PPN population whereas long-range projections to, among others, excitatory subthalamic nucleus (STN)^{77,91,92} could promote the initiation of movement⁹³. In accordance, after haloperidol STN neurons in the indirect pathway are expected to be strongly active and their inhibition by GABAergic PPN neurons would therefore have a rescuing effect whereas when the direct pathway is silenced (SCH23390) locomotion cannot be improved by stimulating GABAergic PPN neurons. In support of his suggestion is that procedures that reduce STN output have been found to reverse the behavioral effects of dopamine depletion in rodents^{94,95}, primates⁹⁶, and humans⁹⁷. A concomitant activation of long-range and intrinsic connectivity may therefore explain the GABAergic PPN behavioral phenotype.

Whatever the precise mechanism is, it is the continuous re-engagement in walking gait, reported here, which opened the possibility that by promoting caudal GABAergic PPN neuronal activity some level of motor recovery could be achieved.”

Last but not least, it is worth pointing out a recently published study (Fougere et al 2021, PNAS) describing the effects of CnF stimulation in mice with chronic dopamine depletion (6-hydroxidopamine model). It would be pertinent to compare the data between studies in the context of the differences between acute and chronic dopamine depletion.

Thanks for bringing this up. We are aware of this publication. This paper *does not* show a chronic depleted state – investigations were done 3-4 days after injection and there is no data showing behavior or anatomical information beyond the first week. Please see our comments on modelling approaches stated above (**R#3, point 1**). We have now added a short section explaining the issues with using the 6-OHDA model as a proxy for motor PD symptoms (**lines 723-730**) and also mentioned the CnF stimulation in the last paragraph (**lines 793-797 and 876-879**) and elsewhere in the manuscript.

Please add data of D1 and D2 antagonist effects to the kinematic analysis of **Figure 10**. Currently it is only shown that controls and halo/SCH animals with PPN stimulation are not significantly different, but it is important to show the impairment in these tasks following dopamine antagonists administration to appreciate how they improved with PPN stimulation.

Unfortunately, these mice are highly akinetic. We did try to analyze their gait, but the number of spontaneous locomotor events was very low and therefore the number of samples was insufficient to make an appropriate gait analysis. The purpose of the kinematic analysis was to show that the rescued gait is comparable to gait in healthy wild type (as requested previously by R#1). We believe that we have succeeded in doing so and that the data fully support this conjecture.

I don't understand what **Fig. S2e** is showing or what is its purpose. Is this purely to represent the number of neurons recorded?

Yes. This figure was made based on a previous request from R#2 and it shows neurons that could be followed between experimental days (green lines that bifurcate) or where only found in one experimental day (black lines, which are unique). We increased the legend size.

Review round 2 (received 5th November 2021, resubmitted 16th November 2021)

I was unable to access to the video files.

This is unfortunate. We have added the videos again on this review round. If the problem remains, please reach out to the editor.

Reviewers' Comments:

Reviewer #1:

Remarks to the Author:

I would like to sincerely congratulate Masini and Kiehn for their second revision. They successfully addressed all my minor comments. To me, their study is a must read for the motor control and clinical neuroscience communities.

Best regards,
Dimitri Ryczko

Reviewer #2:

Remarks to the Author:

The authors have done a tremendous job responding to all the reviewers' criticisms, in my view. I see no reason to delay publication any further.

Reviewer #3:

Remarks to the Author:

The authors have addressed my concerns and expanded the discussion which now compares across recent related studies. Overall, this is a careful and elegant study. I fully support its publication in Nature Communications.

Review round 3 (received 16th December 2021)

We would like to thank the reviewers. We understand the time and effort you placed on reviewing this study. The feedback given was greatly appreciated and your suggestions helped strengthen the work. Thanks to your insightful comments the manuscript improved not only scientifically but also in clarity and readability.

Best regards,

Débora Masini and Ole Kiehn

Reviewer #1 (R#1)

I would like to sincerely congratulate Masini and Kiehn for their second revision. They successfully addressed all my minor comments. To me, their study is a must read for the motor control and clinical neuroscience communities.

Best regards,
Dimitri Ryczko

We thank the reviewer.

Reviewer #2 (R#2)

The authors have done a tremendous job responding to all the reviewers' criticisms, in my view. I see no reason to delay publication any further.

We thank the reviewer.

Reviewer #3 (R#3)

The authors have addressed my concerns and expanded the discussion which now compares across recent related studies. Overall, this is a careful and elegant study. I fully support its publication in Nature Communications.

We thank the reviewer.